# The xIV-LDDMM toolkit of image-varifold based technologies for mapping 3D images and spatial-omics across scales

Kaitlin M. Stouffer [1,2,3,4] ✉, Xiaoyin Chen [5], Hongkui Zeng [5], Benjamin Charlier[6], Laurent Younes [1,7], Alain Trouvé [4] & Michael I. Miller [1,2,3]

Advancements in imaging and molecular techniques enable the collection of subcellular-scale data. Diversity in measured features, resolution, and physical scope of capture across technologies and experimental protocols pose numerous challenges to integrating data with reference coordinate systems and across scales. This paper describes a collection of technologies that we have developed for mapping data across scales and modalities, such as genes to tissues, specifically in a 3D setting. Our collection of technologies include (i) an explicit censored data representation for the partial matching problem mapping whole brains to subsampled subvolumes, (ii) a multi, scale-space optimization technology for generating resampling grids optimized to represent spatial geometry at fixed complexities, and (iii) mutual-information based functional feature selection. We integrate these technologies with our cross-modality mapping algorithm through the use of image-varifold measure norms to represent universally data across scales and imaging modalities. Collectively, these methods afford efficient representations of peta-scale imagery providing the algorithms for mapping from the nano to millimeter scales, which we term cross-modality image-varifold LDDMM (xIV-LDDMM).

Emerging technologies in spatial transcriptomics coupled with advancements in imaging over the last decade are continuing to burgeon the scale and scope of biological data that can be interrogated. The uniqueness in phenomenological angles each method offers coupled to choice of employment amidst different species, across different tissue windows, and under different experimental conditions afford incredible breadth of data concurrently with the much-needed depth and replicate numbers that are often limited in what remain resource-heavy endeavors. Nevertheless, both the sheer amount of data coupled with the diversity in measured features, resolution, and physical window of capture across technologies and experiments pose challenges to integrating data across them and with reference coordinate systems at coarser scales. This integration is necessary, however, for downstream analysis and fully gleaning what understanding these datasets in combination might elucidate.

With regard to magnitude, detections are occurring at increasingly high resolutions (e.g. individual mRNA at submicron scale) and covering measurements of increasingly large feature sets (e.g. 20,000 different genes in the full mouse genome)[1], which often reach tera or peta-scale complexity as 3D image representations. Technologies are typically categorized into

those which are spot-based (e.g. SlideSeq, Visium) versus those which are image-based (e.g. MERFISH, BARseq). The former group prioritizes feature set size at the expense of achieving only near-single cell resolution while the latter prioritizes single and even subcellular resolution at the expense of reducing the measured feature set (e.g. <1000 genes).

For instance, in a single mouse coronal section measuring 500 different genes with MERFISH technology[2,3], typically on the order of 100 million mRNA transcripts might be detected. With sections taken at every 100 micron and spanning about 65–75% of the mouse brain anterior to posterior, the total number of mRNA detections across the resulting 50 sections is on the order of 6 billion. Covering a volume of about $10 \times 10 \times 10$ mm$^3$, representation of these 6 billion mRNA with even a 1-μm voxel grid resolution would thus require 1 trillion (tera-scale) voxels, without even affording complete differentiation of neighboring mRNA from one another.

With regard to diversity, functional measures vary with both scale and type, with subsets of possible features often selected per experiment given the relevance to the problem at hand. At the tissue scale, measures interrogate both structural properties of anatomy (e.g. MRI, CT) as well as function (e.g. fMRI) that have often been summarized in the form of

[1]Center for Imaging Science, Johns Hopkins University, Baltimore, MD, USA. [2]Department of Biomedical Engineering, Johns Hopkins University, Baltimore, MD, USA. [3]Kavli Neuroscience Discovery Institute, Johns Hopkins University, Baltimore, MD, USA. [4]Centre Borelli, ENS Paris-Saclay, Gif-Sur-Yvette, France. [5]Allen Institute for Brain Science, Seattle, WA, USA. [6]IMAG, Université de Montpellier, CNRS, Montpellier, France. [7]Department of Applied Mathematics and Statistics, Johns Hopkins University, Baltimore, MD, USA. ✉e-mail: kstouff4@jhmi.edu

regional atlases (e.g. Allen CCFv3[4]). Spatial-omics technologies, for instance, operate across scales of molecules to cells, coupling measures from imaging, sequencing, and mass-spectrometry[5] to downstream segmentation and clustering processes[6] to generate either molecule- or cell-based datasets sporting molecule (e.g. gene, protein, metabolite) type or cell type. While many such technologies necessarily capture static assessments of organisms, technologies focused on epigenomics[7] or synapse activity[8] are often coupling such profiles over time to capture the dynamics of both interacting cells/ molecules. Consequently, spatial representations necessitating tera-scale complexity are further expanded even to peta-scale complexity with hundreds of possible functional measures taken statically or over time.

Deep learning methods that might accommodate such data complexity have naturally arisen to analyze spatial transcriptomic data both independently (e.g. for cell typing) or in the context of histological images taken of the same tissue sample[9]. However, many of these methods have continued to suffer from data complexity, necessarily breaking up 2D datasets into smaller patches that then fail to achieve consistency across their boundaries with regard to end analysis[10]. Additionally, with regard to data representation, many of these methods fail to offer a clear biological interpretation as they transform the inherently compositional data from its native space[11] to one that can be treated using Euclidean distance metrics[9].

Molecular Computational Anatomy, born recently out of the theory developed for modeling anatomy at tissue scales, naturally accommodates the diversity in scales, scope, and features attributed to data captures across experiments and technologies. In the Molecular Computational Anatomy model[12,13], brain mapping, or relating one brain to another through a similarity metric, follows the D'Arcy Thompson[14] program, transforming a set of objects that we call varifold measures denoted here as $\mu \in M$ (the set of finite signed measures on a given physical and functional space, $\mathbb{R}^3 \times \mathcal{F}$), with a norm $\|\mu\|_M$ to measure closeness[12,13]. Thompson's brain mapping scheme defines transformations which we model here as diffeomorphisms, $\varphi \in Diff$, transforming one brain to the other with $\mu \to \varphi \cdot \mu$, with the end result being an alignment of these objects in the same coordinate system and measurement of their similarity through the magnitude of the mapping (e.g. diffeomorphism) needed to transform one to the other.

Our molecular representations (varifold measures) at the micron scales are represented mathematically as a set of particles, each particle carrying a singular mass weight located in physical space and a conditional probability distribution over the feature space[15] of for instance, genes or cell types. The brain mapping problem estimating $\varphi$: $\mu \mapsto \varphi \cdot \mu$, aims to minimize the normed distance between brain objects $\mu, \mu' \in \mathcal{M}$, optimizing over diffeomorphic transformations between them:

$$\inf_{\varphi \in Diff} \| \varphi \cdot \mu - \mu' \|_M^2. \tag{1}$$

We call this the image-varifold representation using the varifold norm[12,13,15] which generalizes to many types and diversity of features while efficiently representing the particular sampling schemes (e.g. regularized grids versus innate cell placement) of each technology. Importantly, this approach for measuring similarity departs from most deep-learning based methods rooted in Euclidean space and offers greater diversity of input "images" not just from a single tissue specimen[9]. As measurements and atlasing occurs across many scales, the formalism naturally supports hierarchical representations from tissue to molecular scales[12,13]. Our descriptions throughout carry appropriate spatial scales associated to the smoothing kernels which produce a sequence of successive approximations of greater and greater detail in the dimensions of space-scale and gene/cell feature. Furthermore, we accomplish this without the need for subsetting data into discrete patches separately to analyze and therefore, avoiding the discrepancies in analysis that have resulted from deep-learning methods analyzing similar-sized tissue data (o(1 cm²)) in batches[9]. We exploit this multi-scale representation demonstrating mappings between the sub-micron scales of the transcriptome to the millimeter scales of the CCFv3 and EMAP atlases[4,16].

Furthermore, while most spatial transcriptomics technologies report measures in a 2D landscape, increasingly these planes are representative of single sections out of stacks of planes extracted within a 3D volume[2] or even wholly measured subvolumes with newer technologies achieving depths on the order of hundreds of μm[17]. Accordingly, both of these types of datasets need to be integrated with corresponding 3D volumetric images and atlases. We have previously described the framework of the image-varifold representation for particle representations of spatial-omic measurements[13,15]. The focus of this paper is to solve the alignment problem at the complexity of fully 3D, billion transcript measurements. Accordingly, new technologies are needed for accommodating variants in scope and region of tissue capture as well as the possibly peta-scale complexity of the data when spanning 3 dimensions. We describe here the collection of technologies we have developed to address these challenges: independent modules for censoring, scale-space optimized particle approximations, and optimized feature selection. Rooted in the unifying image-varifold representation of data, these technologies can integrate with our established cross-modality image-varifold based large deformation diffeomorphic metric mapping scheme (xIV-LDDMM[15]) to achieve alignment across scales and modalities in 3D.

## Results
### Overview of technologies
The specific modules described herein center on advancements including: (i) an explicit model for solving the partial matching problem mapping whole brain tissue scale atlases to partial volume censored targets, (ii) scale-space approximation methods for optimized particle approximation of targets from molecular to tissue scales, and (iii) mutual-information based feature selection technologies for optimized selections of genes/cell types for optimized sparse dimension reduction. As independent modules, we describe each in turn, but with an emphasis both on individual output and collective output (see Fig. 1).

The core of our set of technologies is the use of the image-varifold representation to model equivalently tissue-scale imagery and highly resolved molecular scale spatial-omics measurements. These representations are equipped with a norm that defines closeness with regard to feature and physical geometric similarity, departing from the Euclidean metric often used to measure similarity, but which fails to capture inherently the dual spatial and functional nature of the data measured by spatial transcriptomics technologies[9]. Consequently, the modules introduced here of censoring, scale-space resampling, and feature selection, seamlessly feed into our established large deformation diffeomorphic metric mapping schemes (xIV-LDDMM) that are image-varifold based and offer single and cross-modality mapping faculties[15] (see Fig. 1).

The first technology we describe is that of censoring. While current technologies afford 3D data collection as a series of 2D sectional data at high resolution, the scope and extent of tissue (3D volume) spanned by the series of sections delivered often varies with the experimental design. Additionally, spatial-omics techniques are still newly emerging and remain quite costly. Consequently, the vast majority of experimental designs often focus collection on tissue subregions, delivering highly variable and partially sampled portions of target organs (e.g. brains). This poses significant challenges to integration with tissue-scale atlases which cover the 3D volumes in their entirety, and is becoming only more prevalent as spatial-omics technologies are applied in settings of larger magnitude, such as primate brains, where measuring full or even hemi-coronal sections, is in itself, a challenge.

The partial subvolume censoring problem thus arises whenever aligning complete 3D tissue-scale atlases to partial volume captures, rendering it central to alignment problems for spatial-omics. To solve this problem, we introduce in our first technology a spatial censoring function as part of the imaging model that punctures the image-varifold matching norm within the scope of the deforming atlas thereby optimizing alignment over the censored sample. We demonstrate the robustness of this technology for capturing rostral-caudal extent and medial-lateral extent of target data for sets of whole brain and half brain coronal sections of both MERFISH and BARseq data.

Our mapping technologies, which are particle-based, include calculation of inner products which are quadratic in the number of particles, placing us into a solution space that is staggeringly peta-scale in complexity at the full

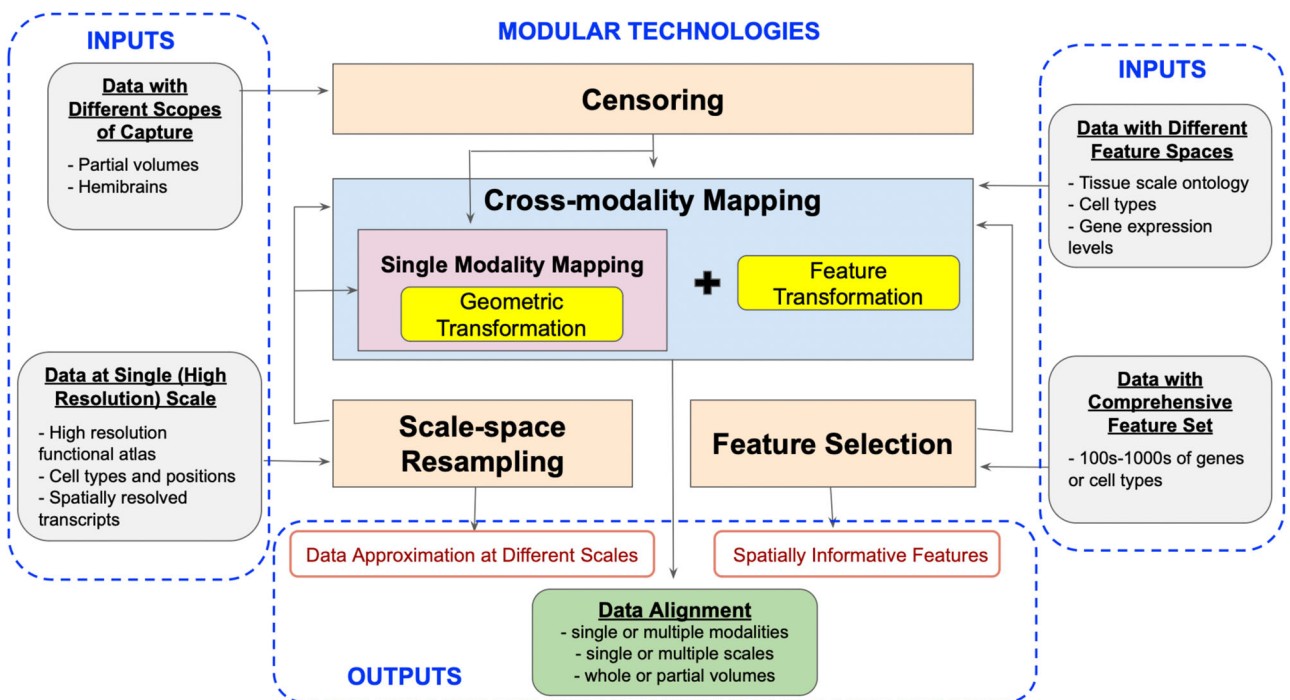

**Fig. 1 | Summary of technologies with designation of input types and examples (left and right), intermediate outputs (red outline) where applicable, and final outputs (green box) resulting from combinations of modular technologies.** xIV-LDDMM (blue box) mapping technology as described[15] offers single-modality mapping based on varifold norms without estimation of feature transformation needed for cross-modality mapping. Censoring, scale-space resampling, and feature selection (orange boxes) denote technologies introduced in this work for accommodating complexities and variability in 3D data. Arrows demonstrate integration of new technologies with single or cross-modality mapping applications.

resolution of unprocessed transcriptomic data. The magnitude of this data that results from integration across sets of sections (e.g. 6 billion mRNA reads and order 1000 genes) present remarkable complexity challenges. Approximation and feature selection remains fundamental to the problem itself. Our second and third technologies address this complexity through reduction along both physical and feature axes, respectively. Specifically, our second technology generates a hierarchy of space-scale approximations, delivering the closest particle solution of a fixed complexity in the gene-cell type approximation metric to the target. Our third technology is an information-based technology for functional feature selection, where dimension reduction is achieved through selection of spatially varying genes/cell types that exhibit a high score based on mutual information.

We demonstrate the efficacy of each of these three modules in the setting of mapping the tissue-scale Allen CCFv3 atlas to cellular and molecular scale target serial sections generated through BARseq and MERFISH technologies, respectively. Generalizability to other technologies and datasets is then demonstrated with the mapping of a whole mouse embryo atlas at day 6.5[16] to a whole mouse embryo at day 6.5–7, sequenced with recently developed technology of cycleHCR[17].

### Data sets: MERFISH, BARseq, and cycleHCR

The results described herein are from four different spatial transcriptomic datasets covering three different technologies. The first is a published MERFISH dataset[2] with 56 coronal sections that cover most of the mouse brain except for the most rostral and caudal part. The second and third were measured with BARseq and comprise a newly collected dataset with 40 coronal sections that cover most of the mouse forebrain, and a set of 8 published datasets[18], each including 32 hemi-coronal sections that cover one hemisphere of the forebrain except the most rostral parts. The fourth dataset is a published measurement of transcripts of 254 genes in a whole mouse embryo aged E6.5–7.0 with cycleHCR technology[17], covering the entire volume at a depth of 310 μm. The MERFISH dataset interrogated about 500 genes which were optimized for distinguishing all cell types across the whole

brain, whereas all BARseq datasets interrogated 104 genes that were optimized for resolving cortical excitatory neuron types, with less emphasis for subcortical cell types. Hence, all three brain datasets are examples of censored subsets relative to the whole brain atlas. Similarly, the 254 selected genes measured by cycleHCR in the whole mouse embryo were lineage-specific, with a goal of identifying differentiating structures early within embryogenesis and primarily intra rather than extra-embryonic.

Our results have focused on demonstrating alignment to atlas coordinates using both cell type labels and gene expression directly. In the brain datasets, the BARseq data were clustered at three different levels[18] to reveal cell type heterogeneity, and we registered the brains using the coarsest level of cell types (52 types in the whole-coronal section dataset and 39 types in the hemi-coronal section datasets). For the MERFISH dataset, we demonstrate results for 3D alignments based on selection of 20 of the most spatially informative genes based on mutual information (see section Feature Selection via Mutual Information). Finally, in the whole mouse embryo, Cellpose was used together with UMAP to identify and cluster cells into a set of 9 distinct types[17], which we map to the EMAP atlas[16], based on ontological annotations from Kaufman[19].

Importantly, each of these datasets reflects capture at a submicron level, with the minimum distance between two data points (e.g. segmented cells) ranging from 0.015 to 0.5 μm (Table 1). Consequently, representation of these datasets as a regularized voxel grid even at 1 μm or 10 μm resolution compromises identifiability of the individual data points, with image-based mapping methods operating on image sizes more typically in the range of 10−30 μm[20]. Additionally, the number of voxels in these representations far outnumbers the data points in the original dataset by $10^3-10^6$, thereby increasing the number of distance computations by $10^6-10^{12}$ in the setting of quadratic kernel calculations.

### Crossing modalities via varifold measures

In each technology described, we unify the molecular and tissue scales using particle (image-varifold) methods based on mathematical measures

**Table 1 | Characteristics of cell-based datasets and comparison of needed data points for particle-based representations versus regularized voxel grid representations**

| Dataset | Min distance between data points (µm) | # Voxels at 1 µm resolution | # Voxels at 10 µm resolution | # Particles (High Res) | # Voxels (200 µm) | # Particles (200 µm) |
|---|---|---|---|---|---|---|
| BARseq | | | | | | |
| Whole brain | 0.1 | $1.59 \times 10^{12}$ | $1.59 \times 10^{9}$ | $3.36 \times 10^{6}$ | $1.96 \times 10^{5}$ | $1.63 \times 10^{5}$ |
| BARseq | | | | | | |
| Hemi-brain | 0.015 | $6.04 \times 10^{11}$ | $6.04 \times 10^{8}$ | $1.00 \times 10^{6}$ | $7.64 \times 10^{4}$ | $6.24 \times 10^{4}$ |
| cycleHCR | | | | | | |
| Embryo | 0.5 | $5.49 \times 10^{7}$ | $5.49 \times 10^{4}$ | $1.10 \times 10^{4}$ | – | – |

constructed from discrete "particle" Diracs, with each particle indexed by $i \in I$. Physical location and functional feature value measured by a given technology are captured together through the construction of a product measure on the product measurable space yielded by each domain (e.g. $\mathbb{R}^3 \times \mathcal{F}$, with $\mathbb{R}^3$ the physical domain captured by the technology and $\mathcal{F}$ the domain of functional values measured). Each particle $i$ carries a singular weighted mass located in physical space $w_i \delta_{x_i}$ and a probability distribution $p_i$ over the feature space[15], which may be genes or cell types or tissue types, resulting in the product measure representation:

$$\mu \doteq \sum_{i \in I} w_i \delta_{x_i} \otimes p_i, \tag{2}$$

Our technology for diffeomorphic mapping of the nano-scales of the transcriptome to the tissue scales of the atlas follows the framework of LDDMM[21] extended to the molecular scales as in[13,15]. We estimate the diffeomorphism $\varphi$ and similitude transformations solving the variational problem minimization of the varifold norm of (3) via LBFGS optimization as adapted from that implemented in PyTorch. Mapping samples measured with a single technology to one another (e.g. sets of coronal sections from two separate mouse brains, imaged with BARseq) proceeds with estimation only of these two transformations via minimization of the varifold norm[22-25].

Unique to mapping molecular/cellular datasets and tissue-scale atlases requires transformation not just in physical space, however, but also between fundamentally distinct feature spaces defined at each respective scale (e.g. gene expression and cell distributions versus functional/anatomical delineations). Indeed, for both cellular (BARseq) and gene-based (MERFISH) datasets used throughout this work, the functional labels differ from each other and with respect to the chosen atlas (CCFv3). For each modality, we represent the functional feature with a probability law on discrete values (e.g. regions, cell types, genes). Closeness between subjects in any modality is then based on a distance (given by the varifold norm as in (3)) between probability laws, independent of the specific feature type[13,15].

Crossing modalities and scales as for aligning atlases to molecular/cellular targets requires us to augment the atlas with the latent variables of molecular/cellular label distributions $p = (p_\ell)_{\ell \in \mathcal{L}}$ with $\mathcal{L}$ the labels in the ontology of the atlas. We assume each region (label) in the atlas is associated to a single probability law over molecular/cellular features that holds over the entire spatial extent of the region. These probability laws are latent dimensions to be estimated at each atlas location on the feature spaces $\mathcal{F}$ of gene or cell type, giving $\mu^p = \sum_i w_i \delta_{x_i} \otimes \sum_\ell \pi_i(\ell) p_\ell$, with $\pi_i(\ell)$, particle $i$'s probability of pertaining to region $\ell$, which for interior (non-boundary) particles, is typically a binary 0/1. The optimization of xIV-LDDMM jointly calculates the diffeomorphism and feature laws representing the target tissue:

$$\inf_{\varphi \in Diff, p} \| \varphi \cdot \mu^p - \mu' \|_M^2 \tag{3}$$

By assuming this stationarity in probability law over each atlas region, correspondence is achieved from the empirical probability laws reconstructed from the atlas and target and the relative purity of the partition, independent of the space that the data lives in. For instance, Fig. 2a depicts a toy matching problem between an atlas parcelled into three regions associated to the feature space $\mathcal{F} = \{R, G, B\}$ and a molecular target with varying expression of two features $\mathcal{F} = \{B, W\}$ across the same tissue span. The correspondence is driven by the inner product between the empirical probability laws themselves, aiming to align homogeneous to homogeneous regions, not the range space of the image value of the domain that the probability law is supported on. Here, the norm distance between deforming atlas and target is thus achieved with 180 degrees of rotation in which square and rectangle boundaries parcelling space into homogeneous atlas regions and regions of homogeneous black and white feature expression are aligned.

In the setting of BARseq cell types, similar regions of homogeneity emerge, for instance, in the cortical layers and areas of the hippocampus where specific excitatory neuron subtypes predominate consistently throughout the region (Fig. 2d). Correspondence between CCFv3 sections (Fig. 2b) is thus achieved with diffeomorphic transformation (Fig. 2c) across the section, expanding (red) and contracting (blue) areas of physical space to align these homogeneous regions while in tandem estimating per each CCFv3 region a probability law over cell types in line with what is seen in the target (Fig. 2d).

The assumption of stationarity we make in this setting stems from the simple principle that has been demonstrated in diverse tissues across organisms that cells close in physical distance (e.g. within the same functional region of tissue) share similarity in transcriptional signature[26]. This assumption is further analogous to that used in deep-learning methods aimed at "spatial domain identification"–identification of presumed functionally and structurally homogenous tissue regions with signatures in histological architecture that we expect correspond to transcriptional signatures[9,27]. Here, we extend this assumption not just from the molecular to cellular scales as evident in the correspondence between transcriptional data and histopathology but also to the tissue scales, as functional atlases and ontologies have often been constructed in conjunction with histopathology. Indeed, this assumption of co-existing and consistent signatures across regions of space at tissue, cellular, and molecular scales is fundamental to diverse efforts within "integrative" structural biology to produce and study multi-scale representations of biology[28].

## Mapping whole brain to partial volume censored targets

For matching onto censored subvolumes, we restrict the subregion of intersection defined by the deforming atlas coordinates and the target. We introduce a set of spatially-varying weights that puncture the deforming atlas so as to select only the subvolume occupied by the target. These weights are given by the censoring function, $\alpha^\lambda : \mathbb{R}^3 \to [0, 1]$ defined apriori in the coordinate system of the molecular/cellular target, with $\alpha_x^\lambda = 1$ for all $x$ within the support of the target and decreasing smoothly to 0. The brain mapping problem optimizes over the diffeomorphism and the support function, with $\lambda$ controlling the smoothness of the support function:

$$\inf_{\varphi \in Diff, \lambda} \| \alpha^\lambda(\varphi \cdot \mu) - \mu' \|_M^2,$$
$$\text{with } \alpha^\lambda(\varphi \cdot \mu) = \sum_{i \in I} \alpha_{\varphi(x_i)}^\lambda |D\varphi|_{x_i} w_i \delta_{\varphi(x_i)} \otimes p_i. \tag{4a}$$

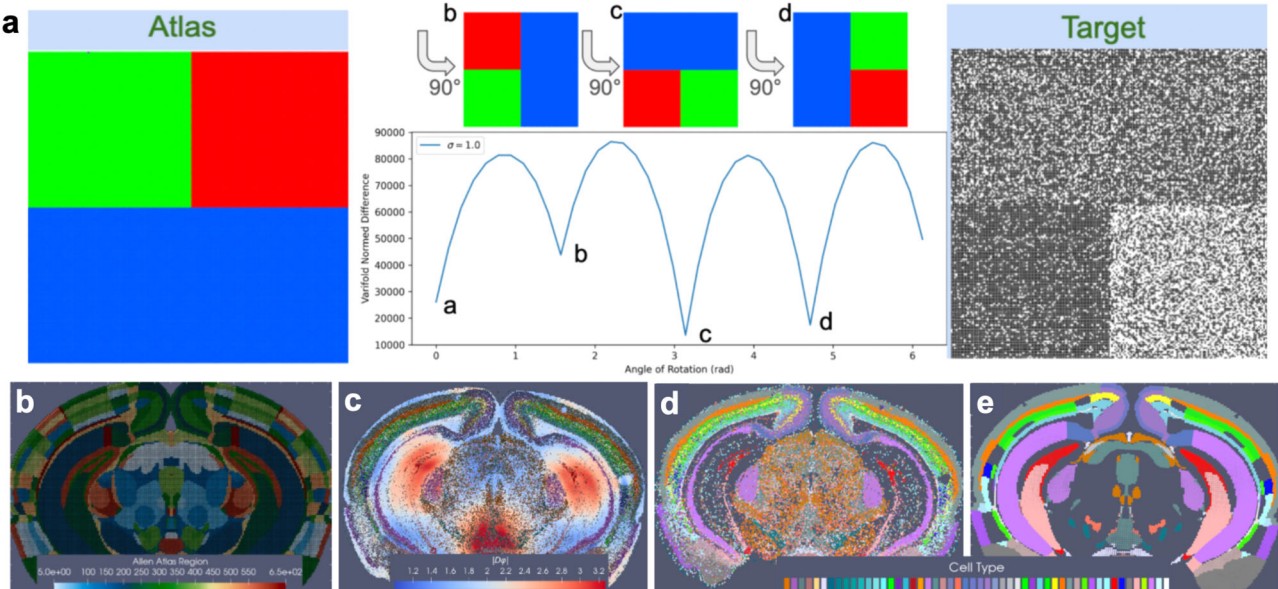

**Fig. 2 | Cross-modality mapping of tissue scale atlas to molecular scale target aligning homogeneous regions in space. a** Shows minimization in varifold norm achieved with 180 degree rotation of atlas to target when homogeneous regions (colors in atlas, black/white probability distributions) in atlas and target are aligned. **b** Shows CCFv3 section 837/1320 (anterior to posterior). **c** Shows transformed CCFv3 section with colors indicating determinant of the Jacobian reflective of areas of expansion (red) and contraction (blue). **d** Shows BARseq measured cells with designated cell type. **e** Shows predicted cell type with highest probability per each transformed CCFv3 partition.

The physical mass of the deforming atlas is given by the product $|D\varphi|_{x_i} w_i$ at each location $\varphi(x_i)$ according to the varifold action of diffeomorphisms[13], with the determinant of the Jacobian $|D\varphi|_{x_i}$ capturing the local change in volume at position $x_i$ as a result of the diffeomorphism. This mass is masked according to $\alpha^\lambda$ at that location to retain atlas regions corresponding to those with measured targets.

Importantly, these support weights allow us to distinguish subvolumes of physical space that were not measured directly in a given experiment from those that were measured but with little to no detection (e.g. gene expression) in the region. This is necessary particularly in the setting of cross-modality mapping between whole brain atlases and partial molecular targets (see Section Crossing Modalities via Varifold Measures) where diffeomorphic transformations are estimated in tandem with target feature distributions that predict both physical density of target mass per atlas region and probability distribution of that mass across features.

In the setting of whole brain coronal sections (Fig. 3b, c, e, f), the support weights are computed as the sum of hyperbolic tangent functions (see Section Estimating the Censoring Function) oriented along the rostral-caudal axis to distinguish the center portion of the CCFv3 brain (red) from the rostral and caudal most parts that were not sectioned for measurement (blue). Initial alignment between the CCFv3 and MERFISH and BARseq serial sections varied grossly in terms of rostral-caudal placement of sections (Fig. 3b) and medial-lateral scale (Fig. 3c), with the intersecting atlas plane largely differing in terms of initial geometry, especially in the setting of MERFISH (Fig. 3h, i). Estimation of support weights facilitates accurate diffeomorphism and similitude estimation, particularly along the rostral-caudal axis in these cases where target sections are then confined to a portion of the whole brain rather than atlas and target needing to span the entire same volumes. This is seen both grossly (Fig. 3e, f) and on a section-by-section basis (Fig. 3l–o) where we see much stronger similarity in geometric compartments (e.g. striatum and cortical layering) following solution of the mapping problem with censoring.

Notably, in addition to measured tissue volumes not needing to span entire volumes of atlases, they also need not be sampled uniformly by the given technology. Indeed, we demonstrate the robustness of our mapping technology for handling disjoint (rather than contiguous) sets of sections with gaps of missing data within the prescribed measured tissue volume (Supplementary Fig. 1). In this setting, unlike in the censoring method here described, we choose not to model these gaps explicitly as unmeasured areas with gaps occurring typically from single or multi-slice dropout and not prescribed apriori by a given experimental setup. Rather, the reduced dataset is used to estimate a similar optimal deformation of the CCFv3 atlas to the whole brain BARseq sections (Supplementary Fig. 1B versus Fig. 3e). The stability of our technologies to estimate maps with full versus fragmented datasets is evidenced by only the slight differences in the extent of deformation observed through comparison of the determinant of the jacobian (Supplementary Fig. 1F versus 1G), and with the near identical appearance of the estimated atlas intersection per a single slice (Supplementary Fig. 1D versus 1E).

Beyond a selected tissue subvolume spanning the rostral-caudal axis, in hemi-brain coronal sections, as measured with BARseq (Fig. 3d, g), support weights additionally contour the irregularly-shaped central volume, with sections covering different extents (e.g. 45–75%) along the medial-lateral axis of the brain. The support function $\alpha_\lambda$ in this setting is estimated as the output of a UNET trained on the given target dataset (class 1), with dummy particles (class 0) placed along the medial boundary of each section. To represent the rostral-caudal extent of the subvolumes which are highly variable, sections of particles are placed rostral and caudal to represent the rostral-most and caudal-most sections, respectively. The final layer of the UNET feeds into a hyperbolic tangent function tanh, with the scaling parameter, $\lambda$, estimated as in the setting of whole brain coronal sections and controlling the bandwith of the transition zone along the smoothly estimated boundary (see Section Estimating the Censoring Function for details).

We estimated particle representations at 200 μm (the given spacing between parallel sections) for two stacks of ≈30 sections from two separate mice, which were initially aligned to within the center of the CCFv3 (Fig. 4a, b). In these hemi-brain samples, hemi-coronal sections varied in span medially to the opposing hemisphere both within brain sample and across brain samples (Fig. 4g–l). Per section, medial-lateral coverage spanned 50–70% of total coronal area. Diffeomorphic maps were estimated at multiple sub millimeter scales (1–0.5 mm, see Section 7 below) including rigid transformations at the coarsest scale to accurately align each set of hemi-sections to right-side hemispheres (Fig. 4c, d). Censoring weights (e.g. Fig. 3d, g) aided in generating optimal diffeomorphisms that exhibited

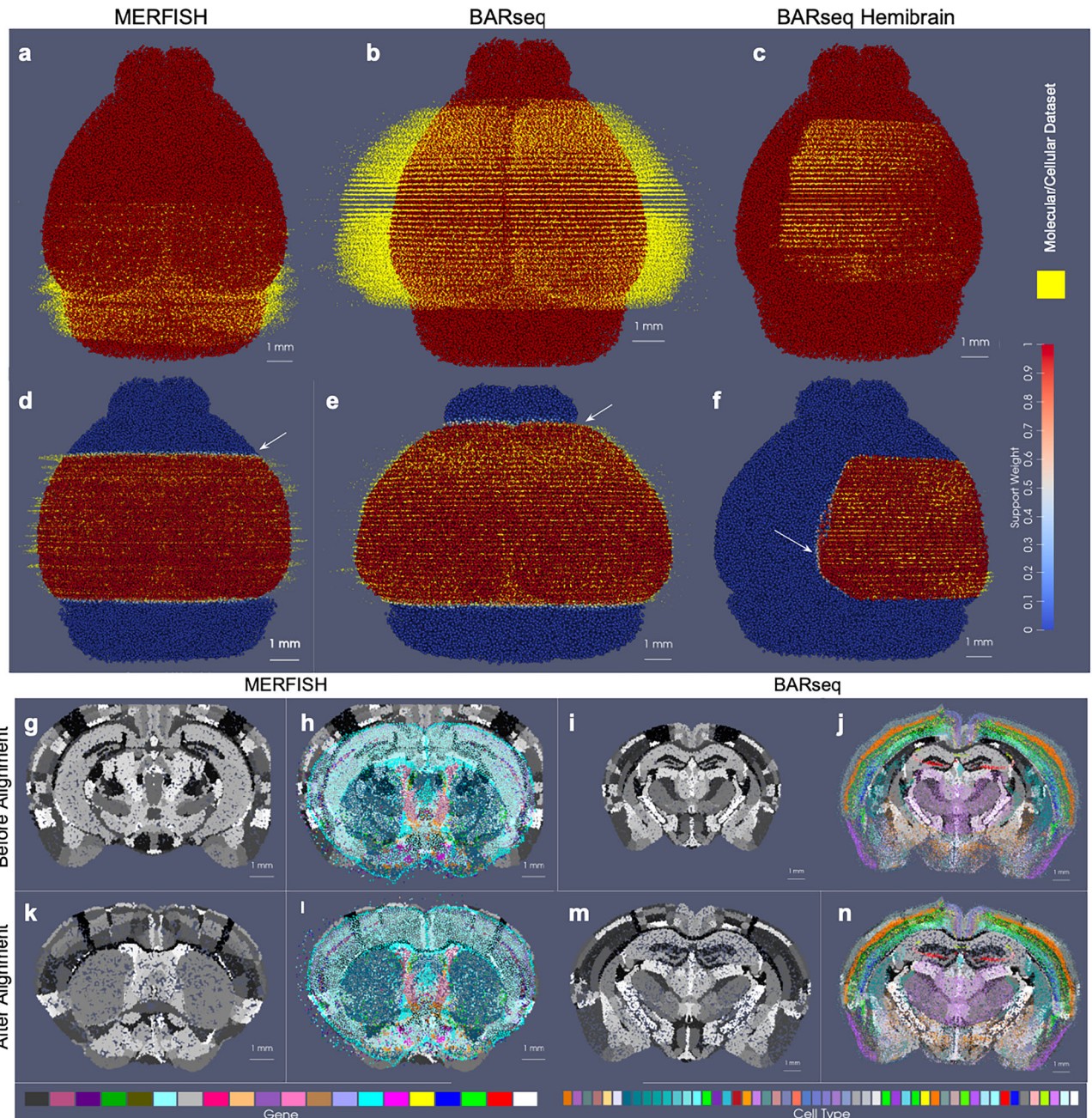

**Fig. 3 | Alignment of 3D atlas to molecular/cellular targets with automatic estimation of target tissue domain. a–f** Show estimated support weights in transforming whole brain CCFv3 to MERFISH and BARseq data. **a–c** Show initial alignment of CCFv3 to parallel sections of **a** MERFISH or **b** BARseq whole brain or **c** hemi-brain data, with initial support weights in atlas set to 1 (red). **d–f** Show transformed CCFv3 optimally aligned to molecular/cellular sections, with estimated support weights per particle. White arrows indicate smooth transition zone from within support (weight of 1) to outside support (weight of 0). Both CCFv3 and molecular/cellular targets depicted as 200 um approximations, with total particles numbering 68k (CCFv3), 95k (MERFISH), 163k (BARseq whole brain), 62k (BARseq hemi-brain). **g–j** Show initial intersecting slice through CCFv3 aligning to single **g, h** MERFISH or **i, j** BARseq section of data. **k–n** Show intersecting slice after alignment.

minimal or absent deformation (e.g. with $|D\varphi| \approx 1$, as shown in Fig. 4c, d) within the regions of the atlas outside the support (e.g. left most side of the brain and rostral and caudal poles). Alignment between CCFv3 and each target stack of sections was observed both globally (Fig. 4c, d) and on a section-wise basis with pockets of expansion interior to the tissue (Fig. 4g–l) yielding alignment of boundaries around the striatum and between cortical layers according to cell type as well as the corpus callosum to regions of low cell count (few colored particles, Fig. 4g–l). Importantly, alignment extended across the midline in settings of cell detections in the left hemisphere, as seen particularly in the placement of layer 2/3 cells within the CCFv3 layer 2/3 designation both on the right and left hemispheres (Fig. 4h–l).

We quantified alignment of designated cell types within corresponding cortical layers of the CCFv3 atlas, yielding comparable percentages of accuracy to what was achieved through manual alignment of each slice individually in seven separate hemi-brains to the CCFv3 atlas (Fig. 4o and Supplementary Table 3). Notably, errors in concordance arise both due to errors in cell typing and in alignment. Additionally, layer 4 typed cells were specifically categorized as layer 4–5 cells and thereby, lower fractions of such cells appeared within the specifically designated layer 4 areas of the CCFv3 (Fig. 4m, n). In contrast, alignment with xIV-LDDMM achieved greater concordance between CCFv3 regions and cell types in the deeper layers (5 and 6) than manual alignment (Fig. 4o). Furthermore, while fractions of correctly aligned cells within layer 4

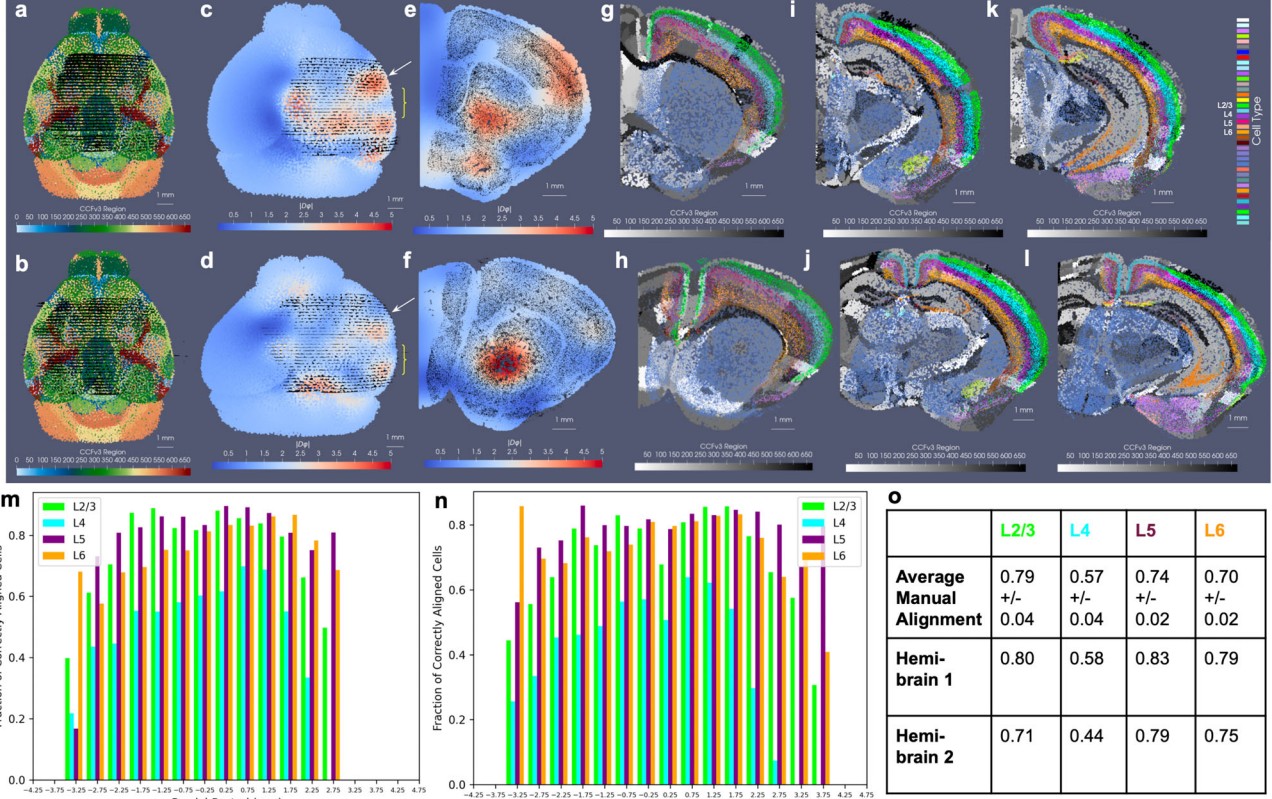

**Fig. 4 | Results for mapping CCFv3 to sets of hemi-brain BARseq sections, aggregated into 200 µm representations for two separate mice. a, b** Show initial alignment of CCFv3 to BARseq sections (black dots) for **a** mouse 1 and **b** mouse 2. **c, d** Show physically transformed CCFv3 to coordinates of target BARseq sections, with determinant of the Jacobian showing areas of expansion (red) and contraction (blue). **e, f** Show intersecting sections of deformed CCFv3 (white arrow) overlaid with corresponding BARseq section (black dots) from each hemi-brain. **g–l** Show intersecting sections of deformed CCFv3 with ontology regions in grayscale and cell type delineations

for BARseq sections overlaid in color, with **g, h** taken at white arrow and **i-l** more posteriorly (yellow bracket in **c, d**). **m, n** Show fraction of correctly aligned cells within each hemi-brain per each of four layer delineations (L2/3, L4, L5, L6) to corresponding CCFv3 region within 500-µm invervals along caudal-rostral axis. Total number of cells within each of four layer delineations given in Supplementary Data 1. **o** Shows average fraction for 7 hemi-brains of correctly aligned cells with manual alignment (data per hemi-brain provided in Supplementary Table 3) versus fractions achieved with xIV-LDDMM for the two hemi-brains shown.

|  | L2/3 | L4 | L5 | L6 |
|---|---|---|---|---|
| Average Manual Alignment | 0.79 +/- 0.04 | 0.57 +/- 0.04 | 0.74 +/- 0.02 | 0.70 +/- 0.02 |
| Hemi-brain 1 | 0.80 | 0.58 | 0.83 | 0.79 |
| Hemi-brain 2 | 0.71 | 0.44 | 0.79 | 0.75 |

decreased with proximity to the caudal and rostral poles of the data, we observed relative stability in fractions of correctly aligned cells across the other layers along the caudal-rostral axis (Fig. 4m, n).

Finally, as indicated in Fig. 1, the censoring technology described here is applicable not only to the alignment of datasets stemming from different modalities, as often result in different scopes of capture, but also to the alignment of datasets from a single modality, capturing different scopes of tissue in different specimens. As evidence of this application, we mapped the two sets of control mouse hemi-brain sections demonstrated in Fig. 4 to one another (Fig. 5), with the template (atlas) set containing 32 individual sections and the target set containing 31 (without the seventh most caudal section). Though the sections were taken at approximately every 200-µm along the rostral-caudal axis, resulting in a similar scope of tissue capture between the two brains, the atlas set extended more rostrally and medially than the target set, as evidenced by their corresponding alignment when mapped to the CCFv3 (Fig. 5a). We initially aligned the hemi-brains slicewise (Fig. 5b), resulting in mismatched anatomical regions, including areas of the hippocampus such as CA3 and subiculum (Fig. 5d, e). With the censoring function, minimization of the varifold normed difference of the hemi-brains was achieved over the domain of the target hemi-brain, yielding an optimized diffeomorphic mapping of atlas to target hemi-brain in which concordance between these internal hippocampal regions and between the outer layers was better achieved (Fig. 5d–f).

We measured this concordance before and after alignment at a resolution of 100 µm by comparing the predominant atlas and target cell types within cubes of 100 µm³ across the domain of every fifth target section

(Fig. 5g, h). Near 80% concordance was achieved specifically within the superficial layers of target cortex (2/3, 4/5) after diffeomorphic alignment compared with 20–30% as initially aligned slicewise. Additionally, we see slightly greater accuracy achieved within the rostral half of the slices (e.g. 15–25) compared with the caudal and with little loss in accuracy (≈10%) towards the rostral and caudal poles of the dataset (e.g. slice 5 and 30).

## Scale-space resampling

The spatial resolution of imaging in MERFISH and BARseq implies detection at the nano-scales of individual mRNA molecules each corresponding to a specific gene. The sheer complexity of what results from this sub micron imaging thus necessitates approximation based on optimal regridding for computational manipulation and downstream analysis. At any fixed complexity the particle measures are an approximation of the near infinity of potentially measured transcripts (or cells) in tissue. Unlike regular lattices used for tissue-scale imaging, the normed space our image-varifold brain objects are placed into enables the opportunity to optimally position the subspace of discrete particles being used to approximate the target at any specified complexity. We optimize resampling of the brain measure based on introducing a scale space of particle approximations, $\mu_\sigma$ over a series of scales, $\sigma$, which ascend in dimension as scale is refined converging to the target brain, $\mu$:

$$\mu_{\sigma_1}, \mu_{\sigma_2}, \ldots, \mu \text{ with } \sigma_1 > \sigma_2 \ldots.$$

Particle approximation rather than being based on regridding is designed based on an optimality process enabling construction of representations at

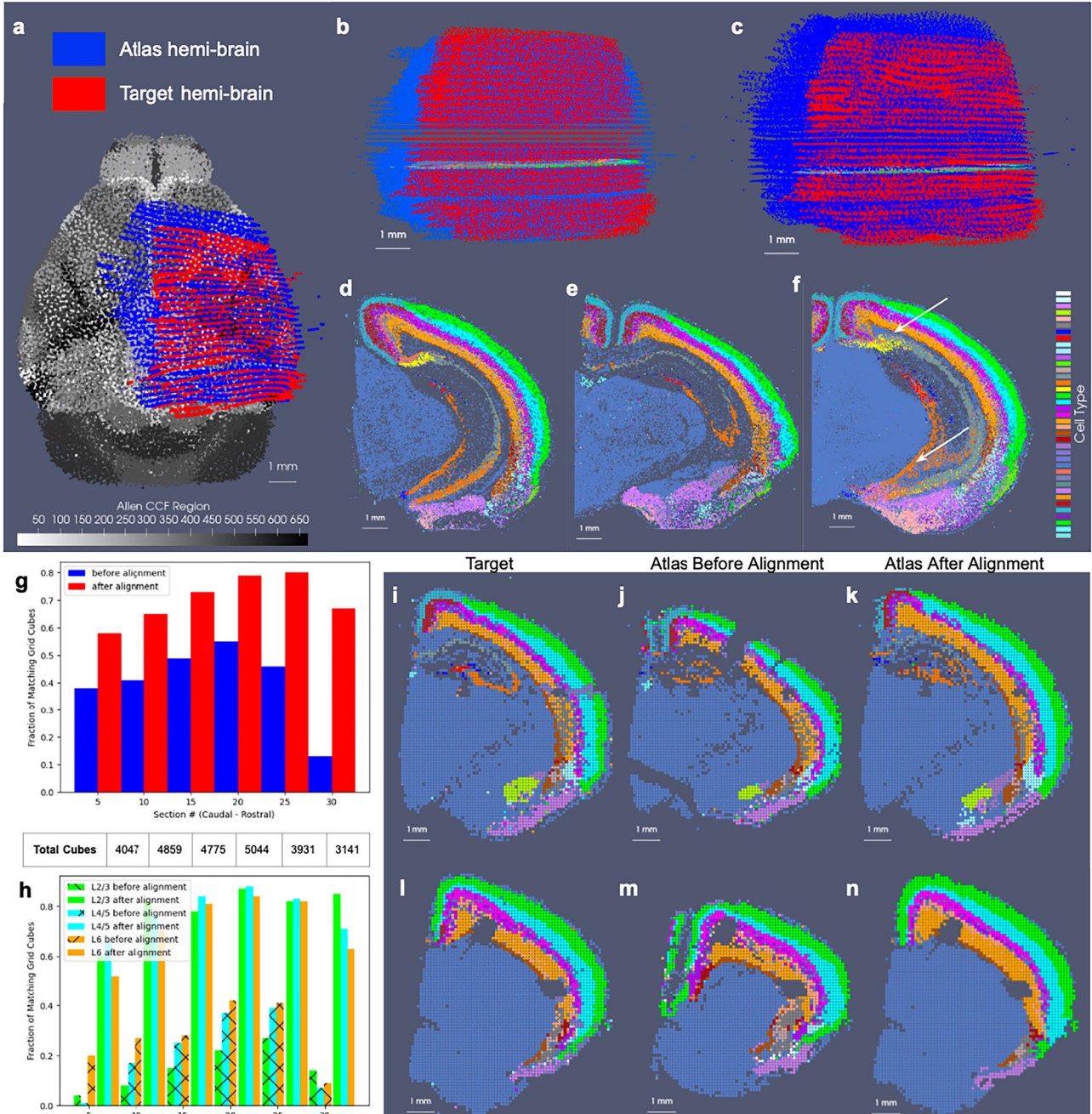

**Fig. 5 | Censoring technology for use in mapping sets of hemi-brain sections from two separate mice to one another. a** Shows each set of hemi-brain sections mapped to CCFv3 for comparison of scope of tissue capture. **b** Shows initial alignment of hemi-brain tissue sections. **c** Shows alignment of tissue sections following estimation of diffeomorphism with censoring to take atlas hemi-brain to target hemi-brain. **d** Shows single target slice with cell type denoted by color. **e, f** Show intersecting atlas slice **e** before and **f** after alignment to target hemi-brain sections, with white arrows indicating regions of hippocampal (Subiculum in yellow, CA3 in orange) alignment post deformation. **g, h** Show the fraction of 100 μm³ cubes across the domain of the selected target sections with matching predominant cell type between atlas and target hemi-brain **g** globally across the whole section and **h** layer-specific. Total number of cubes per layer given in Supplementary Table 4. Predominant cell type per cube is shown for **i, l** target sections, **j, m** intersecting atlas section as initially aligned and **k, n** after diffeomorphic alignment.

different scales (or resolutions). The natural optimality is to define the distance using the varifold norm and constructs the closest approximation in norm to $\mu$. Each particle approximation

$$\mu_{\sigma_j} = \sum_{i \in I_{\sigma_j}} \delta_{x_i} \otimes (w_i p_i), \quad \{x_i, i \in I_\sigma\}, \text{ for } j = 1, 2, \ldots$$

is optimized over particle positions $x_i$, $i \in I_\sigma$ and weighted conditional probability distributions $w_i p_i$ minimizing the normed-distance approximating the true target:

$$\min_{x_i, w_i, p_i, i \in I_\sigma} \| \mu_\sigma - \mu \|_M^2. \tag{5}$$

Complexity is defined in terms of the number of particles representing a given brain object ( $\propto |I_\sigma|$ ). The approximations thus increase in size with descending scale, $|I_{\sigma_1}| < |I_{\sigma_2}| \ldots$; in 2D thick section this scales as $O(1/\sigma^2)$, while for 3D, the complexity scales as $O(1/\sigma^3)$.

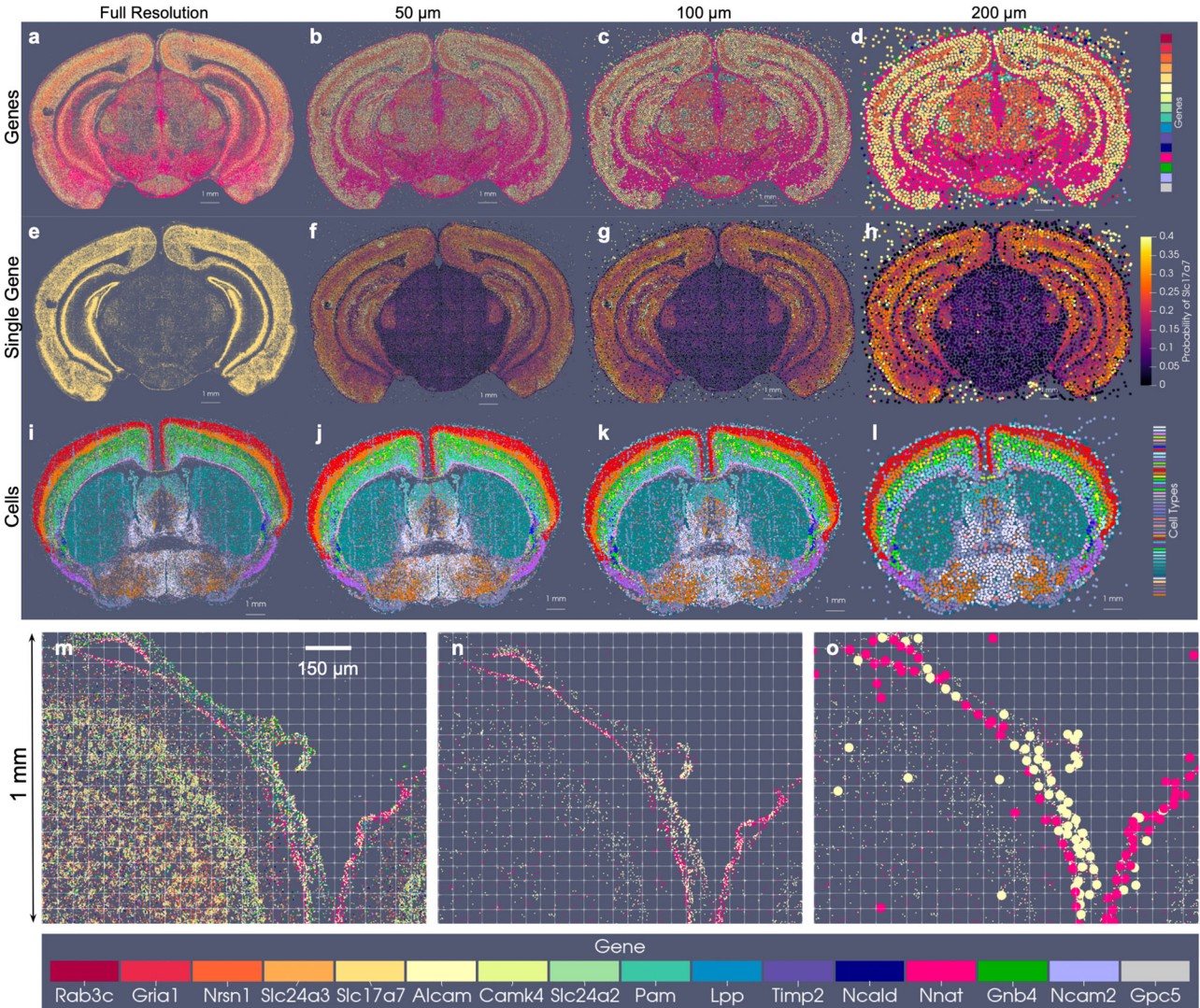

**Fig. 6 | Optimized scale-space resampling based on varifold norm for genes and cell type feature spaces of BARseq. a** Shows a single section with 16 selected gene measurements totaling 48 M transcripts. **b–d** Show the gene with highest probability per particle at 50 μm, 100 μm, 200 μm approximations with 82k, 21k and 6k particles, respectively. **e–h** Show probabilities at each scale of the single *Slc17a7* gene. **i–l** Show a single slice of BARseq cell-segmented data with 104K cells at highest resolution and at 50 μ, 100 μ, and 200 μ approximations with 54k, 18k and 5k cells, respectively.

Each cell is colored with the highest probability cell type. **m** Shows 1 mm × 1.25 mm window of full resolution gene detections for 16 genes (in **a**), and **n** shows detections only of Alcam and Nnat. **o** Shows detections in **n** overlaid with particles whose estimated feature distributions carry the highest probability of expression for Alcam (yellow) and Nnat (pink) in optimally estimated positions along the curvilinear boundary of tissue.

The generalizability of the image varifold representation, as described, extends to this setting where we can estimate cascades of optimized scale-space representations across scales ($\sigma_3 = 50$, $\sigma_2 = 100$, $\sigma_1 = 200$ μm) and for different feature spaces (e.g. gene and cell-types) (Fig. 6a–l). Notably, dominant genes (Fig. 6b–d) and dominant cell types (Fig. 6j–l) estimated per particle reflect the spatial patterns of gene expression and cell distribution in the full resolution dataset (Fig. 6a, i). Furthermore, particles in approximations carry not just a single most likely feature, but distributions over feature types reflective of larger and larger neighborhoods (Fig. 6f–h). This enables the interrogation of the spatial probability distribution of discrete features, such as single genes. In a single BARseq coronal section, for instance, areas of high probability of expression of specific genes (e.g. *Slc17a7*) accurately reflect high density areas of expression of said gene at the highest resolution (Fig. 6e), consistent with the accuracy of our scale-space resampling, with those areas of highest expression being those of the hippocampus and both the outer and innermost layers of cortex compared with the middle layers (Fig. 6e–h).

Importantly, particle measures in the reduced set are initialized with uniform distributions over features and with total mass given by the $\sigma$-neighborhood of highest scale (full resolution) measures. Without restriction to a regularized grid, spatial positions of particles move to optimally align the feature distributions based on the varifold norm, thereby reflecting what is more likely a curvilinear geometry of the tissue elements (Fig. 6). Particles approximated along the central sulcus, for instance, (Fig. 6c) begin with uniform distributions and are further differentiated according to feature distribution with particular genes predominating (probability of Nnat in the pink dots is 0.37 +/− 0.15 and probability of Alcam in yellow dots is 0.33 +/− 0.15), as observed at high resolution (Fig. 6a, b). As we are approximating at a scale of 50 um, particles move on average 9.7 um +/− 6.5 um, to adjust to the specific geometry/curvature of the tissue and distribution of the underlying genes.

We have compared the optimization based scale-space resampling to alternative schemes of regridding and clustering (Supplementary Table 1). For *K*-means clustering, we follow Lloyd's algorithm[29], with number of

clusters, $K = 20000$, and positions initialized to those of a randomly selected subset of $K$ particles from our starting set of $\approx 5$ million. The reduced set of particle measures carry spatial positions as the estimated centroids and weighted conditional probability distributions comprised of the sum of masses per feature of the high resolution particles assigned to the corresponding cluster. For grid resampling, we follow the scheme in ref. 12 for crossing scales, where particle measures are redistributed in space according to a given spatial kernel.

Methods such as $K$-means place the particles on a regularized grid so as to minimize the physical distance only to the high-resolution particles, rather than account for proximity in both physical and feature space. This strategy thus generates particle measures that carry a much greater distance with respect to the varifold norm to the high resolution target than our defined scheme here (Supplementary Table 1). This underscores the relevance of our varifold norm approximation method for achieving data reduction in the context of mapping tissue-scale atlases to molecular and cellular-scale targets with xIV-LDDMM.

## Feature selection via mutual information

The complexity of transcriptomic measurements not only requires reduction in physical space, as captured by scale-space resampling, but often also in feature space via a feature selection mechanism. A common selection mechanism is to consider those genes that are most "spatially variable"[30] or "differentially expressed"[31] under the assumption that expression pattern thereby varies per biologically different regions of tissue. This is particularly relevant, here, in the context of mapping spatial transcriptomics datasets to atlases where we aim to estimate distributions over genes for each region in our atlas that we assume is homogeneous within the region. Various methods have been described for identifying which genes in a spatial transcriptomics dataset are spatially varying including hierarchical classification[32] and modeling based on covariance statistics and Gaussian process models[30,33]. Since particle methods, as used here, imply sparsity in number, methods based on spatial Poisson and point-process models, as introduced in ref. 13, underscore the fundamental roles of Kullback Liebler (KL) divergence between empirical distributions derived from the field to score hypothesis reliability. KL divergence has also been proposed in ref. 31 for selecting genes with differential expression across cells distributed in space.

Here, we introduce information theoretic methods based on mutual information scoring which assess the differential expression of genes in space in a cell-independent manner, scoring highly genes that are most spatially varying (see Section Mutual Information Scoring for Feature Selection). For this we define the family of features $G$ capturing statistics based on mRNA expression (or cell-type) and a random variable $X$ that simulates the following process. Select at random a sub-window in the image, split it into two halves (left/right or up/down), pick a random sub-region in that window and record in which half the subregion fell. A feature in $G$ is spatially informative if its value within the selected region has predictive power on the half window that contains it. This predictive power is measured by mutual information. Our mutual information selection procedure is greedy in ordering the genes independently based on decreasing mutual information score, indicative of lesser discrimination of the up-down/left-right boundaries:

$$I(X; G(1)) > I(X; G(2)) > I(X; G(3)) \ldots \ldots$$

We finally select the first $n$ genes (features) as our subset of most informative genes.

For mapping the Allen CCFv3[4] to serial sections of MERFISH data[2,3], we ordered the 500 genes measured by computed mutual information score and selected the subset of 20 with the highest scores to comprise a reduced feature set. Scores ranged from $\approx 470,000$ to $200,000$ roughly following a single mode distribution. In line with expectations, 75% of the genes comprising those with scores in the bottom 25% of the total 500 genes were decoy genes (e.g. 'BLANK') without biological meaning but used as controls

for assuring the quality of the dataset. Non-decoy genes exhibiting scores in the bottom 2% of the entire gene set were *Chodl*, *Brs3*, and *Hpse2*, whose expression patterns carried high local variance, but with stationarity observed over entire tissue sections, from region to region (Fig. 7d–f, j–l). In contrast, genes with the highest mutual information scores included *Gfap*, *Trp53i11*, and *Wipf3*, where certain areas of tissue exhibited high local density in contrast to others (Fig. 7a–c, g–i). For such genes, the observed delineations with respect to expression level equated to many complementary delineations given in the CCFv3 ontology, such as that of the corpus callosum exhibiting high levels of *Gfap* expression (Fig. 7p) or the lateral septal nucleus exhibiting high levels of *Wipf3*, (Fig. 7r), thereby underscoring the relevance in these genes' expression patterns as target feature measures for aligning to the regions in the CCFv3 and the fundamental assumption we make of a stationary probability law within each atlas region.

Indeed, we quantified the extent to which the stationarity assumption holds through assessment of intra-regional versus inter- regional variance in the context of genes with high mutual information scores versus low. We built empirical distributions onto 50-μm resolution CCFv3 slices of particles following deformation (Fig. 7m–x). Each empirical distribution pertained to a set of 20 (non-decoy) measured genes taken to be either those with the highest mutual information scores or the lowest. Intra-regional variance was computed for selected genes based on the normalized empirical (probability) distributions per CCFv3 particle (Fig. 7o, r, u, x) with larger variance seen amongst lower mutual information gene probabilities (Fig. 7o, u) than those of higher mutual information (Fig. 7r, x). Consequently, pairwise two-sided Mann–Whitney U tests between each region's distribution of particle gene probabilities revealed many more pairs of regions significantly differentiated based on high mutual information gene probabilities than low, where significance was computed at a level of 0.05 corrected for multiple comparisons with a bonferroni correction (Fig. 7y). In the case of high mutual information genes, 43–68% of region pairs could be significantly distinguished, supporting high inter-regional variance compared to intra-regional variance, supporting our assumption of a stationary probability law. In contrast, low mutual information genes yielding higher intra-regional variance in probability distribution resulted in less than 15% of region pairs significantly distinguished. Hence, this suggests effective selection of features (e.g. cell types or measured genes) particularly in the context of anticipated atlas parcellations governs the extent to which the assumption of stationarity holds, but with methods of selection based on spatial variability, as described here, leading to features typically well aligned with the assumption.

## 3D mapping of BARseq to Allen CCFv3 atlas based on cell types

We evaluated the efficacy of our joint optimization scheme through association of delineated cell types in BARseq whole brain sections to respective cortical layers delineated in the CCFv3. The cell types were clustered with an algorithm similar to Louvain-clustering to achieve a resolution analogous to the "subclass" defined in RNA-seq studies[32]. These included both inhibitory and excitatory cell types specific to layers 2, 3, 4, 5, and 6 (8). Global alignment of the CCFv3 to BARseq stack was observed along the rostral-caudal axis and medial-lateral boundaries (Fig. 8b) as a result of estimating both similitude transformations and diffeomorphism to correct differences in scale and tissue geometry as seen in Fig. 8a. Within the tissue plane of individual sections (Fig. 8c–f), we observed alignment between regions predominated by single BARseq cell types and corresponding regional CCFv3 delineations both ventrally and dorsally. For instance, subtypes of gabaergic neurons predominant in areas of the superior colliculus (yellow arrow at dorsal orange, c), lateral lemniscus (yellow arrow at ventral green, c), and medial mammillary nucleus (yellow arrow at ventral peach, d) generated contours of these regions that aligned to the corresponding delineations of the CCFv3. Likewise, designations of pyramidal cells in areas of the hippocampus (e.g. CA1, yellow arrow at purple band, d) localize closer to the lateral/deeper aspects of the CCFv3 designated regions, as we expect anatomically.

**Fig. 7 | Selection of gene space in MERFISH measured ≈ 500 genes and illustration of within region and between region variance. a–l** Show relative expression of genes with high and low mutual information (MI) scores across two MER-FISH tissue sections at coronal levels **a–f** $Z = 385$ and **g–l** $Z = 485$ in CCFv3 coordinates, respectively. Genes with highest mutual information scores **a–c, g–i** are *Gfap*, *Trp53i11*, *Wipf3*) and lowest **d–f, j–l** are *Chodl*, *Brs3*, *Hpse2*. **m–x** Exhibit empirical distribution of genes in slices 1 and 2 deformed to corresponding CCFv3 slice. **m, p, s, v** Show probability of single **m, s** low MI gene or **p, v** high MI gene per atlas particle at 50-μm resolution. **n, q, t, w** Show average probability of single **n, t** low MI gene or **q, w** high MI gene. **o, r, u, x** Show variance within each region of probability of single **o, u** low MI gene or **r, x** high MI gene. **y** indicates the number of pairs of regions with significant differences in probabilities for three genes with high MI and low MI, with significance calculated at 0.05 but corrected for multiple comparisons with bonferroni correction as 0.00003 and 0.00001 based on the number of regions as 60 and 94 in slice 1 and 2 respectively. All computed *p*-values provided in Supplementary Data 3.

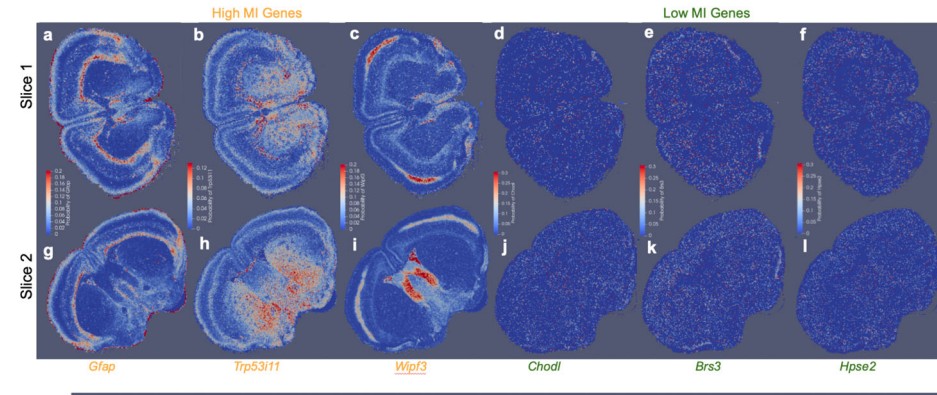

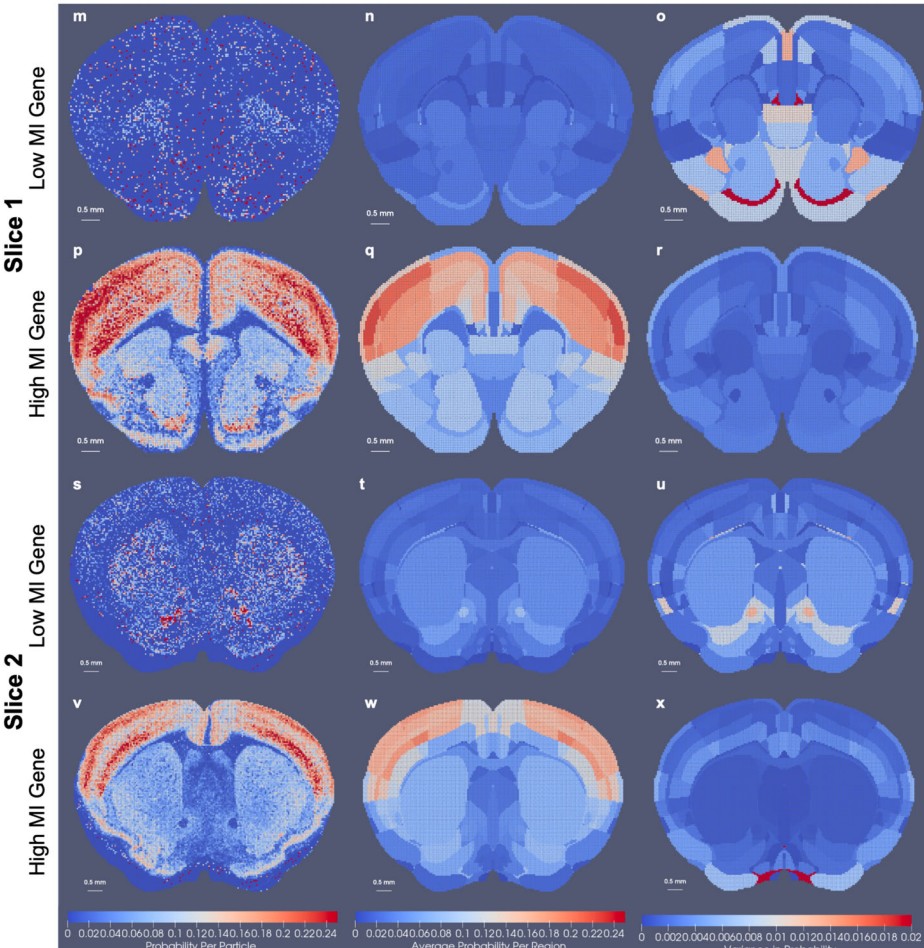

**y**

**Pairwise Mann-Whitney U Tests with P-value < Corrected 0.05**

|  | Slice 1 | | Slice 2 | |
|---|---|---|---|---|
|  | High MI | Low MI | High MI | Low MI |
| Gene 1 | 755 | 181 | 2212 | 338 |
| Gene 2 | 1210 | 217 | 2545 | 613 |
| Gene 3 | 997 | 245 | 2319 | 330 |
| Total Pairwise Comparisons | 1770 | | 4371 | |

We further quantified the accuracy of alignment across the cortical layers within the neighborhood of the primary somatosensory cortex and lateral visual area (dotted white circles) where each of the layers were present with a thickness of at least 150−200 m. As described in Section Mapping Whole Brain to Partial Volume Censored Targets with regard to matching cell types within hemi-brains to CCFv3 regions, we assessed the fraction of full resolution cells aligning to the correct corresponding CCFv3 layer according to cell type, as well as the distance of incorrectly aligned cells to the correct region. Cortical cells were selected bilaterally across ≈10 sections intersecting these regions. Percentages of correctly aligned cells ranged from ≈70 to 80% across the four layer designations with misaligned cells falling on average ≈50 μm from the correct layer (Fig. 8g, h) and thereby reflecting misalignment to adjacent layers only, with layers ranging in thickness from ≈150 μm to 300 μm (Fig. 8e, f). We compared these percentages to those

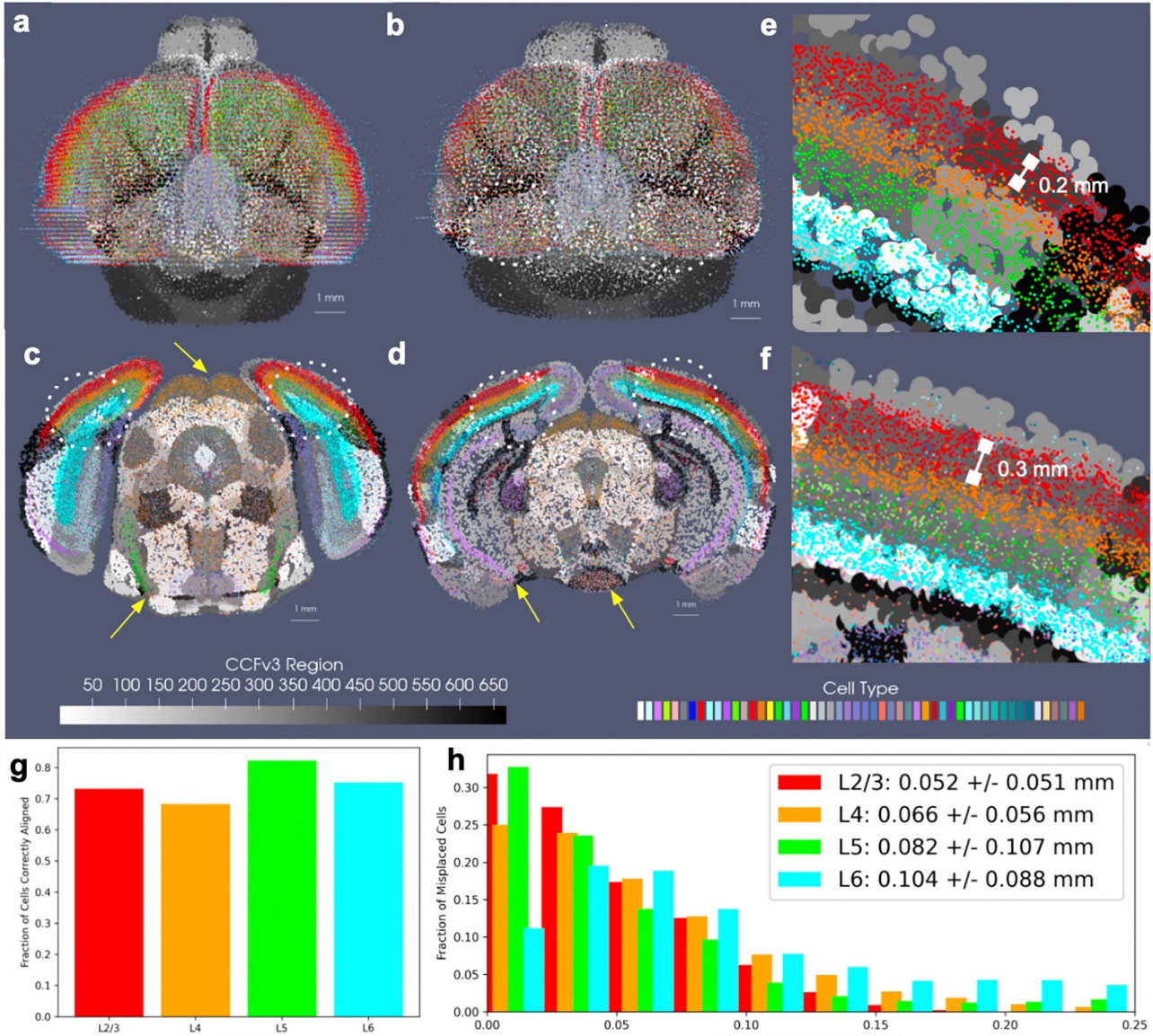

**Fig. 8 | Accuracy of mapping between CCFv3 and whole-brain BARseq sections** ($n = 40$), **carrying cells of ≈50 different subtypes, with each dataset approximated at 200 μm. a** Shows initial alignment of CCFv3 (grayscale) to whole-brain BARseq sections (color). **b** shows alignment in BARseq coordinates following diffeomorphic transformation of CCFv3. **c, d** show intersecting planes of transformed CCFv3 with two BARseq full resolution cellular sections ($n = 80$–$100$k cells per section). Yellow arrows point to areas of correct alignment between designated BARseq cell types and regional CCFv3 delineations. **e, f** Show zoomed in region of of lateral visual area and primary somatosensory cortex (dotted circles) in tissue sections in (**c, d**) with individual cells overlaying corresponding layers in CCFv3 grayscale delineations. Areas of the dentate gyrus and hippocampus show some misalignment in dorsal-ventral placement. **g** Shows percentages of correctly aligned cells across 10 sections bilaterally in the area of the primary somatosensory cortex and lateral visual area to corresponding CCFv3 cortical layer designation. Total cells mapped per layer are 23660 (L2/3), 19796 (L4), 11194 (L5), 22928 (L6). **h** Shows the distance of incorrectly aligned cells to the correct corresponding layer designation, with total number of incorrectly aligned cells per layer as 6345 (L2/3), 6279 (L4), 1975 (L5), 5681 (L6). All computed distances provided in Supplementary Data 2.

achieved on average with manual slicewise alignment of 7 hemi-brains to the CCFv3 (Supplementary Table 3). Across 7 hemi-brain samples, percentages of correctly aligned cells within layers 4, 5, and 6 were lower on average with manual alignment than with xIV-LDDMM ($0.79 \pm 0.04$, $0.57 \pm 0.04$, $0.74 \pm 0.02$, and $0.70 \pm 0.02$ versus 0.73, 0.68, 0.82, 0.75).

Notably, the percentages of correctly aligned cells within especially the deeper layers (5–6) were maintained throughout the entirety of the brain, across all 40 2D sections, with average fraction of correctly aligned cells at 0.82 and 0.76 (Supplementary Fig. 2). Similar to within the hemi-brain alignments, layer 4, in contrast, exhibited lower accuracy particularly towards the rostral and caudal poles of the dataset, likely as a result of cell typing and alignment error. As indicated in Section Mapping Whole Brain to Partial Volume Censored Targets, not only was there variation in cell

typing, naming cells physically within certain layers with type corresponding to a different layer, but the layer 4 cell type was specifically equivocally designated as a layer 4–5 type compared with the singly layer 4 CCFv3 region.

### 3D mapping of MERFISH to Allen CCFv3 based on genes

As illustrated in the setting of cell-typed sections (Section 3D Mapping of BARseq to Allen CCFv3 Atlas Based on Cell Types), there exist certain 1:1 relations between organization of cell types and demarcated tissue regions, as defined in the CCFv3 ontology (e.g. in cortical layers), that serve to anchor alignment between these scales of data and that can be harnessed directly to assess accuracy of this alignment. At the sub cellular scale, however, we observe and expect inherently more variation over space in terms of gene

expression with cell typing typically used to capture stable co-expression relationships among genes. As such, gene expression over physical space organizes into different sizes and shapes of homogeneous regions where some boundaries are preserved across tissue, cellular, and sub cellular scales (e.g. striatal boundaries) and others differ (e.g. medial to lateral and dorsal to ventral within the cortical layers). This inherently increases the difficulty in achieving alignment between these subcellular datasets and tissue-scale atlases that exist at even further discordant scales.

To demonstrate the amenability of our mapping strategy to bridging the tissue to the subcellular scales, we mapped the CCFv3 to a set of 60 MERFISH sections exhibiting expression of 20 of the 500 measured genes we selected as spatially variable (see Section Feature Selection via Mutual Information) according to mutual information score (Fig. 9). Alignment along the rostral-caudal axis, as facilitated by our incorporation of censoring to focus accuracy of matching within the MERFISH tissue support (see Section Mapping Whole Brain to Partial Volume Censored Targets) was observed globally (Fig. 9d) and on a section-by-section basis with areas of the striatum and hippocampus appearing in similar proportions and shapes in both the CCFv3 and gene space of MERFISH (Fig. 9g–z). Medial-lateral and inferior-superior alignment was also observed in the matching of cortical versus striatal boundaries that appeared in both CCFv3 ontological delineations and the gene space of MERFISH (Fig. 9b–f). Hence, our mapping strategy is amenable to these different granulations of parcellation in succeeding to situate both interior (e.g. striatal) and exterior (foreground/background) geometric delineations to one another across tissue and cellular architecture (Fig. 8) as well as ultimately tissue to gene expression architecture (Fig. 9).

### 3D mapping of CycleHCR whole mouse embryo to EMAP atlas

In both MERFISH and BARseq captures, volumes were interrogated on a slice-by-slice basis with tissue slices spaced 100 μm or 200 μm apart, respectively. Increasingly, newer technologies are emerging to accommodate single three-dimensional captures either directly or as part of a post-processing assembly built into the technology, itself[34]. Consequently, there is a need for mapping platforms, as that presented here, to accommodate datasets captured as single blocks, multiple tissue blocks, or multiple sections in an effort to integrate data across technologies and within the framework of common coordinate systems and standardized ontological atlases.

Additionally, as a result of these 3D captures, tissue types and shapes of volumes measured have continued to diversify with the capture of in tact specimens, not easily sectioned. One example is that of the whole mouse embryo at stage E6.5-7.0 measured with the novel deep-tissue spatial transcriptomics method cycleHCR[17]. To demonstrate the generalizability of our mapping technologies to one such deep-tissue capture and to specimens beyond those of the brain, we mapped time points of the corresponding EMAP atlas[16], developed from Kaufman ontological annotations[19] to this whole mouse embryo.

Notably, in the context of developing anatomy in contrast to adult anatomy, atlases are often comprised of select time points spanning the evolution of the organism, with individuality in the exact time course observed per specimen. Consequently, this creates even further challenges to alignment of atlases to targets where atlases often represent averaging over both space and time, which is rapidly evolving. Here, the two closest time points within the atlas series to stage E6.5–7.0 of the cycleHCR embryo are the ts09 time point at E6.0–6.5 and the ts10 time point at E7.0 (Fig. 10b and k). We mapped each of these atlases (initially designated at a resolution of $2 \times 2 \times 2$ μm) as $8 \times 8 \times 8$ μm particle representations to a cell-based particle representation of the cycleHCR embryo, containing 11,029 cells (Fig. 10a). We assessed the alignment qualitatively through observation of concordant areas overlapping, such as the areas of the primitive streak, mesoderm, and parietal endoderm in each atlas with cells typed similarly as constituting those regions (black and white arrows in Fig. 10f–j, h–s). Quantitatively, we assessed alignment by computing the minimum distance of each cell of the primitive streak and parietal endoderm to its nearest atlas neighbor within

the same corresponding region, with an observed reduction in the average distance 4.5 fold before versus after deformation (Fig. 10c).

Interestingly, the geometry exhibited in the cycleHCR embryo appeared to be a hybrid of those prescribed by the ts09 and ts10 atlases. To compare the alignment between the the cycleHCR embryo and each corresponding atlas, we again considered the regions where cell type corresponded directly to region (e.g. primitive streak and parietal endoderm). Here, the minimum distances achieved following alignment to the ts09 atlas within the primitive streak and parietal endoderm were 0.016 ± 0.019 and 0.016 ± 0.016 mm, respectively. In contrast, the minimum distances were greater in the case of the ts10 atlas, 0.031 ± 0.024 and 0.057 ± 0.07 mm (Supplementary Fig. 3), respectively, suggesting poorer similarity in geometry resulting in larger distance discrepancies between the regions. Nevertheless, cross-sectional views of the cycleHCR embryo and each of the atlases revealed greater visual concordance between the aligning geometry of the primitive streak (blue) and mesoderm (yellow) between the cycleHCR embryo and the ts10 atlas compared with the ts09 atlas (Fig. 10r versus Fig. 10j). In contrast, the ts10 atlas exhibited further differentiation of the extraembryonic tissues into the amniotic fold, not yet present within the cycleHCR embryo (Fig. 10q–s, dotted arrows). However, the cells typed as precursors of the amniotic fold including those of the extraembryonic mesoderm and ectoderm corresponded locationally to the area of the amniotic fold in the ts10 atlas, suggesting further concordance of alignment in the setting of anticipated differentiation of these cells. Finally, with both the ts09 atlas and the ts10 atlas, the measured cycleHCR embryo comparably represented only a subset of the entire volume, similar to the partial volumes captured within the whole brains measured with BARseq and MERFISH. In both cases, the majority of the polar trophectoderm, not imaged with the cycleHCR technology aligned to outside the support of the target, as expected.

### Computational complexity of 3D particle codes

We have examined the effects on runtime and estimated diffeomorphism in mapping a single consistent tissue-scale atlas to particle representations of varying size and feature spaces of the stack of 40 whole brain coronal BARseq sections. Particle representations at scales of 200, 100, and 50 μm corresponding to discrete particle numbers of $o(5000)$, $o(20,000)$, $o(50,000)$ per single section, respectively, were estimated with optimized scale-space resampling (see Section Scale-Space Resampling) for the BARseq sections over the entire feature space of 52 cell types (Supplementary Fig. 5D–F).

Supplementary Table 2 shows the complexity and corresponding memory requirements to store the particle representations. Runtime is computed as total time to run 150 iterations of the optimization scheme for estimating the geometric transformation and feature distributions. Memory requirements are tallied as the amount of storage per target dataset given as .pt file. Resolution, # particles, and # features are all given for the target image varifold. Runtime measures were computed on an NVIDIA RTX A5000 GPU. Feature distributions and diffeomorphisms, parameterized by a set of initial momenta numbering $o(60000)$ for the number of particles in the 200 μm CCFv3 representation, were estimated to transform the CCFv3 atlas to each of the target representations via minimization of (9), with $K_\sigma$ chosen as the sum of two gaussian kernels at 150 and 750 μm.

Both memory requirements and runtime increased, particularly in the setting of utilizing a 50 μm aggregated representation compared with one at 200 μm (Supplementary Table 2), thereby supporting the use for reductionist mechanisms to treat the high initial complexity of this data. Importantly, however, we observed no difference in estimated diffeomorphic transformation, as evidenced by the determinant of the Jacobian depicting the distribution and magnitude of expansion and contraction (Supplementary Fig. 5G–I). The equivalence in mapping results across the experiments run with different scales of target approximation was also reiterated in the observed global alignment and slice-wise alignment of BARseq particle measures (black) overlaying corresponding areas of CCFv3 delineated tissue (Supplementary Fig. 5A–C, J–L). Alignment of the cortical layers was visually estimated in each case to be within 25−50 μm, as

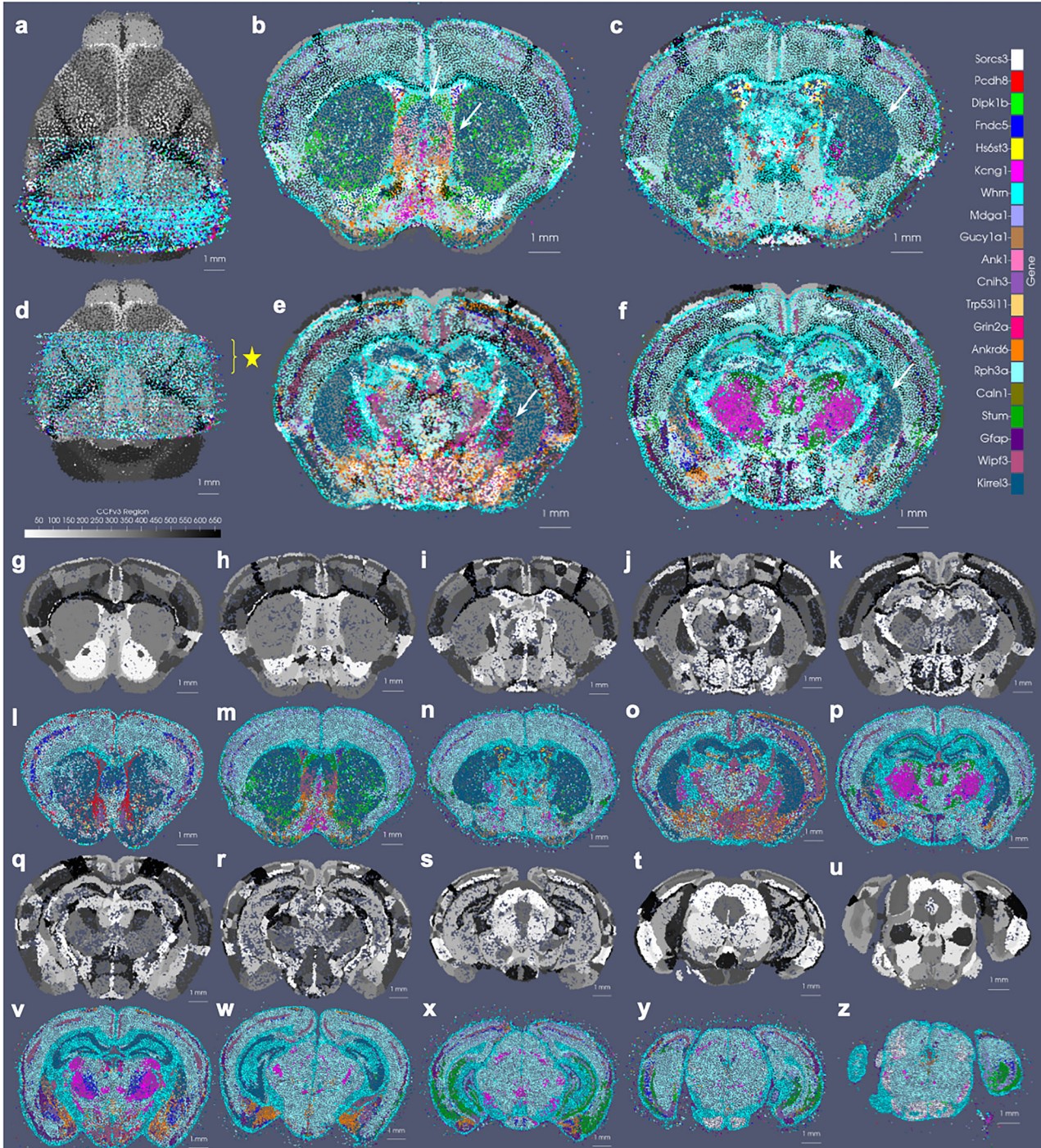

**Fig. 9 | CCFv3 Alignment to MERFISH reads of 20 selected spatially variable genes. a** Shows initial alignment of CCFv3 to stack of 60 sections, each sampled at 200-μm resolution. **d** Shows final alignment of CCFv3 to stack of MERFISH sections following rigid and diffeomorphic transformation. **b–f** Show in-plane alignment of intersecting CCFv3 to 4 selected MERFISH sections in the anterior half of the mouse brain (spaced 500 microns apart, yellow star in (**d**)). Alignment between foreground-background tissue boundaries is seen particularly along the ventral and lateral edges. White arrows show matching contours of outer and inner boundaries of the CCF striatum to areas predominantly expressing Kirrel3. White arrows also illustrate alignment within the area of the lateral septal nucleus (**b**). **g–z** Show sections **l–p**, **v–z** every 0.5 mm with **g–k**, **q–u** corresponding intersection through deformed atlas following alignment, as in (**d**).

evidenced by layer 1 of the CCFv3 falling outside the outermost cell type of layer 2/3, and layer 6 dorsal to the corpus callosum (Supplementary Fig. 5M–O). Notably, the equivalence in estimated mappings across these three scales of particle representations supports the integrity with which the scale-space resampling scheme (Section Scale-Space Resampling) holds to the high resolution data at each scale.

## Discussion

This paper describes a set of technologies that enable cross-modality mapping across tissue, cellular, and molecular scale data in 3D settings. We have specifically introduced modules for (i) matching partial to whole tissue volumes with censoring, (ii) scale-space optimization for multi-scale resampling, and (iii) mutual information-based feature selection. Importantly, while these

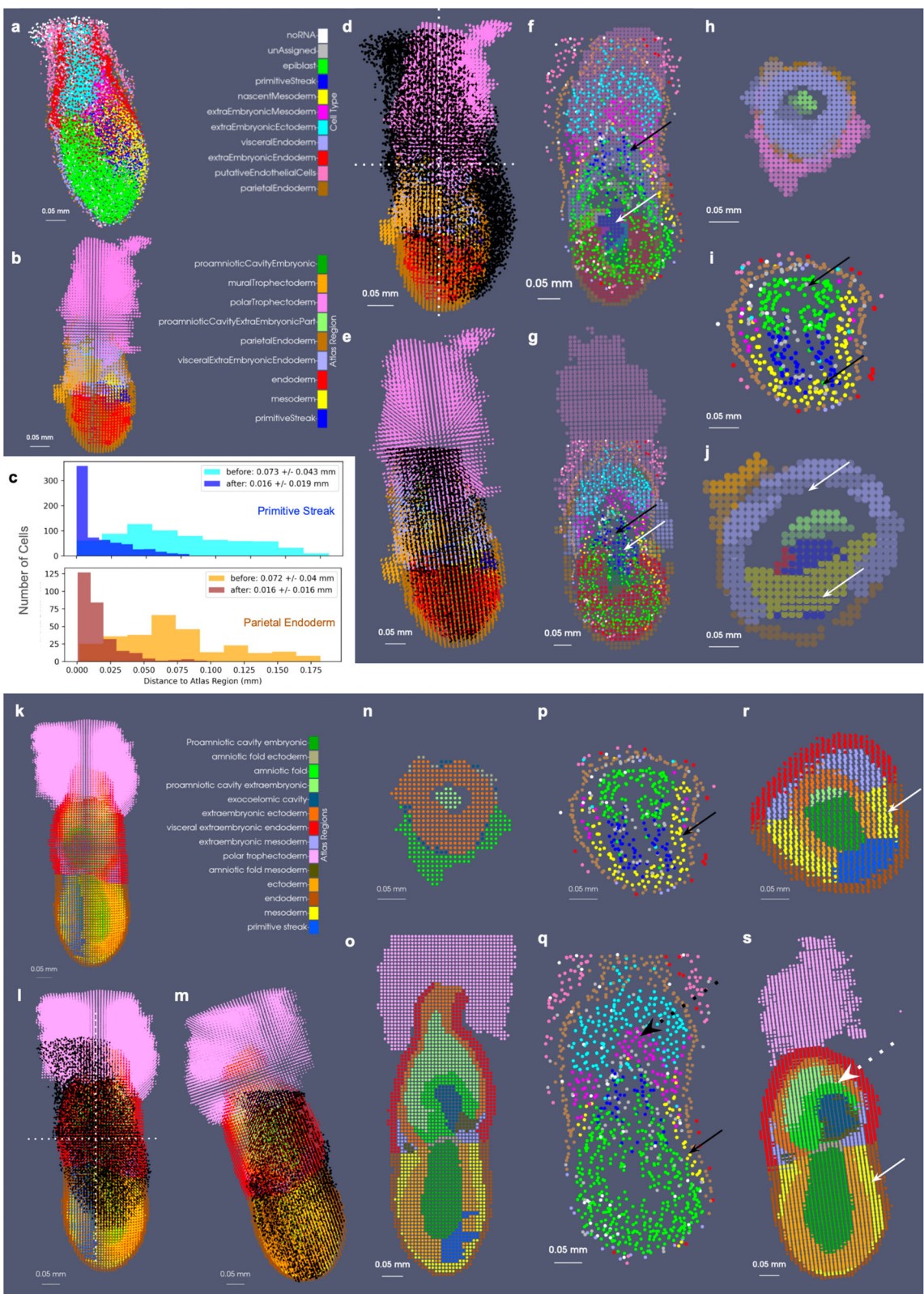

technologies can be employed independently for data reduction and analysis, they are rooted in the image-varifold (particle measure) representation to facilitate integration amongst them and with existing modules for cross-modality mapping[13,15]. Collectively, these technologies thus afford efficient representations of potentially peta-scale data and mapping with increasing refinement across millimeter to nanometer scales.

While we have focused, here, on the complexity challenges posed even in the setting of mouse anatomy, we would expect this complexity only to increase in the setting of larger primate and ultimately human anatomy, where we already see similarities and need for the censoring and reduction schemes we introduced here. In brain imaging problems, for instance, subvolumes and/or sparse data sets are almost always collected, with a focus

**Fig. 10 | Mapping of EMAP ts09 and ts10 mouse embryo atlases (8 × 8 × 8 µm resolution) to cycleHCR whole mouse embryo cells. a** Shows cell centers of measured embryo with prescribed cell type based on UMAP clustering[17]. **b, k** Show ts09 and ts10 atlases with 9 and 14 regions, respectively. **d, l** Show initial alignment of each atlas to target embryo (black dots) versus **e, m** Show alignment following estimation of diffeomorphism with xIV-LDDMM. **f, g** Show vertical cross-section through alignment of ts09 atlas to target before and after deformation, with black arrows indicating area of primitive streak in target embryo and white arrows indicating primitive streak in atlas. **h–j** Show horizontal cross-section of ts09 atlas **h, j** before and after deformation and of **i** target embryo with arrows indicating primitive streak and proamniotic/amniotic cavity. **c** Shows distance of each embryo cell with primitive streak and parietal endoderm type to corresponding type in ts09 atlas before and after alignment. Full set of computed distances provided in Supplementary Data 2. **n–p** Show horizontal cross-section of ts10 atlas **n, r** before and after deformation versus that of **p** target embryo with arrows indicating area of mesoderm. **o–s** Show vertical cross-section of ts10 atlas **o, s** before and after deformation versus **q** target embryo with solid arrows indicating mesoderm and dotted arrows highlighting area of developing amniotic fold.

on particular regions involved in particular diseases. For instance, in Alzheimer's disease, the medial temporal lobe[35] is often the region of study, whereas in Huntington's disease, the subcortical structures[36] are the focus. Spatial transcriptomics technologies have similarly limited capture to within subregions, often to optimize resolution within these regions at the expense of further breath across tissue. In mapping problems, while whole volume objects could be subsectioned prior to mapping according to known subvolume covered by a molecular or cellular dataset, this often is not known apriori and prevents easy integration of multiple molecular and cellular datasets into a single unified coordinate framework that may only partially overlap with regard to subvolumes measured.

In differential geometry, the emphasis is to represent the whole via a collection of local charts which form a complete covering; the atlas is thus a collection of local charts. This implies the diffeomorphic mapping problem is surjective, as we have examined in ref. 37. Here, we provided a general strategy for solving the "chart problem", which significantly expands the partial matching approaches in the setting of medical images modeled in ref. 38, by appending the inner-product with a censoring function ?? (Section Mapping Whole Brain to Partial Volume Censored Targets). The scheme for partial matching or censoring the full tissue-scale atlas to molecular and cellular datasets spanning only subvolumes uses a masking function that punctures the corresponding subvolume of the target within our deforming atlas. This effectively restricts the matching criteria to the subvolume within which molecular/cellular data has been detected to drive both geometric transformation and feature distributions fundamentally to reflect the biological boundaries in the data present and not those artificially induced through only partial captures of tissue volume.

The optimization based multi-scale resampling we propose benefits from the fact that we place the brains into a Hilbert space (e.g. with an inner product generating a distance metric), which enables optimized approximations at any scale, i.e. complexity. Furthermore, while the scheme parallels traditional clustering schemes (e.g. *K*-means), these clustering schemes, similar to many learning-based methods for alignment[9], are often limited to defining distance only with regard to "location" in either physical or feature space, but not both, as is innately captured by our image-varifold normed space. This coupled distance metric therefore offers a more consistent biological interpretation of distance as a measure of similarity in gene expression or cell types dispersed in physical space. Rasterization (as in ref. 20) and resampling onto regular lattices, for instance, as means of aggregation focus exclusively on proximity in physical space. Clustering based exclusively on feature type (e.g. gene) has also been described, with particular support of clustering schemes not assuming particular shapes or sizes to clusters (e.g. Louvain cluster)[39]. While some efforts to combine clustering along both axes have achieved this by applying tactics independently and sequentially, many have focused on jointly estimating a single lower-dimensional representation that aptly captures the structure of the data along both physical and feature axes[40–43]. Indeed, as with many generative schemes, such as SpatialPCA and BayesSpace, these joint approaches have been used not only to reduce data to set of lower dimensions, but to use such dimensions to generate higher (subspot) resolution of original datasets[41,42]. Here, we demonstrated that for a fixed particle grid, the distances of the measure approximation to the original varifold are significantly less than many of the other regridding methods including *K*-means and nearest-neighbor. Furthermore, we showed in our optimization scheme that particle measures not only change position to approximate the ideal, often curvilinear geometry of the tissue, but they also change

dramatically the functional feature to more appropriately match the transcriptome or cellular target.

Finally, we also have defined an information theoretic greedy procedure for choosing a subset of functional features from $\mathcal{F}$ according to the mutual information gain (Section Feature Selection via Mutual Information). The goal is to identify features with highest spatial variability and therefore, most likely to be informative in denoting boundaries to which tissue-scale atlas regions might align. Indeed, choice of features with higher mutual information scores achieved significant differentiation in 10 times the number of pairs of regions as those features with lower mutual information with regard to the spatial distribution of feature (e.g. gene) expression within versus between the regions.

Here, we have demonstrated the efficacy of xIV-LDDMM, shown efficacious previously in a 2D setting[15], but now coupled to these schemes of data reduction and partial matching (censoring) for mapping a complete 3D CCFv3 to molecular (gene-based) and cellular transcriptomics datasets from different technologies (MERFISH and BARseq) and with varying scopes of capture from whole brain to hemi-brain sections. We have additionally demonstrated its efficacy at handling different tissue types and wholly 3D transcriptomics datasets with the mapping of multiple EMAP atlas timepoints along the continuum of development to a whole mouse embryo molecular dataset measured with cycleHCR[17].

Strengths of the approach presented here first include the efficiency of the image-varifold particle representation for 3D data sampled sparsely within a volume. Comparable regularized image grids require orders of magnitudes more data points (e.g. $10^3 - 10^6$) even without full resolution of adjacent measured data, and resulting in orders of magnitude more computations, which in the setting of distance metrics, are often quadratic in the number of data points. Second, the image-varifold representation coupled with the specific censoring method enhances the generalizability of our suite of technologies to accommodate mapping data with different scopes of capture and sampling schemes (e.g. spot-technologies versus single-cell measurement schemes) and to handle situations of drop out or poor data quality (e.g. disjoint and missing slices, as demonstrated in Supplementary Fig. 1). Indeed, in the advent of technologies with even deeper tissue measurements leading to wholly 3D captures (e.g. cycleHCR), our methodologies can easily adapt to settings of 2D sections versus these volumetric captures with further ability to integrate them into a single common coordinate framework. In contrast, many current schemes[44,45] are built to accomodate 2D sections which they treat independently, necessarily requiring apriori selection of the 3D atlas section to which to align each molecular/cellular section. This is challenging in typical settings in which tissue is not necessarily sectioned perpendicular to a given axis. Finally, a third strength of our approach is in our joint modeling of physical and feature space, as measured directly by the given technology and therefore affording easy biological interpretability of distance measures. While we introduce two reductionist schemes (e.g. scale-space resampling and mutual information-based feature selection), the reduced data representation still maintains the same inherent physical space and feature type, unlike many of the machine-learning based approaches which wholly transform data to a space often uninterpretable with regard to the biological underpinnings[9].

Amongst these strengths, the generalizability of our methods together with the following limitations offer areas for future work. With regard to generalizability, first, we have exhibited one axis of generalizability with the demonstration of our method for mapping molecular and cellular datasets

from a range of technologies (MERFISH, BARseq, cycleHCR) measuring both mouse brains and mouse embryos. While the focus of this manuscript has been on the description and evaluation of the data reduction and mapping schemes associated to xIV-LDDMM, one avenue of future work includes further exhibition of the generalizability of the approach to mapping molecular and cellular datasets from additional technologies (e.g. SlideSeq, synapse imaging[8]) with increasing variability and size as well as changing anatomy over time. Indeed, as evidenced in our mapping two separate time points of the EMAP embryo atlas to the whole mouse embryo measured with cycleHCR, the opportunity and need naturally presents itself for extension of the cross-modality image varifold mapping framework to incorporate longitudinal faculties involving the estimation of an optimal entry point of target data into a potential continuum of atlas time points. Similar longitudinal image-based LDDMM methods have been established with success at measuring shape changes overtime[35,46,47] and therefore provide the basis for development of such an image-varifold based pipeline in the future, as a second area of future work. Lastly, as described in ref. 15, estimation of a diffeomorphic transformation taking tissue-scale atlas to molecular or cellular-scale target affords the pulling back of each target into the atlas common coordinate framework via the inverse transformation. This consequently enables integration of measures across replicates and technologies for comparison of feature distributions across them. Hence, the continued development of post processing analysis techniques and biology-based interrogation of integrated and aligned datasets is a third avenue of future work.

Regarding feature modeling and selection, there are opportunities for future work both in refining the mutual information-based subsampling procedure and in expanding the feature spaces modeled per image-varifold object. First, we have described a greedy feature selection procedure looking at genes (or cell types) independently to select an informative subset. It is natural to study such selection in the context of non-independence, examining pairs and triples, with many cell types or genes distributed similarly with functional associations. This is something we are currently pursuing. Second, our cross-modality mapping method is rooted in an assumption of stationarity with respect to the distribution over features within the space of each atlas region. While the mutual information method as presented here selects features that typically enable significant differentiation between at least 50% of regions, the use of more tailored boundaries specific to atlas choice is a potential area for future investigation affording closer alignment with this assumption of stationarity. Here, instead, we have opted to consider more general horizontal and vertical planes as more tailored boundaries would inherently be more limiting in being application specific. Finally, though each dataset here was represented as an image varifold over a single physical and feature space, the coupled modeling of different feature spaces (e.g. genes and cell types) over which a single image varifold is defined naturally addresses the incorporation becoming prevalent at the stage of data collection. For instance, cell-based datasets, such as the BARseq example analyzed here, often carry both gene and cell-level information based on integration of raw measurements of mRNA with segmentation and clustering schemes (e.g. histological stains, clustering). The image-varifold representation lends itself to associating to physical spatial measures not just single feature measures but potentially feature measures over differing feature spaces (e.g. genes, cell types) that could simultaneously be used for mapping between tissue-scale atlases and these molecular/cellular scale datasets.

Lastly, a significant challenge of the xIV-LDDMM algorithm is the peta scale nature of the quadratic computation. At its core, each particle and feature are compared to each other one in the inner-product that generates the varifold norm. This computational complexity is illustrated by the run times on the order of 24 h for computing single mappings (see Supplementary Table 2). Our strategy thus far has been to exploit the parallelism of GPUs. The multi-scale resampling scheme described in Section Scale-Space Resampling naturally lends itself to the multi-scale mapping scheme initially introduced in ref. 12, in which successive refinements can be made at coarse to fine scales of transformations estimated with corresponding coarse to fine

approximations of data. Currently, multiple scales of transformations are estimated simultaneously with a single set of particles, thereby limited in coarseness and sparsity to the finest scale of desired transformation. In the multi-scale approach previously described[12], transformations are estimated first at a coarser scale with a coarser representation of data (e.g. fewer particles) needed to capture our atlas and target objects. Successive addition of finer scale representations for refining these coarse scale mappings can then be achieved with likely fewer iterations, reducing the overall computational load these mapping schemes encompass. We are currently working to adapt the scheme described here to a multi-scale setting for improved efficiency in the future and with the prospect also of accommodating different feature spaces over which particle representations might be defined at each scale.

## Methods
### The brain mapping model for molecular scales

Our molecular representations at the micron and submicron scales are represented mathematically as a set of discrete "particle" Diracs indexed by $i \in I$, each carrying a singular mass weight located in physical space at $(w_i \delta_{x_i})_{i \in I}$ and a conditional probability distribution $(p_i)_{i \in I}$ over a feature space[15] $\mu \doteq \sum_{i \in I} w_i \delta_{x_i} \otimes p_i$. The norm is associated to the brains modeled in the Hilbert space with inner product giving

$$\langle \mu, \mu \rangle_M \doteq \sum_{i,j \in I} w_i w_j K_\sigma(x_i, x_j) \sum_{f,g \in \mathcal{F}} K_F(f, g) p_i(f) p_j(g). \tag{6}$$

The normed-space takes the brains as a reproducing kernel Hilbert space[12,13,15] with space kernel, $K_\sigma$, typically a Gaussian or other forms of Matern kernels, and $K_F$ often used as an indicator function for gene or cell labels or a Euclidean inner product for real-valued features[13]. Thompson's brain mapping[14] is defined through the space of transformations defined as group actions by diffeomorphisms $\varphi \in Diff$,

$$\varphi \cdot \mu \doteq \sum_{i \in I} |D\varphi|_{x_i} w_i \delta_{\varphi(x_i)} \otimes p_i \tag{7}$$

The normed distance between deforming atlas and target is then:

$$\| \varphi \cdot \mu - \mu' \|_M^2 = \langle \varphi \cdot \mu, \varphi \cdot \mu \rangle_M - 2\langle \varphi \cdot \mu, \mu' \rangle_M + \langle \mu', \mu' \rangle_M \tag{8}$$

with the brain mapping problem transforming one brain to the other by minimizing

$$\inf_{\varphi \in Diff} \| \varphi \cdot \mu - \mu' \|_M^2.$$

Flows are introduced to generate the diffeomorphisms giving the variational problem.

Variational Problem 1.

$$\inf_{v \in L^2([0,1], V)} \frac{1}{2} \int_0^1 \| v_t \|_V^2 dt + \| \varphi_1^v \cdot \mu - \mu' \|_M^2$$
$$\text{with } \varphi_t^v = \int_0^t v_s \circ \varphi_s^v ds + Id, t \in [0, 1]. \tag{9}$$

### Mapping across scales to atlases

Tissue scale atlases have an associated partitioning scheme with individual regions indexed by $\ell \in \mathcal{L}$, (e.g. $|\mathcal{L}| < 1000$ for CCFv3), but do not generally carry the functional features of genes and cell-types. For diffeomorphic mapping of tissue-scale atlases onto the feature space of molecular/cellular scale targets, we associate the latent gene/cell-type feature laws to the atlas $\mu_A^P$. We index by partition class the non-normalized laws on features, $(p_\ell(f), f \in \mathcal{F}), \ell \in \mathcal{L}$, with normalized probability laws given as $\bar{p}_\ell = \frac{p_\ell}{\sum_{f \in \mathcal{F}} p_\ell(f)}$. Associated to each point in the atlas, per its representation as

an image-varifold object, is the probability of a partition class, $\pi_i(\ell), \ell \in \mathcal{L}$, which then dictates the molecular/cellular feature law to which it is associated, giving the atlas with estimated gene/cell-type features as:

$$\mu_A^p = \sum_{i \in I} w_i \delta_{x_i} \otimes \sum_{\ell \in \mathcal{L}} \pi_i(\ell) p_\ell.$$

Diffeomorphic transformation then acts on this object to give:

$$\varphi \cdot \mu_A^p \doteq \sum_{i \in I} |D\varphi|_{x_i} w_i \delta_{\varphi(x_i)} \otimes \sum_{\ell \in \mathcal{L}} \pi_i(\ell) p_\ell \quad (10)$$

which is aligned to molecular/cellular target. Both diffeomorphism and feature distributions, $\varphi$ and $(p_\ell)_{\ell \in \mathcal{L}}$ are estimated through minimization of the varifold normed distance between transformed atlas and target:

$$\| \varphi \cdot \mu_A^p - \mu' \|_M^2 = \langle \varphi \cdot \mu_A^p, \varphi \cdot \mu_A^p \rangle_M - 2\langle \varphi \cdot \mu_A^p, \mu' \rangle_M + \langle \mu', \mu' \rangle_M \quad (11)$$

which gives the variational problem associated to our cross-modality mapping scheme (xIV-LDDMM).

**Variational Problem 2.**

$$\inf_{\substack{v \in L^2([0,1], V), \\ p_\ell, \ell \in \mathcal{L}}} \frac{1}{2} \int_0^1 \| v_t \|_V^2 dt + \sum_{\ell \in \mathcal{L}} J_{KL}(p_\ell) + \| \varphi_1^v \cdot \mu_A^p - \mu_T \|_M^2$$

$$\varphi_t^v = \int_0^t v_s \circ \varphi_s^v ds + Id, \, t \in [0,1]. \quad (12)$$

Here note we have added a "prior" term $J_{KL}(p_\ell)$ weighting the probability laws towards the uniform distribution with KL divergence (see Section LDDMM Mapping Tissue-scale Atlas to Molecular/Cellular-scale Targets).

### Rigid alignment of tissue sections

A priori, individual sections in a given stack from a single specimen may not be aligned to each other. Hence, the 2D sections are first rigidly aligned to one another, independent of an atlas, as in ref. 48 to account for differences in orientations that result from the imaging process. To solve for the low-dimensional rigid motions to bring each section into alignment with each other, we directly minimize with LBFGS (with implementation adapted from that in PyTorch) the varifold norms between each section associating to them the rigid motions. Each section is centered around its origin. For a stack of $N$ sections, we fix first ($n = 1$) and last ($n = N$) sections (e.g. rostral-most and caudal-most) and then estimate for all intermediate sections a rotation, and 2D translation ($R_\theta, \tau$) minimizing the pairwise varifold normed distance between adjacent sections in the stack:

$$\min_{(\theta_n, \tau_n), 2 \leq n \leq N-1} \| \mu_1 - \mu_2^{\theta_2, \tau_2} \|_M^2 + \| \mu_{N-1}^{\theta_{N-1}, \tau_{N-1}} - \mu_N \|_M^2$$

$$+ \sum_{n=2}^{N-2} \| \mu_n^{\theta_n, \tau_n} - \mu_{n+1}^{\theta_{n+1}, \tau_{n+1}} \|_M^2 \quad (13)$$

$$\text{with } \mu_n^{\theta_n, \tau_n} \doteq \sum_{i \in I_n} w_i \delta_{R_{\theta_n} x_i + \tau_n} \otimes p_i.$$

The varifold normed distance $\| \cdot \|_M^2$ is defined as in (6), with $K_\sigma$ a gaussian kernel and $K_F$ the identity kernel.

### LDDMM mapping tissue-scale atlas to molecular/cellular-scale targets

To solve for the high-dimensional diffeomorphisms for the variational problems of (1) and (2), we directly minimize with LBFGS the varifold norms using geodesic shooting to generate the flow of diffeomorphisms

onto the target. Previously we have described alternating optimization schemes separating the unknown laws $(p_\ell)_{\ell \in \mathcal{L}}$ from the diffeomorphism using quadratic programming[13,15]. Here we use a single optimization scheme introducing regularization with constraints for $p_\ell > 0$.

As introduced in refs. 12,49, in geodesic shooting we model the velocity $v$ as the control of the dynamical system which is parameterized in the momentum variables of the system $(\rho_i^x, \rho_i^w)$ representing "space" (vector) momentum and "mass" (scalar) momentum, respectively. For Gaussian kernels, this gives:

$$v(x) \doteq \sum_{i \in I} k_\sigma(x, x_i) \rho_i^x + \frac{x - x_i}{\sigma^2} w_i \rho_i^w k_\sigma(x, x_i). \quad (14)$$

We define the Gaussian kernel $k_\sigma(x, y) = \exp(-\frac{\|x-y\|_2^2}{2\sigma^2})$ specifically with scale bandwith $\sigma$. The kernel $k_\sigma$ determines the spatial scale of the system of solutions defining the Green's kernel of the RKHS with norm $\| \cdot \|_V^2$ controlling the smoothness of the flows of diffeomorphisms[50].

In geodesic shooting, velocity of the system controls the flow and is reparameterized by only the initial value of momentum at time $t = 0, (\rho_0^x, \rho_0^w)$ which determines the entire geodesic path of the atlas to target. The Hamiltonian of the system determines the geodesic equations for the flow of the state $q_{it} = (x_{it}, w_{it})$, $t \in [0,1]$ and the momentum $(\rho_{it}^x, \rho_{it}^w, t \in [0,1])$ whose dynamics are given by the Hamiltonian system:

$$\dot{q}_i = \nabla_{\rho_i} H = \begin{cases} \dot{x}_i = v(x_i) \\ \dot{w}_i = \text{div}(v(x_i)) w_i \end{cases}. \quad (15)$$

$$\dot{\rho}_i = -\nabla_{q_i} H. \quad (16)$$

with

$$H(\boldsymbol{q}, \boldsymbol{\rho}, v) = \sum_{i \in I} (\rho_i^x)^T v(x_i) + \rho_i^w \text{div}(v(x_i)) w_i - \frac{1}{2} \| v \|_V^2,$$

where $\boldsymbol{x}, \boldsymbol{w}, \boldsymbol{\rho^x}, \boldsymbol{\rho^w}$ denote the set of variables per particle $i \in I$.

As each particle measure is associated to a pair of momenta $(\rho_i^x, \rho_i^w)$, the total parameter set is $o(|I|)$, parameterizing the estimated control.

In the joint optimization scheme, we additionally need to constrain our estimates of the latent gene/cell-type feature laws. To constrain these $p_\ell > 0$, we use a penalty defined by KL divergence of $\bar{p}_\ell$ (the normalized probability law) to a uniform distribution $\frac{1}{|\mathcal{F}|}, \forall f \in \mathcal{F}$. We weight this cost, $J_{KL}(p_\ell)$ for each $\ell \in \mathcal{L}$ according to what fraction of the overall mass in transformed atlas this distribution contributes to, with most weight attributed to larger atlas regions that correspondingly map to areas of high target density and lowest weight attributed to smaller atlas regions and those mapping to areas of low target density:

$$J_{KL}(p_\ell) \doteq \frac{M_\ell^A}{\sum_{f \in \mathcal{F}} M_f^T} \sum_{f \in \mathcal{F}} p_\ell(f) \log(\frac{\bar{p}_\ell(f)}{1/|\mathcal{F}|}), \quad (17)$$

with $M_\ell^A$, the total atlas mass of region $\ell$ following deformation and $M_f^T$, the total target mass of feature $f$, computed in each case as the sum over the entire image varifold object.

We use LBFGS to optimize these pairs of momenta per particle together with the distributions $(p_\ell)_{\ell \in \mathcal{L}}$ and bandwidth $\lambda$, for controlling the censoring function, so as to minimize the cost as given in (12), with the regularization on $v_t$ re-parameterized with the Hamiltonian:

$$H(\boldsymbol{q}, \boldsymbol{\rho}, v) + \sum_{\ell \in \mathcal{L}} J_{KL}(p_\ell) + \| \varphi_1^v \cdot \mu_A^{p,\lambda} - \mu_T \|_M^2. \quad (18)$$

### Mutual information scoring for feature selection

To deduce which features are spatially variant, we assign to each feature a score based on mutual information. This score aims to capture not global

variance in a feature distribution, but rather, within a local neighborhood, whether the spatial distribution of a given feature value (e.g. gene, cell type) is organized along particular boundaries. We simplify these boundaries to be vertical or horizontal lines only, and use mutual information to deduce how closely the distributions of feature values organize along these boundaries within local neighborhoods, sized according to a chosen scale.

In detail, this score specifically measures the mutual information between a random variable, $M^g$, that reflects the number of counts of feature $g$ (e.g. a gene type, a cell type) in a given neighborhood, and a random variable, $X$, that partitions this neighborhood vertically or horizontally into two domains. We describe, here, a method for computing this score particularly in settings of large amounts of data, where discretization is favorable for computational efficiency. As an example, we specifically describe the setting of serial tissue sections measured for gene expression, as with MERFISH or BARseq. The method, as illustrated in Supplementary Fig. 4, is applied for each gene independently on each measured section of tissue, where collective scores per gene can be garnered by tallying each gene's score per section across the entire set of sections.

The support of the tissue section is first covered by a grid, as shown in the left panel of Supplementary Fig. 4, with squares of size $\sigma \times \sigma$. In the results shown in Section Feature Selection via Mutual Information, we choose $\sigma = 50\,\mu m$. In each square, we compute the total number of mRNA expressed per each gene in that square, denoted by $N^g$ for gene $g$. Let $F^g(t) = P(N^g \le t)$ be the cumulative distribution function for gene $g$, estimated from the empirical distribution of $N^g$ across all squares in our grid. We define the binning function $\phi^g(n) = \sum_{k=1}^{q} \mathbf{1}_{n \ge t_k}$ for $t_k = \inf\{t \ge 0 | F^g(t) \ge k/q\}$ and with $k \in [1, q]$ denoting the $k$-th $q$-quantile. This gives a discrete (normalized) value of mRNA counts for gene $g$ in each square of the grid, as shown in the middle panel of Supplementary Fig. 4 for $g =$ Gfap.

We define our discrete neighborhoods as megasquares, denoted $(Q_c)_{c \in \mathcal{C}}$, with each comprised of a continuous set of $2K \times 2K$ grid squares. We consider all possible megasquares that can be defined across the grid, and index the squares within each megasquare by column index $i = 1, \ldots, 2K$ and row index $j = 1, \ldots, 2K$, giving $Q_c = \bigcup_{(i,j) \in \{1,\cdots,2K\}^2} Q_{c,i,j}$. Finally, we define two partitioning schemes, denoted $\updownarrow$ and $\leftrightarrow$, corresponding to the partitioning of a megasquare into two equal vertical or two equal horizontal domains, with each domain in each scheme containing $2K^2$ squares. The right panel of Fig. 4 shows a sample of 4 megasquares from the entire set $(Q_c)_{c \in \mathcal{C}}$ that cover the grid.

The random variables of interest, $X$ and $M^g$ are specified as functions of $\omega = (c, i, j, d) \in \Omega$ with $\Omega = \mathcal{C} \times [1, 2K]^2 \times \{\updownarrow, \leftrightarrow\}$, the set of all possible selections of megasquare, square within the megasquare, and partitioning of the megasquare. Specifically, we denote $C(\omega) = c$, the index of the megasquare, $N^g(\omega)$ the counts of gene $g$ for the square $Q_{c,i,j}$ in megasquare, $c$, giving $M^g(\omega) = \phi(N^g(\omega)) \in [1, q]$, the q-quantile of the gene count, and $X(\omega) \in \{b, t, l, r\}$, the partition $Q_{c,i,j}$ belongs to, dictated by direction $d$ in $\omega$ as:

$$X(\omega) = \begin{cases} l & \text{if} & d = \leftrightarrow, & i \le K \\ r & \text{if} & d = \leftrightarrow, & i > K \\ b & \text{if} & d = \updownarrow, & j \le K \\ t & \text{if} & d = \updownarrow, & j > K \end{cases} \tag{19}$$

Choice of $\omega$ is made uniformly, with $P = \frac{1}{|\Omega|} \sum_{\omega \in \Omega} \delta_\omega$. Our score is thus, the conditional mutual information between $X$ and $M^g$ given $C$:

$$\begin{aligned} &I(X; M^g | C) \\ &= \sum_{c,x,m} P(X = x, M^g = m, C = c) \log\left(\frac{P(X = x, M^g = m | C = c)}{P(X = x | C = c)P(M^g = m | C = c)}\right) \end{aligned} \tag{20}$$

### Optimization based resampling algorithm for multi-scale representation

Section Scale-Space Resampling introduced the scale-space resampling method for generating data approximations at different scales. We generate a hierarchy of approximation measures $\mu_\sigma, \sigma_1 > \sigma_2 \ldots$ with increasing complexity as we descend in scale, $|I_{\sigma_1}| < |I_{\sigma_2}| < \ldots$. Algorithm 1 describes

the generation of this reduced set of particles at scale, $\sigma$, where the physical locations, $\tilde{x}_i$, mass weights, $\tilde{w}_i$, and conditional probability distributions, $\tilde{p}_i$ for each particle in our reduced set $i \in \tilde{I}$ are optimized according to (5).

**Algorithm 1**. Initialization:

1. Define lattice spanning support of data, with cubes $\propto \sigma^3$.
2. For each cube containing at least one high resolution particle:
   (a) Select one particle, $i \in I$, in the cube at random.
   (b) Initialize a particle, $\tilde{i} \in \tilde{I}$, as $\tilde{w}_i \delta_{\tilde{x}_i} \otimes p^u$, where $\tilde{w}_i = \sum_j w_j$ for all particles, $j$, within the cube and $p^u(f) = \frac{1}{|\mathcal{F}|} \, \forall f \in \mathcal{F}$ (the uniform distribution).

**Single optimization:**

1. Fix $\tilde{x}_i \, \forall i \in \tilde{I}$.
2. Minimize (5) with respect to all $\tilde{w}_i, \tilde{p}_i$ with LBFGS for 200 iterations.

**Joint optimization:**

1. Minimize (5) with respect to all $\tilde{x}_i, \tilde{w}_i, \tilde{p}_i$ with LBFGS for 200 iterations.

An implementation of Algorithm 1 utilizing LBFGS from PyTorch can be found at https://github.com/kstouff4/VarifoldApproximation/. The implementation was developed and tested on a single Quadro RTX 8000 GPU with 48GB memory. To increase speed efficiency, particles in both the high resolution and reduced set were first grouped into cubes and ordered in memory according to physical location. Total kernel operations needed in each optimization scheme were reduced by relegating all those between particles in cubes whose centers were at a distance greater than threshold (e.g. $4\sigma$) a value of 0 without active computation. To reduce active memory load, larger datasets were halved or quartered along axes of physical space, with each section within a stack of sections aggregated into a reduced set of particles independently. The resulting reduced particle sets for each fraction were rejoined after optimization into a single image varifold object.

### Estimating the censoring function

For brains with censored planes corresponding to the rostral-caudal directions only (e.g. whole coronal sections), we apply a censoring function oriented only to the rostral-caudal axis. This function of space is given in molecular target coordinates as the sum of two hyperbolic tangent functions, giving for a location $x \in \mathbb{R}^3$:

$$\alpha_x^\lambda \doteq \frac{1}{2}\left(\tanh\left(\frac{\langle x - a_0, n_0 \rangle}{\lambda}\right) + \tanh\left(\frac{\langle x - a_1, n_1 \rangle}{\lambda}\right)\right)$$

with $a_0, a_1$ two points in the first and last coronal sections of the molecular data, respectively, and $n_0, n_1$ the respective normal vectors to the plane of each section, pointing towards the interior sections of the dataset. Support weights, $\alpha_\lambda(x)$ are thus within the range, $[0, 1] \, \forall x \in \mathbb{R}^3$, with $\alpha_\lambda(x) = 1$ indicating the support of the molecular dataset and 0 outside the support, with the bandwith $\lambda$ controlling the width of the transition zone from inside to outside the support with $0 < \alpha_\lambda(x) < 1$. We estimate $\lambda$ jointly with geometric transformations and feature distributions, with added regularization cost:

$$J_s(\lambda) \doteq \frac{\lambda^2}{0.1} \ln\left(\frac{\lambda^2}{0.1}\right) + 1 - \frac{\lambda^2}{0.1}. \tag{21}$$

In the setting of stacks of hemibrain sections, as described in ref. 18, the support weights necessarily capture boundaries around the target measured dataset at rostral, caudal, and medial edges. The medial boundary typically varies along the rostral-caudal axis with differences in the extent of tissue capture per section measured. To accommodate this irregularity with a smooth transition function from the interior to exterior of the support, we

take the support weight function, $\alpha_\lambda$, to be the output of a UNET trained to delineate interior support from exterior in the target physical space.

Training data is generated at a scale of 200 μm, where high resolution particle measures have been aggregated into a reduced set of particle measures, of size $o(1000)$ per section (see Section Scale-Space Resampling). To each section, 15–20 particles are placed manually along the medial boundary. A label of 1 is assigned to all particles in the aggregation and 0 to those particles manually added to each section. The particles per section are stacked according to given spacing (e.g. 200 μm) and treated as a single 3D object for estimation of the support. Added particles with label 0 are weighted 1000 : 1 to balance the discrepancy in numbers of particles contained within versus outside the support.

Here, we use the NeuralNetClassifier from skorch to implement a 3 layer neural network. Two linear layers, sized $3 \times 15$ and $15 \times 5$, respectively are each followed by an exponential linear unit. The final output linear layer, sized $5 \times 1$, is smoothed with a hyperbolic tangent, analogous to the setting of single axis weights described above (see Section Estimating the Censoring Function) and shifted and rescaled to be in the interval [0, 1]. As in (21), a bandwith parameter, $\lambda$, is estimated in tandem with geometric and feature transformations, controlling the width of the transition zone from interior to exterior of the support.

## Statistics and reproducibility

Throughout this work, we have demonstrated the mapping capabilities of our technologies on datasets measured from three separate technologies: MERFISH, BARseq, and cycleHCR. The results presented can be reproduced accordingly using the algorithmic implementations provided in the associated coding repositories[51,52] together with these publicly available datasets[3,17,53,54] and the publicly available CCFv3 mouse brain atlas[4] and EMAP mouse embryo atlases[16]. Results between and across these technologies were compared qualitatively and not statistically differentiated, with both MERFISH and BARseq offering single samples of whole mouse brain coronal sections, BARseq only offering samples of hemi-brain coronal sections, and cycleHCR measuring a whole mouse embryo.

Statistical testing was conducted in the setting of examining the stationarity assumption integral to our estimation of latent distribution laws over gene or cell types per atlas region. To test this assumption for both genes with high mutual information scores and genes with low mutual information scores, we used pairwise two-sided Mann–Whitney U tests for pairs of atlas regions to assess inter-regional versus intra-regional differences in gene expression as measured per spatially distributed particle (see Section Feature Section via Mutual Information). Significance was determined at the level of 0.05, corrected for multiple comparisons using a bonferroni correction.

## Reporting summary

Further information on research design is available in the Nature Portfolio Reporting Summary linked to this article.

## Data availability

Serial MERFISH sections from the Allen Institute were produced under the BRAIN Initiative Cell Census Network (BICCN, www.biccn.org, RRID:SCR_015820) and are available at the Brain Image Library (BIL, https://www.brainimagelibrary.org/index.html) under https://doi.org/10.35077/g.610[3]. Serial hemi-brain BARseq sections with cell-level data are available at Mendeley data (https://doi.org/10.17632/8bhhk7c5n9.1[53] and 10.17632/5xfzcb4kn8.1[54]). Serial whole-brain BARseq sections are available at Mendeley data (https://doi.org/10.17632/byp7p32gpx)[55]. The Allen CCFv3 used in this study is available at https://download.alleninstitute.org/informatics-archive/current-release/mouse_ccf/annotation/ccf_2022/.

## Code availability

Implementation of the optimized aggregation algorithm and mutual information score for data reduction can be found at: https://github.com/

kstouff4/IV-Particle.gitv1.0.0[51]. Implementation of cross-modality mapping (xIV-LDDMM) can be found at: https://github.com/kstouff4/xIV-LDDMM-Particlev1.0.0[52].

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

## Acknowledgements

This work was supported by the National Institutes of Health (1F30AG077736-01 and T32-GM13677 (K.S.); R01EB020062, R01NS102670, U19AG033655, P41-EB031771, and R01MH105660 (M.M.); NIH Brain Initiative Grant U19MH114830 (H.Z.); DP2MH132940, R01MH133181, and U01NS132161 (X.C.)); the National Science Foundation (NSF) (16-569 NeuroNex contract 1707298 (M.M.); the Computational Anatomy Science Gateway (M.M..) as part of the Extreme Science and Engineering Discovery Environment (XSEDE Towns et al., 2014), which is supported by the NSF grant ACI1548562; NSF 2124230 and NSF 2309683 (L.Y.)); and the Kavli Neuroscience Discovery Institute supported by the Kavli Foundation (M.M.). We acknowledge Josua Sassen for his work on structuring the NNET classifier used for estimating target support weights.

## Author contributions

M.M., A.T., and L.Y. developed the mathematical theory of the method described in the manuscript. K.S. and M.M. drafted the manuscript. K.S., B.C., and A.T. generated codes for algorithms described in the manuscript. K.S. created the figures in the manuscript. H.Z. generated serial MERFISH data. X.C. generated and cell typed the BARseq data. All authors contributed to the editing of the final manuscript.

## Competing interests

Under a license agreement between AnatomyWorks and the Johns Hopkins University, M.I.M. and the University are entitled to royalty distributions related to technology described in the study discussed in this. M.I.M. is a founder of and holds equity in AnatomyWorks. This arrangement has been reviewed and approved by the Johns Hopkins University in accordance with its conflict of interest policies. The remaining authors declare no conflicts of interest.
