## [Transparent Peer Review file · Communications Biology]

The xIV-LDDMM Toolkit of Image-Varifold Based Technologies for Mapping 3D Images and Spatial-omics Across Scales

Corresponding Author: Ms Kaitlin Stouffer

Version 0:

Reviewer comments:

Reviewer #1

(Remarks to the Author)

The paper describes a collection of methodologies for integrating spatial transcriptomics measurements in brains to image data and atlases at the tissue scales, thereby addressing the significant underlying differences in scale and modality. The methodologies are based on the framework of geodesic diffeomorphisms, in particular on the idea to estimate optimal nonlinear mappings between the spatial coordinates of measurements and the underlying latent feature distributions simultaneously. This principle has been described as a methodological framework called xIV-LDDMM in a previous publication by some of the authors (Stouffer et al., Nat Comms 2024) and can be considered a state of the art solution.

While the previous work focused on mapping of mouse brain atlases to 2D tissue sections from BARseq, MERFISH and histopathology, the present paper extends the xIV-LDDMM framework by a range of solutions that enable their use for larger and more complex 3D measurements with only partial fields of view. These solutions include a mathematical extension for incorporating the partial volume masking, and reduction principles for spatial resolution (a greedy space-scale approximation approach) and feature depth (a selection approach based on mutual information) to reduce the high computational complexity of the overall approach for larger datasets. Overall, the framework makes an important contribution for integrating strongly growing sub cellular data resources to atlas-based reference frameworks at the level of the whole organ. The paper therefore has significant relevance especially for the cell biology and neuroscience communities. However, it has to be considered and studied in close conjunction with the previous publication on the xIV-LDDMM approach, and the clear demarcation and connection with this important previous work represents a certain challenge for the reader.

The manuscript is overall very well written; the language and technical depth are of a high standard. I did not find any significant flaws or inconsistencies which might prevent its publication. The results are original, presented in an appropriate fashion and should be of significant interest for the audience of the journal, in particular to researchers in cell biology and brain mapping.

The paper is structured along four main contributions, which are applied for mapping two already published (MERFISH and BARseq) and one new (BARseq) datasets from the mouse brain to reference atlases.

The first is an extension of the particle Dirac model in the xIV-LDDMM approach with a censoring function that is controlled by a smoothness parameter to be optimized together with the diffeomorphic mapping function. This allows explicit estimation of the measured partial volume as a support function over the whole brain space in the atlas. The efficacy of the approach is demonstrated by visualization of the resulting support regions for the partial mouse brain volumes in the experimental data. The second contribution improves the computation of image varifold norms between molecular and tissue-scale measurements wrt. the previous work, which jointly address distance measures in coordinate space and feature space. For this specific part (paragraph from l. 185; Sec. 2.4; and corresponding entries in the methods), I found it tricky to understand the demarcation from the previous work without going deep into the methods, and think it would be helpful to point it out more clearly at a higher level in Sec. 2.

The third contribution is a scale-space particle approximation scheme for determining complexity levels for the cross-modality mappings, effectively identifying optimal neighborhood and kernel sizes used for resampling the data and saving computational complexity. The approach integrates naturally with the mapping framework and is shown to be more effective

than using explicit clustering and regridding schemes.

The fourth contribution is a feature selection mechanism based on information theory, which uses mutual information between sub cellular features and partition boundaries to implement a greedy optimization strategy. It is evaluated by discussing the plausibility of particularly low and high scores obtained for genes that provide good interpretability in this respect.

The methodology as a whole is assessed by comparing independently computed clusters of cell types in the (whole-brain section) example datasets to cortical layer structures mapped from the CCFv3 atlas using the proposed framework. Considering the clusterings as an intrinsic reference of the data, the evaluation reports e.g. fractions of cells that were aligned to the expected layers and statistics of their distances to the correct layers. Furthermore, distributions of selected spatially variable genes from the MERFISH dataset are plotted before and after the alignment with atlas region to examine the degree of alignment with the regions. The results are convincing, but can of course only provide a qualitative assessment. Here, a more quantitative presentation and discussion of gene expressions per brain region in terms of the improvement in intra- and inter-region variance following alignment would have been helpful to further support the assessment.

The paper concludes that the proposed methodologies enable the matching of 3D partial volumes from single-cell transcriptomics to tissue atlases, and that the approach enables solutions for „peta-scale“ datasets. While I see convincing qualitative and some quantitative support of the first in the experiments, I do not find the aspect of petq-scale data discussed so explicitly in the paper. The discussion and reporting of compute times is not systematic wrt. to scalability with growing data sizes, but rather descriptive in terms of the effect of complexity reduction through the proposed selection mechanisms in physical and feature space. I therefore suggest to explain more explicitly in the introduction and discussion what „peta-scale complexity“ implies and how the proposed framework can manage it.

The paper is of high technical quality and methodological depth, which can also make it difficult for readers who are not so familiar with the principles of diffeomorphic mappings and related physical and mathematical concepts. From my perspective, it would be helpful if Sec. 2 would assume less technical background and include a couple more high-level explanations of the mathematical concepts. For example, Sec. 2.3 assumes solid knowledge of the previous paper on xIV-LDDMM, and would benefit from a couple of brief higher-level explanations of relevant terms such as the Jacobian determinant in (4b), the product operator in (2), and the concept of Hilbert spaces.

I found the diagram in Fig 1A quite helpful to understand the high-level scope of the work, but it is in my eyes not as attractive and prominently placed as it deserves to be (especially compared to the good quality of other figures). I suggest to make Fig 1A visually more attractive by exposing the different levels therein (methods, measurements, output & uses) more obviously, e.g. by stronger visual grouping rather than only using font colors. Ideally, the figure would appear closer to Sec 2.1 already, which it supports nicely.

I find it important for the reader to properly understand the motivation and validity of the underlying stationary probability law assumptions of atlas ontologies for the mapping, which drive the alignment processes towards mapping homogenous areas in feature space to delineated brain areas. In the strong multimodal setting addressed here, and depending on the type of reference parcellation, features might show significant intra-region variance wrt. to the atlas ontology and not always fulfill this assumption. One might argue that this is rather an exception to the rule, and that the feature selection strategies will help in identifying features with strong support of the atlas maps, but such discussion is coming rather short in the paper given its relevance for the topic - most at the start of Sec. 2.8. I would suggest to expand this topic a bit more and motivate it already more explicitly in the introduction of the paper.

Lastly, I should say that I did not feel sufficiently familiar with the mathematical foundations of variation calculus and the statistical properties of genes to fully assess the math in subsections 4.2, 4.4 and 4.5. The derivations and explanations appear plausible and solid to me, but I was not able to validate all derivations therein in detail.

While reading, I found a smaller number of typos and inconsistencies which should be addressed in a final version:

- * In Eq. (3), I was partly confused whether λ is used as subscript or exponent in the different places. I assume it is a subscript for μ but an exponent for α - is this correct?
- * I find the distinction of the terms „mapping“ and „alignment“ not trivial, since they have slightly different meanings in the communities. Given their central role in the paper (e.g. already in Fig 1.A), I suggest to clarify their interpretation briefly at the beginning of the paper.
- * In l. 110, it is at first unclear what M is. I suggest to introduce more explicitly.
- * In l. 112, „Diff“ not properly formatted
- * The determinant of Jacobian $|D_\phi|$ in (4b) should be briefly explained in the text
- * l. 339, typo: „similtude“
- * l. 525, word missing?
- * l. 591, should that be „totalling“?
- * l. 645 should be „Kullback Leibler“
- * L 385, grammar? „E,F show single intersecting section of deformed“
- * Refs 11 and 14 incomplete or wrongly formatted

(Remarks to the Author)

Summary:

This paper introduces xIV-LDDMM, a computational toolkit for integrating spatial-omics and imaging data across scales. The framework leverages image-varifold norms for cross-modality mapping and implements multi-scale optimization and mutual-information-based feature selection to align molecular and tissue-scale datasets. The authors demonstrate its utility on mouse brain datasets from MERFISH and BARseq technologies, showcasing robust alignment of partial and whole-brain data with the Allen CCFv3 atlas.

Review Points:

1. Limited Generalizability Assessment: The toolkit is evaluated primarily on mouse brain datasets, limiting insights if the method works equally well on other organisms or technologies, e.g., the authors could consider additional datasets like SlideSeq or on larger human/primate datasets to substantiate its broader utility.
2. Inadequate Quantitative Benchmarks: While the paper provides qualitative examples and runtime analysis, it lacks comprehensive quantitative comparisons with alternative mapping methods such as SpatialPCA, BayesSpace, or other generative approaches. Established metrics for spatial-omics alignment (e.g., mutual information, accuracy of boundary delineations) should be incorporated to strengthen benchmarking claims.
3. Insufficient Literature Context: The comparison focuses on older methods (e.g., K-means, grid resampling) but omits more recent, competitive approaches. The paper does not discuss advanced deep learning-based models, which could serve as relevant benchmarks for multi-scale mapping.
4. Scalability Challenges: The method exhibits significant computational overhead, requiring up to 24 hours per mapping iteration. Although GPU parallelization is used, the discussion of scalability for larger datasets remains speculative. Strategies to reduce computational costs, such as hybrid multi-scale approaches, could enhance practicality.
5. Limited Discussion of Limitations: The paper provides little discussion on potential constraints, such as failure modes in complex datasets, sensitivity to input data quality, and computational scalability. A deeper examination of these factors would provide valuable context for future applications.

Recommendation:

The manuscript is a promising contribution to the field. In addition, I recommend the authors to:

- Test the method on additional datasets from other species or spatial-omics technologies.
- Incorporate established quantitative metrics for comparison with state-of-the-art methods.
- Expand benchmarks to include deep learning-based approaches.
- Optimize computational efficiency and assess scalability.
- Provide a thorough discussion of limitations and failure modes.

Reviewer #3

(Remarks to the Author)

[Summary]

Image-based technologies (e.g., BARseq, MERFISH) sample irregularly across tissue, making it crucial to integrate and align data with reference coordinate systems. Thus, the authors proposed a diffeomorphic mapping suite for 3D image-transcriptome registration. This method extended existing 2D image-transcriptome registration method, xIV-LDDMM, into 3D image voxel space and provided a comprehensive suite for its application by adding modules that mapping whole tissue volume to subvolume, multi-scale resampling, feature selection.

The suite consists of four modules. The first module enables matching partial to whole tissue volumes with censoring, second module conducts cross-modality mapping based on an image-varifold measure norm, third module scale-space optimization for multi-scale resampling, and the fourth module enables mutual information-based feature (i.e. gene, cell) selection. The authors mapped the molecular varifold of MERFISH and BARseq dataset into the reference 3D image template of mouse brain, CCFv3, and demonstrated that this suite can provide correct transformation and alignment of CCFv3 to MERFISH/BARseq varifold. By comparing runtime/memory efficiency across multiple resolutions, authors also demonstrated that this suite is efficient to represent extremely large-scale images (peta-scale) and include algorithms that enable mapping across scales, from nanometers to millimeters.

[Comments]

[Originality & Significance]

- Authors emphasise on the efficacy of 3D mapping and the usability of suite rather than emphasizing novelty. Nonetheless, the evaluation over runtime/memory efficiency hardly backs up the significance of efficiency-the comparison is conducted without baseline.
- Since the authors emphasize that efficient peta-scale mapping is possible with this suite and compared runtime/memory efficiency across multiple resolutions, it would be valuable to provide qualitative/quantitative measures of efficiency compared to baseline methods. For instance, comparisons with diffeomorphic-based baseline methods, such as the mesh-based diffeomorphic mapping in mesh-based diffeomorphic mapping in [2] and another LDDMM-based spatial dataset mapping method.

[1] Clifton, Kalen, et al. "STalign: Alignment of spatial transcriptomics data using diffeomorphic metric mapping." Nature

communications 14.1 (2023): 8123.

[2] Stouffer, Kaitlin M., et al. "Cross-modality mapping using image varifolds to align tissue-scale atlases to molecular-scale measures with application to 2D brain sections." *Nature Communications* 15.1 (2024): 3530.

[Clarity]

- The mapping from 3D images to 2D images assumes stationarity in the probability law of molecular/cellular features over each atlas region, making the mapping dimensionality-agnostic. However, gene expression measured from CCFv3 and MERFISH/BARseq may differ in gene panels and techniques. Is this assumption valid? It would be helpful to provide evidence supporting this assumption.

[Quality]

- Figures except figure 5 provides qualitative measure, the quality of alignment for few selected sections. Qualitative measures that summarize all the alignments would back up the alignment performance of the xIV-LDDMM toolkit.
- The results presented in Figures 1 and 2 demonstrate whole tissue volume to subvolume mapping when given consecutive sections. However, does the alignment perform well even if the sections are disjoint?

Version 1:

Reviewer comments:

Reviewer #1

(Remarks to the Author)

The remarks and suggestions on the first version of the paper have been carefully and thoughtfully addressed by the authors. The revision has significantly improved the clarity and flow of the manuscript, and additional more quantitative measures have been added. I appreciate the effort put into the revision and responses. I think that the manuscript can be published.

Reviewer #3

(Remarks to the Author)

The authors have addressed the suggestions thoroughly, especially by clearly demonstrating the stationary assumption with the data, additional experiments on mapping disjoint sections.

[Clarity]

One question left is the approximate pixel dimensions of each section image used in the study. The authors claim a novelty in being able to perform alignment without patching (lines 154–156, 112–124), which differs from previous studies on whole-slide histopathology images where patching is typically necessary due to the gigapixel scale of the data. In such cases, it is generally infeasible to feed the entire image into a model without dividing it into patches. Is successful alignment without patching in this study is mainly due to specific components of the model, such as partial mapping or resampling? If it was simply because the section images used were small enough in size that patching was not required in the first place.

Please find below each of the reviewer’s comments copied in black text. Our responses are written in blue text with specific changes made to the manuscript file copied over in red text beneath each response where relevant.

1 Reviewer 1

The paper describes a collection of methodologies for integrating spatial transcriptomics measurements in brains to image data and atlases at the tissue scales, thereby addressing the significant underlying differences in scale and modality. The methodologies are based on the framework of geodesic diffeomorphisms, in particular on the idea to estimate optimal nonlinear mappings between the spatial coordinates of measurements and the underlying latent feature distributions simultaneously. This principle has been described as a methodological framework called xIV-LDDMM in a previous publication by some of the authors (Stouffer et al., Nat Comms 2024) and can be considered a state of the art solution.

While the previous work focused on mapping of mouse brain atlases to 2D tissue sections from BARseq, MERFISH and histopathology, the present paper extends the xIV-LDDMM framework by a range of solutions that enable their use for larger and more complex 3D measurements with only partial fields of view. These solutions include a mathematical extension for incorporating the partial volume masking, and reduction principles for spatial resolution (a greedy space-scale approximation approach) and feature depth (a selection approach based on mutual information) to reduce the high computational complexity of the overall approach for larger datasets. Overall, the framework makes an important contribution for integrating strongly growing sub cellular data resources to atlas-based reference frameworks at the level of the whole organ. The paper therefore has significant relevance especially for the cell biology and neuroscience communities. However, it has to be considered and studied in close conjunction with the previous publication on the xIV-LDDMM approach, and the clear demarcation and connection with this important previous work represents a certain challenge for the reader.

The manuscript is overall very well written; the language and technical depth are of a high standard. I did not find any significant flaws or inconsistencies which might prevent its publication. The results are original, presented in an appropriate fashion and should be of significant interest for the audience of the journal, in particular to researchers in cell biology and brain mapping.

The paper is structured along four main contributions, which are applied for mapping two already published (MERFISH and BARseq) and one new (BARseq) datasets from the mouse brain to reference atlases. The first is an extension of the particle Dirac model in the xIV-LDDMM approach with a censoring function that is controlled by a smoothness parameter to be optimized together with the diffeomorphic mapping function. This allows explicit estimation of the measured partial volume as a support function over the whole brain space in the atlas. The efficacy of the approach is demonstrated by visualization of the resulting support regions for the partial mouse brain volumes in the experimental data.

The second contribution improves the computation of image varifold norms between molecular and tissue-scale measurements wrt. the previous work, which jointly address distance measures in coordinate space and feature space. For this specific part (paragraph from l. 185; Sec. 2.4; and corresponding entries in the methods), I found it tricky to understand the demarcation from the previous work without going deep into the methods, and think it would be helpful to point it out more clearly at a higher level in Sec. 2.

1. We thank the reviewer for their attention to this link to our previous work in 2D and welcome the opportunity to clarify. We have re-organized our manuscript to help emphasize how the technologies we are specifically highlighting as novel in this manuscript are the censoring, scale-space resampling, and feature selection, all of which were developed to handle 3D data in particular, as presenting more challenges computationally with regard to its diversity and scale. The benefits of these new technologies are specifically in their being constructed on the same data representation (image-varifold) as our established mapping technology (xIV-LDDMM) and therefore, their seamless integration into this mapping technology for mapping more variable datasets to one another across scales, ultimately aligning them into the same coordinate system for further evaluation and association of the signatures across them. We clarified this through our revised Figure 1 that distinguishes this technology (blue) from the ones introduced in this manuscript (orange). We have also rearranged the ordering of our sections first to review the highlights of this established technology (Section 2.3), as also suggested by the reviewer below. Finally, we have added indications of what is specifically emphasized in this manuscript as novel technologies versus integration with our previous technology throughout the introduction and Section 2.1. Our specific changes are summarized below.

We changed the abstract as follows:

This paper describes a collection of technologies that we have developed for **mapping data across scales and modalities, such as genes to tissues, specifically in a 3D setting**. Our collection of technologies include (i) an explicit censored data representation for the partial matching problem mapping whole brains to subsampled subvolumes, (ii) a multi, scale-space optimization technology for generating resampling grids optimized to represent spatial geometry at fixed complexities, and (iii) mutual-information based functional feature selection. **We integrate these technologies with our cross-modality mapping algorithm through the use of image-varifold measure norms to represent universally data across scales and imaging modalities.**

We changed the following in the introduction with regard to distinguishing what is novel in this manuscript specifically from the platform we have described before.

We describe here the collection of technologies we have developed to address these challenges: independent modules for censoring, scale-space optimized particle approximations, and optimized feature selection. **Rooted in the unifying image-varifold representation of data, these technologies can integrate with our established cross-modality image-varifold based large deformation diffeomorphic metric mapping scheme (xIV-LDDMM, [1]) to achieve alignment across scales and modalities in 3D.**

We changed Section 2.1 as indicated in our response to the comment below in relation to the revised Figure 1.

The third contribution is a scale-space particle approximation scheme for determining complexity levels for the cross-modality mappings, effectively identifying optimal neighborhood and kernel sizes used for resampling the data and saving computational complexity. The approach integrates naturally with the mapping framework and is shown to be more effective than using explicit clustering and regridding schemes.

The fourth contribution is a feature selection mechanism based on information theory, which uses mutual information between sub cellular features and partition boundaries to implement a greedy optimization strategy. It is evaluated by discussing the plausibility of particularly low and high scores obtained for genes that provide good interpretability in this respect.

The methodology as a whole is assessed by comparing independently computed clusters of cell types in the (whole-brain section) example datasets to cortical layer structures mapped from the CCFv3 atlas using the proposed framework. Considering the clusterings as an intrinsic reference of the data, the evaluation reports e.g. fractions of cells that were aligned to the expected layers and statistics of their distances to the correct layers. Furthermore, distributions of selected spatially variable genes from the MERFISH dataset are plotted before and after the alignment with atlas region to examine the degree of alignment with the regions. **The results are convincing, but can of course only provide a qualitative assessment. Here, a more quantitative presentation and discussion of gene expressions per brain region in terms of the improvement in intra- and inter-region variance following alignment would have been helpful to further support the assessment.**

2. We thank the reviewer for their support of the relevance and efficacy of our method and for their suggestion for more quantitative assessment of the efficacy. Regarding quantitative metrics, we have added quantitative measures for our added examples of mapping a hemi-brain to hemi-brain and an atlas to a cycleHCR embryo with the added section 2.9 in the revised manuscript (see detailed response to reviewer 2's first comment). Notably, we have harnessed the correspondence in specific cell types and/or atlas regions here, similar to the methods used in our examples with whole brain BARseq mapping to assess accuracy. Per the 4th comment of Reviewer 3 specifically to include metrics capturing alignment accuracy along the entire scope of the tissue, we have also addressed this by computing alignment accuracy not only within the area of the primary visual cortex but within the cortical layers across the whole brain. We have done this, for instance, in the example of mapping hemi-brain to hemi-brain where we indicate concordance slice wise from caudal to rostral (Figure 5) as well as in our added supplementary figure 2 with metrics spanning the whole brain and in our added subfigures M-O to Figure 4 with metrics for 2 hemi-brains all measured with BARseq as the examples we mapped for this manuscript. Please see our response to Reviewer 3's 4th comment for further details and the specific changes incorporated into the manuscript.

Specifically regarding quantitative presentation and discussion around intra and inter regional variance, we have expanded on this in the context of presenting our mutual information based feature selection method (Section 2.6). We have specifically contrasted the levels of intra regional variance with regards to the expression pattern of genes over space for genes with high mutual information scores (resulting in lower levels of intra regional variance) versus genes with low mutual information scores. These are depicted visually for some example genes in our revised Figure 7. Additionally, to assess the level of intra versus inter regional variance, we conducted pairwise Mann-Whitney U tests between each pair of regions on individual slices and found after use of bonferroni correction for multiple comparisons that approximately 50% or more of region pairs could be significantly differentiated from one another based on the distribution of gene expression as assessed at $50\mu\text{m}$ resolution compared with fewer than 15% in the case of genes with low mutual information scores. We discuss these results in section 2.6 as well as in the discussion, highlighting the expected extent to which the stationarity assumption holds as well the importance of choosing spatially variable features to underscore this assumption.

Changes made within section 2.6 are copied below as well as the new figure caption for figure 7.

Indeed, we quantified the extent to which the stationarity assumption holds through assessment of intra-regional versus inter-regional variance in the context of genes with high mutual information scores versus low. We built empirical distributions onto $50\mu\text{m}$ resolution CCFv3 slices of particles following deformation (Figure 7M-X). Each empirical distribution pertained to a set of 20 (non-decoy) measured genes taken to be either those with the highest mutual information scores or the lowest. Intra-regional variance was computed for selected genes based on the normalized empirical (probability) distributions per CCFv3 particle (Figure 7O,R,U,X) with larger variance seen amongst lower mutual information gene probabilities (Figure 7O,U) than those of higher mutual information (Figure 7R,X). Consequently, pairwise Mann-Whitney U tests between each region’s distribution of particle gene probabilities revealed many more pairs of regions significantly differentiated based on high mutual information gene probabilities than low, where significance was computed at a level of 0.05 corrected for multiple comparisons with a bonferroni correction (Figure 7Y). In the case of high mutual information genes, 43–68% of region pairs could be significantly distinguished, supporting high inter-regional variance compared to intra-regional variance, supporting our assumption of a stationary probability law. In contrast, low mutual information genes yielding higher intra-regional variance in probability distribution resulted in less than 15% of region pairs significantly distinguished. Hence, this suggests effective selection of features (e.g. cell types or measured genes) particularly in the context of anticipated atlas parcellations governs the extent to which the assumption of stationarity holds, but with methods of selection based on spatial variability, as described here, leading to features typically well aligned with the assumption.

Figure 7 Caption: Selection of gene space in MERFISH measured ≈ 500 genes and illustration of within region and between region variance. A-L show relative expression of genes with high and low mutual information (MI) scores across two MERFISH

tissue sections (A-F, G-L) at coronal levels $Z = 385$ and $Z = 485$ in CCFv3 coordinates, respectively. Genes with highest mutual information scores (A-C, G-I) are *Gfap*, *Trp53i11*, *Wipf3*) and lowest (D-F, J-L) are *Chodl*, *Brs3*, *Hpse2*. M-X exhibit empirical distribution of genes in slices 1 and 2 deformed to corresponding CCFv3 slice. M,P,S, and V show probability of single low MI gene (M,S) or high MI gene (P,V) per atlas particle at $50 \mu\text{m}$ resolution. N,Q,T, and W show average probability of single low MI gene (N,T) or high MI gene (Q,W). O,R,U, and X show variance within each region of probability of single low MI gene (O,U) or high MI gene (R,X). Y indicates the number of pairs of regions with significant differences in probabilities for three genes with high MI and low MI, with significance calculated at 0.05 but corrected for multiple comparisons with bonferroni correction as 0.00003 and 0.00001 based on the number of regions as 60 and 94 in slice 1 and 2 respectively.

Please see our response to comment 5 from the 2nd reviewer for additions made to the discussion including the extent to which we can expect the stationarity assumption to hold and potential areas for future work on feature selection to continue to adhere to this assumption.

The paper concludes that the proposed methodologies enable the matching of 3D partial volumes from single-cell transcriptomics to tissue atlases, and that the approach enables solutions for “peta-scale” datasets. **While I see convincing qualitative and some quantitative support of the first in the experiments, I do not find the aspect of petq-scale data discussed so explicitly in the paper. The discussion and reporting of compute times is not systematic wrt. to scalability with growing data sizes, but rather descriptive in terms of the effect of complexity reduction through the proposed selection mechanisms in physical and feature space. I therefore suggest to explain more explicitly in the introduction and discussion what “peta-scale complexity” implies and how the proposed framework can manage it.**

3. We thank the reviewer for their suggestion to more explicitly demonstrate the scalability of our method and clarify the conclusion of peta-scale complexity being targeted. We have computed the peta-scale complexity of image representations of such data by assuming a representation on a regularized voxel grid of 100s of functional measures (e.g. gene or cell types) measured at a submicron level across sections covering a volume of about $10 \times 10 \times 10 \text{ mm}^3$ (e.g. the volume of a mouse brain used as examples throughout this work). We have made this computation more explicit in the introduction. Additionally, we emphasize that through our image-varifold representation as well as our methods of feature selection and scale-space resampling, we aim to reduce this data complexity, associated to standardized image representations, to a level necessarily more amenable to computation, analysis, and mapping. We have also made this clearer with the inclusion of a table comparing the number of datapoints needed to represent each of our cell-based datasets as a particle representation versus as an image representation.

We have added the following details to the introduction: Covering a volume of about $10 \times 10 \times 10 \text{ mm}^3$, representation of these 6 billion mRNA with even a $1 \mu\text{m}$ voxel grid resolution would thus require 1 trillion (tera-scale) voxels, without even affording

complete differentiation of neighboring mRNA from one another... Consequently, spatial representations necessitating tera-scale complexity are further expanded even to peta-scale complexity with hundreds of possible functional measures taken statically or over time.

We have also added Table 1 in the text with the following description in Section 2.2 and caption:

Importantly, each of these datasets reflects capture at a submicron level, with the minimum distance between two data points (e.g. segmented cells) ranging from 0.015 to 0.5 μm (Table 1). Consequently, representation of these datasets as a regularized voxel grid even at $1\mu\text{m}$ or $10\mu\text{m}$ resolution compromises identifiability of the individual data points, with image-based mapping methods operating on image sizes more typically in the range of $10 - 30\mu\text{m}$ [2]. Additionally, the number of voxels in these representations far outnumbers the data points in the original dataset by $10^3 - 10^6$, thereby increasing the number of distance computations by $10^6 - 10^{12}$ in the setting of quadratic kernel calculations.

Caption: Characteristics of cell-based datasets and comparison of needed data points for particle-based representations versus regularized voxel grid representations.

The paper is of high technical quality and methodological depth, which can also make it difficult for readers who are not so familiar with the principles of diffeomorphic mappings and related physical and mathematical concepts. From my perspective, it would be helpful if Sec. 2 would assume less technical background and include a couple more high-level explanations of the mathematical concepts. For example, Sec. 2.3 assumes solid knowledge of the previous paper on xIV-LDDMM, and would benefit from a couple of brief higher-level explanations of relevant terms such as the Jacobian determinant in (4b), the product operator in (2), and the concept of Hilbert spaces.

4. We thank the reviewer for their suggestion of reiterating some of the notation and concepts introduced in the previous paper for readers in this context. We have included explanations specifically of the terms requested (Jacobian determinant and product operator), but for the purposes of broadening the interpretability of the paper to a wider audience, have chosen to eliminate the specification of a Hilbert space from the early results sections. A Hilbert space is specifically a vector space equipped with an inner product that is used to generate a distance, thereby classifying a Hilbert space as a complete metric space. To emphasize the novelty in our current three technologies posed in this paper, we have deferred a further explanation of Hilbert spaces from the main body of this paper, however, as they have been elaborated on in the manuscripts focused more on the theory behind our image varifold representations, for instance [3]. The changes made to the introduction, discussion, and sections 2.3 and 2.4 for clarity are highlighted below.

Introduction

To solve this problem, we introduce in our first technology a spatial censoring function as part of the imaging model that punctures the **image-varifold** matching norm

within the scope of the deforming atlas thereby optimizing alignment over the censored sample.

Section 2.3:

Physical location and functional feature value measured by a given technology are captured together through the construction of a product measure on the product measurable space yielded by each domain (e.g. $\mathbb{R}^3 \times \mathcal{F}$, with \mathbb{R}^3 the physical domain captured by the technology and \mathcal{F} the domain of functional values measured). Each particle i carries a singular weighted mass located in physical space $w_i \delta_{x_i}$ and a probability distribution p_i over the feature space [1], which may be genes or cell types or tissue types, resulting in the product measure representation

Section 2.4:

The physical mass of the deforming atlas is given by the product $|D\varphi|_{x_i} w_i$ at each location $\varphi(x_i)$ according to the varifold action of diffeomorphisms [4], with the determinant of the Jacobian $|D\varphi|_{x_i}$ capturing the local change in volume at position x_i as a result of the diffeomorphism. This mass is masked according to α^λ at that location to retain atlas regions corresponding to those with measured targets.

Discussion

The optimization based multi-scale resampling we propose benefits from the fact that we place the brains into a Hilbert space (e.g. with an inner product generating a distance metric),...At its core, each particle and feature are compared to each other one in the inner-product that generates the varifold norm.

I found the diagram in Fig 1A quite helpful to understand the high-level scope of the work, but it is in my eyes not as attractive and prominently placed as it deserves to be (especially compared to the good quality of other figures). I suggest to make Fig 1A visually more attractive by exposing the different levels therein (methods, measurements, output & uses) more obviously, e.g. by stronger visual grouping rather than only using font colors. Ideally, the figure would appear closer to Sec 2.1 already, which it supports nicely.

5. We thank the reviewer for their suggestion to emphasize and expand the diagram in Figure 1A. We have separated it out as a new figure (Figure 1) in our revision. We have also adjusted the layout and added details, as suggested, in terms of indicating examples of each of the inputs explicitly in bullet points. Importantly, as per your earlier comment, we have also tried to distinguish with different colors of the modular boxes the technology of “cross-modality mapping” which we introduced in a 2D setting in our previous publication (Stouffer et al, Nat Comms 2024) from the three separate modules of censoring, scale-space resampling, and feature selection that we are emphasizing in this manuscript. These three latter modules we have developed specifically for dealing with challenges introduced in a 3D setting (e.g. more extensive and variable data) to facilitate both single modality and cross modality mapping in 3D across datasets. In our revised figure, we have also illustrated this capability in breaking down the feature transformation and geometric transformation components involved in our cross-modality scheme, which facilitate using these building blocks

either to map single modality captures of different specimens or multiple modality captures of different specimens. As an explicit example, we have also added Figure 5 in the revised manuscript to illustrate use of the censoring technology in the context of single modality mapping to map two hemi-brains of BARseq data to one another, but with different scopes of capture.

The new figure caption reads:

Summary of technologies with designation of input types and examples (left and right), intermediate outputs (red outline) where applicable, and final outputs (green box) resulting from combinations of modular technologies. xIV-LDDMM (blue box) mapping technology as described [1] offers single-modality mapping based on varifold norms without estimation of feature transformation needed for cross-modality mapping. Censoring, scale-space resampling, and feature selection (orange boxes) denote technologies introduced in this work for accommodating complexities and variability in 3D data. Arrows demonstrate integration of new technologies with single or cross-modality mapping applications.

The new added text in Section 2.1 reads:

The specific modules described herein center on advancements including: (i) an explicit model for solving the partial matching problem mapping whole brain tissue scale atlases to partial volume censored targets, (ii) scale-space approximation methods for optimized particle approximation of targets from molecular to tissue scales, and (iii) mutual-information based feature selection technologies for optimized selections of genes/cell types for optimized sparse dimension reduction. As independent modules, we describe each in turn, but with an emphasis both on individual output and collective output (see Figure 1).

The core of our set of technologies is the use of the image-varifold representation to model equivalently tissue-scale imagery and highly resolved molecular scale spatial-omics measurements. These representations are equipped with a norm that defines closeness with regard to feature and physical geometric similarity, departing from the Euclidean metric often used to measure similarity, but which fails to capture inherently the dual spatial and functional nature (compositional) of the data measured by spatial transcriptomics technologies [5]. Consequently, the modules introduced here of censoring, scale-space resampling, and feature selection, seamlessly feed into our established large deformation diffeomorphic metric mapping schemes (xIV-LDDMM) that are image-varifold based and offer single and cross-modality mapping faculties [1] (see Figure 1).

I find it important for the reader to properly understand the motivation and validity of the underlying stationary probability law assumptions of atlas ontologies for the mapping, which drive the alignment processes towards mapping homogenous areas in feature space to delineated brain areas. In the strong multimodal setting addressed here, and depending on the type of reference parcellation, features might show significant intra-region variance wrt. to the atlas ontology and not always fulfill this assumption. One might argue that this is rather an exception to the rule, and that the feature selection strategies will help in identifying features with strong support of the atlas maps, but such discussion is coming rather short in the paper given its relevance

for the topic - most at the start of Sec. 2.8. I would suggest to expand this topic a bit more and motivate it already more explicitly in the introduction of the paper.

6. We thank the reviewer for their suggestion and the opportunity to further motivate and validate the stationarity assumption in this approach. We have supported our stationarity assumption with a reflection of a similar assumption made in different efforts in integrative structural biology and spatial domain identification that stresses the existence of specific patterns of organization and consistency across space at both scales of gene expression, cellular architecture, and even tissue functionality. We have included this in our introduction of the assumption in Section 2.3 with added references ([6–8]).

The assumption of stationarity we make in this setting stems from the simple principle that has been demonstrated in diverse tissues across organisms that cells close in physical distance (e.g. within the same functional region of tissue) share similarity in transcriptional signature [6]. This assumption is further analogous to that used in deep-learning methods aimed at “spatial domain identification”—identification of presumed functionally and structurally homogenous tissue regions with signatures in histological architecture that we expect correspond to transcriptional signatures [5, 7]. Here, we extend this assumption not just from the molecular to cellular scales as evident in the correspondence between transcriptional data and histopathology but also to the tissue scales, as functional atlases and ontologies have often been constructed in conjunction with histopathology. Indeed, this assumption of co-existing and consistent signatures across regions of space at tissue, cellular, and molecular scales is fundamental to diverse efforts within “integrative” structural biology to produce and study multi-scale representations of biology [8].

We have also included details on how selection of features via our mutual information score specifically support the assumption of stationarity in expression across tissue regions. Please see our response to your second comment for further details on how this was added in Figure 7.

Lastly, I should say that I did not feel sufficiently familiar with the mathematical foundations of variation calculus and the statistical properties of genes to fully assess the math in subsections 4.2, 4.4 and 4.5. The derivations and explanations appear plausible and solid to me, but I was not able to validate all derivations therein in detail.

7. We thank the reviewer for their honest comment. To facilitate the the dissemination of our approach to a scope of readers interested in utilizing our methodology and also perhaps interested in further developing and understanding some of the fundamentals, we have included explicit derivations in the methods section rather than in the main results section. To balance understanding across a diverse readership, we believe the changes we’ve made to Figure 1A that helps describe at a high level the motivation and theory of our approach as well as the added text in the introduction and discussion regarding the stationary assumption should help appeal to readers of more diverse biological backgrounds.

While reading, I found a smaller number of typos and inconsistencies which should be addressed in a final version:

1. In Eq. (3), I was partly confused whether λ is used as subscript or exponent in the different places. I assume it is a subscript for μ but an exponent for α - is this correct?

We thank the reviewer for their question and the opportunity to clarify. The function α depends on the parameter λ and takes as its input a location in space. We have removed the parameter λ from its place with μ to make this distinction clearer and instead indicate α operating on the deformed measure $\varphi \cdot \mu$. We have made these changes explicit in section 2.3 as follows.

These weights are given by the censoring function, $\alpha^\lambda : \mathbb{R}^3 \rightarrow [0, 1]$ defined a priori in the coordinate system of the molecular/cellular target, with $\alpha_x^\lambda = 1$ for all x within the support of the target and decreasing smoothly to 0. The brain mapping problem optimizes over the diffeomorphism and the support function, **with λ controlling the smoothness of the support function:**

2. I find the distinction of the terms “mapping” and “alignment” not trivial, since they have slightly different meanings in the communities. Given their central role in the paper (e.g. already in Fig 1.A), I suggest to clarify their interpretation briefly at the beginning of the paper.

We thank the reviewer for their suggestion. In our work, we consider the mapping the process that takes a given atlas to a target with the end result of mapping ultimately being alignment between the atlas and target. We have aimed to clarify this in the introduction with the following edits:

In the Molecular Computational Anatomy model [4, 9], brain mapping, **or relating one brain to another through a similarity metric**, follows the D’Arcy Thompson [10] program, transforming a set of objects that we call varifold measures denoted here as $\mu \in \mathcal{M}$ **(the set of finite positive measures on a given physical and functional space, $\mathbb{R}^3 \times \mathcal{F}$)**, with a norm $\|\mu\|_M$ to measure closeness [4, 9]. Thompson’s brain mapping scheme defines transformations which we model here as diffeomorphisms, $\varphi \in Diff$, transforming one brain to the other with $\mu \rightarrow \varphi \cdot \mu$, **with the end result being an alignment of these objects in the same coordinate system and measurement of their similarity through the magnitude of the mapping (e.g. diffeomorphism) needed to transform one to the other.**

3. In l. 110, it is at first unclear what M is. I suggest to introduce more explicitly.

We thank the reviewer for their comment. M is the set of finite positive measures on a given physical space and feature space. We made this more explicit in the introduction with: ...follows the D’Arcy Thompson [10] program, transforming a set of objects that we call varifold measures denoted here as $\mu \in \mathcal{M}$ **(the set of finite positive measures on a given physical and functional space, $\mathbb{R}^3 \times \mathcal{F}$)**, with a norm $\|\mu\|_M$ to measure closeness

4. In l. 112, “Diff” not properly formatted

We thank the reviewer for their comment. We have reformatted the diffeomorphism group as: *Diff*

5. The determinant of Jacobian $-D_\phi-$ in (4b) should be briefly explained in the text

We thank the reviewer for their comment. We have explained the Jacobian determinant as capturing the local change in volume and added this to the text in section 2.3:

The physical mass of the deforming atlas is given by the product $|D\varphi|_{x_i}w_i$ at each location $\varphi(x_i)$ according to the varifold action of diffeomorphisms [4], with the determinant of the Jacobian $|D\varphi|_{x_i}$ capturing the local change in volume at position x_i as a result of the diffeomorphism. This mass is masked according to α^λ at that location to retain atlas regions corresponding to those with measured targets.

6. l. 339, typo: “similtude”

We have corrected this typo.

7. l. 525, word missing?

We have corrected this typo by removing two extra words.

8. l. 591, should that be “totalling”?

We have made this correction.

9. l. 645 should be “Kullback Leibler”

We have adjusted the typo.

10. L 385, grammar? “E,F show single intersecting section of deformed”

We have adjusted this as **intersecting sections**.

11. Refs 11 and 14 incomplete or wrongly formatted

We have reformatted the reference 11 and removed the reference 14 as it was superfluous and as suggested, wrongly formatted.

2 Reviewer 2

Summary:

This paper introduces xIV-LDDMM, a computational toolkit for integrating spatial-omics and imaging data across scales. The framework leverages image-varifold norms for cross-modality mapping and implements multi-scale optimization and mutual-information-based feature selection to align molecular and tissue-scale datasets. The authors demonstrate its utility on mouse brain datasets from MERFISH and BARseq technologies, showcasing robust alignment of partial and whole-brain data with the Allen CCFv3 atlas.

Review Points:

1. Limited Generalizability Assessment: The toolkit is evaluated primarily on mouse brain datasets, limiting insights if the methods works equally well on other organisms or technologies, e.g., the authors could consider additional datasets like SlideSeq or on larger human/primate datasets to substantiate its broader utility.

We thank the reviewer for their suggestion to include additional datasets with broader utility. As such we have added examples of mapping one hemi-brain

to another (as an instance of our single modality platform) and an example of mapping a whole mouse embryo to a corresponding atlas. This embryo was measured with a novel technology called cycleHCR that has the distinct power of measuring thicker tissue sections and thus generating definitively 3D data. Consequently, it is particularly apt for mapping by our programs that are tailored to handle such 3D data. We have described this dataset and the whole mouse embryo used from the EMAP project based on Kaufman’s original annotations in section 2.2 of the revised manuscript as follows:

The results described herein are from four different spatial transcriptomic datasets covering three different technologies. The first is a published MERFISH dataset [11] with 56 coronal sections that cover most of the mouse brain except for the most rostral and caudal part. The second and third were measured with BARseq and comprise ...The fourth dataset is a published measurement of transcripts of 254 genes in a whole mouse embryo aged E6.5-7.0 with cycleHCR technology [12], covering the entire volume at a depth of 310 μm Similarly, the 254 selected genes measured by cycleHCR in the whole mouse embryo were lineage-specific, with a goal of identifying differentiating structures early within embryogenesis and primarily intra rather than extra-embryonic.

Our results have focused on demonstrating alignment to atlas coordinates using both cell type labels and gene expression directly. In the brain datasets, ... Finally, in the whole mouse embryo, Cellpose was used together with UMAP to identify and cluster cells into a set of 9 distinct types [12], which we map to the ts09 EMAP atlas [13], based on ontological annotations from Kaufman [14].

We have explored these results specifically in the added section in the manuscript of 2.9. Key points to highlight in this setting are effective mapping achieved between an atlas and dataset that differs from the prior examples in its being (1) entirely 3D, (2) generated with a different type of spatial transcriptomics technology, and (3) measuring a whole mouse embryo as an entirely separate entity from the exclusive brain work we have focused on thus far. An additional complexity introduced with this example is that of varying levels of geometric differences between atlas and target, which is inherent to datasets spanning development in which geometry of organs and tissues varies more diversely over both time and space than datasets limited to adult morphology. Indeed, we demonstrate our methodology’s handling of atlases of two different time points of mouse embryo development to illustrate both accuracy and limitations of alignment in these settings of some level of expected mismatch. We further expand on this example in our discussion with suggestions for future work involving the development of longitudinal image-varifold based LDDMM modules, based on our prior longitudinal image-based LDDMM methods for estimating correspondingly not only alignment between atlas and target, but optimal placement of target within the continuum of atlas timepoints.

These results and discussion are included below with figure captions for both the main text figure and supplementary figure added as part of this cycleHCR embryo example.

In both MERFISH and BARseq captures, volumes were interrogated on a slice-by-slice basis with tissue slices spaced $100\ \mu\text{m}$ or $200\ \mu\text{m}$ apart, respectively. Increasingly, newer technologies are emerging to accommodate single three-dimensional captures either directly or as part of a post-processing assembly built into the technology, itself [15]. Consequently, there is a need for mapping platforms, as that presented here, to accommodate datasets captured as single blocks, multiple tissue blocks, or multiple sections in an effort to integrate data across technologies and within the framework of common coordinate systems and standardized ontological atlases.

Additionally, as a result of these 3D captures, tissue types and shapes of volumes measured have continued to diversify with the capture of in tact specimens, not easily sectioned. One example is that of the whole mouse embryo at stage E6.5-7.0 measured with the novel deep-tissue spatial transcriptomics method cycleHCR [12]. To demonstrate the generalizability of our mapping technologies to one such deep-tissue capture and to specimens beyond those of the brain, we mapped time points of the corresponding EMAP atlas [13], developed from Kaufman ontological annotations [14] to this whole mouse embryo.

Notably, in the context of developing anatomy in contrast to adult anatomy, atlases are often comprised of select time points spanning the evolution of the organism, with individuality in the exact time course observed per specimen. Consequently, this creates even further challenges to alignment of atlases to targets where atlases often represent averaging over both space and time, which is rapidly evolving. Here, the two closest time points within the atlas series to stage E6.5-7.0 of the cycleHCR embryo are the ts09 time point at E6.0-6.5 and the ts10 time point at E7.0 (Figure 10B and K). We mapped each of these atlases (initially designated at a resolution of $2 \times 2 \times 2\ \mu\text{m}$) as $8 \times 8 \times 8\ \mu\text{m}$ particle representations to a cell-based particle representation of the cycleHCR embryo, containing 11,029 cells (Figure 10A). We assessed the alignment qualitatively through observation of concordant areas overlapping, such as the areas of the primitive streak, mesoderm, and parietal endoderm in each atlas with cells typed similarly as constituting those regions (black and white arrows in Figure 10F-J,H-S). Quantitatively, we assessed alignment by computing the minimum distance of each cell of the primitive streak and parietal endoderm to its nearest atlas neighbor within the same corresponding region, with an observed reduction in the average distance 4.5 fold before versus after deformation (Figure 10C).

Interestingly, the geometry exhibited in the cycleHCR embryo appeared to be a hybrid of those prescribed by the ts09 and ts10 atlases. To compare the alignment between the the cycleHCR embryo and each corresponding atlas, we again considered the regions where cell type corresponded directly to region (e.g. primitive streak and parietal endoderm). Here, the minimum distances achieved following alignment to the ts09 atlas within the primitive streak and parietal endoderm were 0.016 ± 0.019 and 0.016 ± 0.016 mm, respectively. In contrast, the minimum distances were greater in the case of the ts10 atlas, 0.031 ± 0.024 and 0.057 ± 0.07 mm (Supplementary Figure 3), respectively, suggesting poorer similarity in geometry

resulting in larger distance discrepancies between the regions. Nevertheless, cross-sectional views of the cycleHCR embryo and each of the atlases revealed greater visual concordance between the aligning geometry of the primitive streak (blue) and mesoderm (yellow) between the cycleHCR embryo and the ts10 atlas compared with the ts09 atlas (Figure 10R versus Figure 10J). In contrast, the ts10 atlas exhibited further differentiation of the extraembryonic tissues into the amniotic fold, not yet present within the cycleHCR embryo (Figure 10Q-S, dotted arrows). However, the cells typed as precursors of the amniotic fold including those of the extraembryonic mesoderm and ectoderm corresponded locationally to the area of the amniotic fold in the ts10 atlas, suggesting further concordance of alignment in the setting of anticipated differentiation of these cells. Finally, with both the ts09 atlas and the ts10 atlas, the measured cycleHCR embryo comparably represented only a subset of the entire volume, similar to the partial volumes captured within the whole brains measured with BARseq and MERFISH. In both cases, the majority of the polar trophoderm, not imaged with the cycleHCR technology aligned to outside the support of the target, as expected.

Figure Caption: Mapping of EMAP ts09 and ts10 mouse embryo atlases ($8 \times 8 \times 8 \mu\text{m}$ resolution) to cycleHCR whole mouse embryo cells. A shows cell centers of measured embryo with prescribed cell type based on UMAP clustering [12]. B and K show ts09 and ts10 atlases with 9 and 14 regions, respectively. D and L show initial alignment of each atlas to target embryo (black dots) versus E and M show alignment following estimation of diffeomorphism with xIV-LDDMM. F and G show vertical cross-section through alignment of ts09 atlas to target before and after deformation, with black arrows indicating area of primitive streak in target embryo and white arrows indicating primitive streak in atlas. H-J show horizontal cross-section of ts09 atlas before and after deformation (H, J) and of target embryo (I) with arrows indicating primitive streak and proamniotic/amniotic cavity. C shows distance of each embryo cell with primitive streak and parietal endoderm type to corresponding type in ts09 atlas before and after alignment. N-P show horizontal cross-section of ts10 atlas before and after deformation (N,R) versus that of target embryo (P) with arrows indicating area of mesoderm. O-S show vertical cross-section of ts10 atlas before and after deformation (O,S) versus target embryo (Q) with solid arrows indicating mesoderm and dotted arrows highlighting area of developing amniotic fold.

Supplementary Figure Caption: Distance from each cell center in the cycleHCR embryo to nearest particle in corresponding region within the ts10 atlas. Regions include the primitive streak and parietal endoderm, with distances calculated between embryo and atlas before and after deformation.

We have added our discussion of a longitudinal image-varifold based LDDMM module to our discussion, as commented on in our response to your 5th comment. Please see there for changes to the discussion.

2. Inadequate Quantitative Benchmarks: While the paper provides qualitative examples and runtime analysis, it lacks comprehensive quantitative comparisons with alternative mapping methods such as SpatialPCA, BayesSpace, or other generative

approaches. Established metrics for spatial-omics alignment (e.g., mutual information, accuracy of boundary delineations) should be incorporated to strengthen benchmarking claims.

We thank the reviewer for their suggestion to include more quantitative benchmarks and comparisons to other methods. As indicated in our response to your comment below, we have included some discussion of the differences between many of the deep-learning methods that exist to analyze and align spatial transcriptomic data (often to histopathological data) with our method specifically aimed to align and thereby integrate data at the tissue scale with that at the cellular/molecular scales.

With regard to the suggestion to incorporate more quantitative measures of accuracy, we have added such measures in many of the added experiments/figures that we have added in this revision. For instance, we have added a mapping of a hemi-brain to another hemi-brain to underscore another application of the censoring module in a setting in which a single modality is used, generating the same feature space, but in which the scope of tissue capture slightly varies between the two specimens. The addition of this example not only addresses your suggestion above for illustrating additional use cases / applications of our method to underscore generalizability, but it has allowed us to quantify accuracy as a concordance in the position of predominant cell types across the hemi-brains before and after alignment. We do so by selecting the target and atlas predominant cell type within $100 \mu\text{m}^3$ cubes within target sections and computing the percent of cubes with matching predominant cell type between them. These results have been added to Section 2.4 and with the addition of Figure 5. The specific text and figure caption is given below.

Finally, as indicated in Figure 1, the censoring technology described here is applicable not only to the alignment of datasets stemming from different modalities, as often result in different scopes of capture, but also to the alignment of datasets from a single modality, capturing different scopes of tissue in different specimens. As evidence of this application, we mapped the two sets of control mouse hemi-brain sections demonstrated in Figure 4 to one another (Figure 5), with the template (atlas) set containing 32 individual sections and the target set containing 31 (without the seventh most caudal section). Though the sections were taken at approximately every $200 \mu\text{m}$ along the rostral-caudal axis, resulting in a similar scope of tissue capture between the two brains, the atlas set extended more rostrally and medially than the target set, as evidenced by their corresponding alignment when mapped to the CCFv3 (Figure 5A). We initially aligned the hemi-brains slice-wise (Figure 5B), resulting in mismatched anatomical regions, including areas of the hippocampus such as CA3 and subiculum (Figure 5D-E). With the censoring function, minimization of the varifold normed difference of the hemi-brains was achieved over the domain of the target hemi-brain, yielding an optimized diffeomorphic mapping of atlas to target hemi-brain in which concordance between these internal hippocampal regions and between the outer layers was better achieved (Figure 5D-F).

We measured this concordance before and after alignment at a resolution of $100\ \mu\text{m}$ by comparing the predominant atlas and target cell types within cubes of $100\ \mu\text{m}^3$ across the domain of every fifth target section (Figure 5G-H). Near 80% concordance was achieved specifically within the superficial layers of target cortex (2/3, 4/5) after diffeomorphic alignment compared with 20–30% as initially aligned slicewise. Additionally, we see slightly greater accuracy achieved within the rostral half of the slices (e.g. 15-25) compared with the caudal and with little loss in accuracy ($\approx 10\%$) towards the rostral and caudal poles of the dataset (e.g. slice 5 and 30).

Figure Caption:

Censoring technology for use in mapping sets of hemi-brain BARseq sections from two separate mice to one another. A shows each set of hemi-brain sections mapped to CCFv3 for comparison of scope of tissue capture. B shows initial alignment of hemi-brain tissue sections. C shows alignment of tissue sections following estimation of diffeomorphism with censoring to take atlas hemi-brain to target hemi-brain. D shows single target slice with cell type denoted by color. E and F show intersecting atlas slice before (E) and after (F) alignment to target hemi-brain sections, with white arrows indicating regions of hippocampal (Subiculum in yellow, CA3 in orange) alignment post deformation. G-H show the fraction of $100\ \mu\text{m}^3$ cubes across the domain of the selected target sections with matching predominant cell type between atlas and target hemi-brain globally across the whole section (G) and layer-specific (H). Predominant cell type per cube is shown for target sections (I,L), intersecting atlas section as initially aligned (J,M) and after diffeomorphic alignment (K,N).

We have also added quantitative measures of accuracy in our added example of mapping EMAP mouse embryo atlases to a cycleHCR embryo. These measures include computation of the distance between each cell of a given type to the nearest data point within the corresponding region in the atlas. We choose to measure these distances for cells of the primitive streak and parietal endoderm as two areas specifically where there is correspondence in designated region and cell type. We use these quantitative measures as an evaluation of alignment accuracy and as a way to compare alignment of the cycleHCR embryo to the ts09 versus ts10 atlas timepoints. These measures are included in the text in section 2.9 as described in our response to your comment 1.

Finally, as per reviewer 3’s comment number 4 below, we have measured accuracy across the whole brain by looking at cortical layer alignment not only within the primary visual area but across the whole rostral-caudal span. We have summarized these metrics in the added supplementary figure 2 for mapping a whole brain and in the added subfigures (M-O) to figure 4 for mapping two hemi-brains (all measured with BARseq) to the CCFv3. Additionally, these measures of concordance within the cortical layers were compared with manual alignment procedures (Supplementary table 3) as a baseline. Please see our full discussion and specific changes made within our response to Reviewer 3’s 4th comment below.

3. Insufficient Literature Context: The comparison focuses on older methods (e.g., K-means, grid resampling) but omits more recent, competitive approaches. The paper does not discuss advanced deep learning-based models, which could serve as relevant benchmarks for multi-scale mapping.

We thank the reviewer for their assessment and encouragement to expand our comparison of parts of our methods to deep learning-based approaches. While we have found many of the reported deep-learning methods to attempt many useful analyses, we have not found a deep-learning method which specifically aims to align tissue scale atlases to spatial transcriptomic data. Consequently, we have reviewed some of the key differences in data representation and similarity metric (our approach based on varifold norms rather than a Euclidean metric and therefore remaining often in the native feature space) as well as our handling of datasets in a whole volume rather than via patches. We have added a few references regarding deep-learning methods aimed at analyzing spatial transcriptomics data as well. These changes are found mostly in the introduction as follows.

Deep learning methods that might accommodate such data complexity have naturally arisen to analyze spatial transcriptomic data both independently (e.g. for cell typing) or in the context of histological images taken of the same tissue sample [5]. However, many of these methods have continued to suffer from data complexity, necessarily breaking up 2D datasets into smaller patches that then fail to achieve consistency across their boundaries with regard to end analysis [16]. Additionally, with regard to data representation, many of these methods fail to offer a clear biological interpretation as they transform the inherently compositional data from its native space [17] to one that can be treated using Euclidean distance metrics [5].

...Importantly, this approach for measuring similarity departs from most deep-learning based methods rooted in Euclidean space and offers greater diversity of input “images” not just from a single tissue specimen [5]. As measurements and atlasing occurs across many scales, the formalism naturally supports hierarchical representations from tissue to molecular scales [4, 9]. Our descriptions throughout carry appropriate spatial scales associated to the smoothing kernels which produce a sequence of successive approximations of greater and greater detail in the dimensions of space-scale and gene/cell feature. Furthermore, we accomplish this without the need for subsetting data into discrete patches separately to analyze and therefore, avoiding the discrepancies in analysis that have resulted from deep-learning methods analyzing data in batches [5]. We exploit this multi-scale representation demonstrating mappings between the sub-micron scales of the transcriptome to the millimeter scales of the CCFv3.

4. Scalability Challenges: The method exhibits significant computational overhead, requiring up to 24 hours per mapping iteration. Although GPU parallelization is used, the discussion of scalability for larger datasets remains speculative. Strategies to reduce computational costs, such as hybrid multi-scale approaches, could enhance practicality.

We thank the reviewer for their comment. We are currently working to improve the scalability of our methods even further by the development of such

hybrid approaches whereby we will compute mappings at different scales in an approach similar to that we have previously described ([9]). We have emphasized this as an area of future work, particularly how it compares to the current approach, and also per your comment below. Please see our updated discussion of this in our response below.

5. Limited Discussion of Limitations: The paper provides little discussion on potential constraints, such as failure modes in complex datasets, sensitivity to input data quality, and computational scalability. A deeper examination of these factors would provide valuable context for future applications.

We thank the reviewer for their comment. As per Reviewer 3’s final comment (#5 below), we have added an example of one failure mode being the use of disjoint data in which we may have drop out of particular sections throughout a dataset due to poor data capture. We have shown that our mapping technology handles this situation without error in finding nearly identical mappings between atlas and target when all of the data is available. We have specifically added this result as a Supplementary Figure and discussed it in section 2.4, as described in our response to reviewer 3 below.

Additionally, in our addition of a mapping of an embryo atlas to cycle-HCR dataset, we have discussed the feasibility and limitations involved in mapping atlases with expected differences from molecular datasets in terms of geometry in the setting of development. We have included these details in the added section 2.9 (described in detail in our response to your first comment) as well as in the discussion with an area of future work being the incorporation of an image-varifold based longitudinal LDDMM framework into our methods based on our previous work with image-based longitudinal LDDMM for estimating insertion points of atlas/target along a continuum.

Additionally, as addressed in a number of our responses regarding the motivation behind our assumption of stationarity in probability law, we have added an area for future work being a tailoring of our mutual information selection procedure specifically to enhance alignment to this assumption, but in a manner necessarily application specific and therefore compromising some of the generalizability we opted for here.

Updates to our discussion with these more detailed assessments of strengths and limitations are copied below:

Strengths of the approach presented here first include the efficiency of the image-varifold particle representation for 3D data sampled sparsely within a volume. Comparable regularized image grids require orders of magnitudes more data points (e.g. $10^3 - 10^6$) even without full resolution of adjacent measured data, and resulting in orders of magnitude more computations, which in the setting of distance metrics, are often quadratic in the number of data points. Second, the image-varifold representation coupled with the specific censoring method enhances the generalizability of our suite of technologies to accommodate mapping data with different scopes of capture and sampling schemes (e.g. spot-technologies versus single-cell measurement schemes) and to handle situations of drop out or poor data quality (e.g.

disjoint and missing slices, as demonstrated in Supplementary Figure ??). Indeed, in the advent of technologies with even deeper tissue measurements leading to wholly 3D captures (e.g. cycleHCR), our methodologies can easily adapt to settings of 2D sections versus these volumetric captures with further ability to integrate them into a single common coordinate framework. In contrast, many current schemes [18, 19] are built to accommodate 2D sections which they treat independently, necessarily requiring a priori selection of the 3D atlas section to which to align each molecular/cellular section. This is challenging in typical settings in which tissue is not necessarily sectioned perpendicular to a given axis. Finally, a third strength of our approach is in our joint modeling of physical and feature space, as measured directly by the given technology and therefore affording easy biological interpretability of distance measures. While we introduce two reductionist schemes (e.g. scale-space resampling and mutual information-based feature selection), the reduced data representation still maintains the same inherent physical space and feature type, unlike many of the machine-learning based approaches which wholly transform data to a space often uninterpretable with regard to the biological underpinnings [5]).

Amongst these strengths, the generalizability of our methods together with the following limitations offer areas for future work. With regard to generalizability, first, we have exhibited one axis of generalizability with the demonstration of our method for mapping molecular and cellular datasets from a range of technologies (MERFISH, BARseq, cycleHCR) measuring both mouse brains and mouse embryos. While the focus of this manuscript has been on the description and evaluation of the data reduction and mapping schemes associated to xIV-LDDMM, one avenue of future work includes further exhibition of the generalizability of the approach to mapping molecular and cellular datasets from additional technologies (e.g. SlideSeq, synapse imaging [20]) with increasing variability and size as well as changing anatomy over time. Indeed, as evidenced in our mapping two separate time points of the EMAP embryo atlas to the whole mouse embryo measured with cycleHCR, the opportunity and need naturally presents itself for extension of the cross-modality image varifold mapping framework to incorporate longitudinal faculties involving the estimation of an optimal entry point of target data into a potential continuum of atlas time points. Similar longitudinal image-based LDDMM methods have been established with success at measuring shape changes overtime [21–23] and therefore provide the basis for development of such an image-varifold based pipeline in the future, as a second area of future work. Lastly, as described in [1], estimation of a diffeomorphic transformation taking tissue-scale atlas to molecular or cellular-scale target affords the pulling back of each target into the atlas common coordinate framework via the inverse transformation. This consequently enables integration of measures across replicates and technologies for comparison of feature distributions across them. Hence, the continued development of post processing analysis techniques and biology-based interrogation of integrated and aligned datasets is a third avenue of future work.

Regarding feature modeling and selection, there are opportunities for future work both in refining the mutual information-based subsampling procedure and in expanding the feature spaces modeled per image-varifold object. First, we have

described a greedy feature selection procedure looking at genes (or cell types) independently to select an informative subset. It is natural to study such selection in the context of non-independence, examining pairs and triples, with many cell types or genes distributed similarly with functional associations. This is something we are currently pursuing. **Second, our cross-modality mapping method is rooted in an assumption of stationarity with respect to the distribution over features within the space of each atlas region. While the mutual information method as presented here selects features that typically enable significant differentiation between at least 50% of regions, the use of more tailored boundaries specific to atlas choice is a potential area for future investigation affording closer alignment with this assumption of stationarity. Here, instead, we have opted to consider more general horizontal and vertical planes as more tailored boundaries would inherently be more limiting in being application specific.** Finally, though each dataset here was represented as an image varifold over a single physical and feature space, the coupled modeling of different feature spaces (e.g. genes and cell types) over which a single image varifold is defined naturally addresses the incorporation becoming prevalent at the stage of data collection. For instance, cell-based datasets, such as the BARseq example analyzed here, often carry both gene and cell-level information based on integration of raw measurements of mRNA with segmentation and clustering schemes (e.g. histological stains, clustering). The image-varifold representation lends itself to associating to physical spatial measures not just single feature measures but potentially feature measures over differing feature spaces (e.g. genes, cell types) that could simultaneously be used for mapping between tissue-scale atlases and these molecular/cellular scale datasets.

Lastly, a significant challenge of the xIV-LDDMM algorithm is the peta scale nature of the quadratic computation. **At its core, each particle and feature are compared to each other one in the inner-product that generates the varifold norm.** This computational complexity is illustrated by the run times on the order of 24 hours for computing single mappings (see Table 2). Our strategy thus far has been to exploit the parallelism of GPUs. The multi-scale resampling scheme described in Section 2.5 naturally lends itself to the multi-scale mapping scheme initially introduced in [9], in which successive refinements can be made at coarse to fine scales of transformations **estimated with corresponding coarse to fine approximations of data. Currently, multiple scales of transformations are estimated simultaneously with a single set of particles, thereby limited in coarseness and sparsity to the finest scale of desired transformation.** In the multi-scale approach previously described [9], transformations are estimated first at a coarser scale with a coarser representation of data (e.g. fewer particles) needed to capture our atlas and target objects. Successive addition of finer scale representations for refining these coarse scale mappings can then be achieved with likely fewer iterations, reducing the overall computational load these mapping schemes encompass. We are currently working to adapt the scheme described here to a multi-scale setting for improved efficiency in the future and with the prospect also of accommodating different feature spaces over which particle representations might be defined at each scale.

Recommendation:

The manuscript is a promising contribution to the field. In addition, I recommend the authors to:

1. Test the method on additional datasets from other species or spatial-omics technologies.
2. Incorporate established quantitative metrics for comparison with state-of-the-art methods.
3. Expand benchmarks to include deep learning-based approaches.
4. Optimize computational efficiency and assess scalability.
5. Provide a thorough discussion of limitations and failure modes.

We thank you for this summary of what is contained in your detailed comments above. Please see the above responses to these points correspondingly.

3 Reviewer 3

[Summary] Image-based technologies (e.g., BARseq, MERFISH) sample irregularly across tissue, making it crucial to integrate and align data with reference coordinate systems. Thus, the authors proposed a diffeomorphic mapping suite for 3D image-transcriptome registration. This method extended existing 2D image-transcriptome registration method, xIV-LDDMM, into 3D image voxel space and provided a comprehensive suite for its application by adding modules that mapping whole tissue volume to subvolume, multi-scale resampling, feature selection. The suite consists of four modules. The first module enables matching partial to whole tissue volumes with censoring, second module conducts cross-modality mapping based on an image-varifold measure norm, third module scale-space optimization for multi-scale resampling, and the fourth module enables mutual information-based feature (i.e. gene, cell) selection. The authors mapped the molecular varifold of MERFISH and BARseq dataset into the reference 3D image template of mouse brain, CCFv3, and demonstrated that this suite can provide correct transformation and alignment of CCFv3 to MERFISH/BARseq varifold. By comparing untime/memory efficiency across multiple resolutions, authors also demonstrated that this suite is efficient to represent extremely large-scale images (peta-scale) and include algorithms that enable mapping across scales, from nanometers to millimeters.

[Comments] [Originality And Significance] - Authors emphasise on the efficacy of 3D mapping and the usability of suite rather than emphasizing novelty. Nonetheless, the evaluation over runtime/memory efficiency hardly backs up the significance of efficiency-the comparison is conducted without baseline.

1. We thank the reviewer for their comment and that below which relates to comparison to a baseline. Notably, as indicated below and in our third response to Reviewer 1, one of the axes of efficiency we have highlighted is in our representation of the data as sparse particles rather than within the framework of a regularized grid. LDDMM in the setting of image varifolds, as used here, and in the setting of images, necessarily carries quadratic computations in the number of deforming datapoints. Therefore, reduction in these computations by orders of magnitude (e.g. 10^3 or 10^6) captures improvements in efficiency both with respect to memory and

time. In comparison, image-based LDDMM methods such as STalign (cited below) operate on these regularized grids and have been demonstrated at these coarser scales of $10 - 30\mu\text{m}$. Here, we underscore this efficiency with respect to the data size rather than measurement of runtime directly given the large differences in computational resources available to end users, which will inherently lead to differences in exact runtime. Furthermore, the increasing prevalence of shared resources amongst groups/centers inhibits direct comparison of runtime between experiments with varying number of users and processes often occurring simultaneously.

- Since the authors emphasize that efficient peta-scale mapping is possible with this suite and compared runtime/memory efficiency across multiple resolutions, it would be valuable to provide qualitative/quantitative measures of efficiency compared to baseline methods. For instance, comparisons with diffeomorphic-based baseline methods, such as the mesh-based diffeomorphic mapping in esh-based diffeomorphic mapping in [2] and another LDDMM-based spatial dataset mapping method. [1] Clifton, Kalen, et al. "STalign: Alignment of spatial transcriptomics data using diffeomorphic metric mapping." *Nature communications* 14.1 (2023): 8123. [2] Stouffer, Kaitlin M., et al. "Cross-modality mapping using image varifolds to align tissue-scale atlases to molecular-scale measures with application to 2D brain sections." *Nature Communications* 15.1 (2024): 3530.

2. We thank the reviewer for their suggestion. Notably, we have compared the efficacy of mapping using LDDMM-based procedures with images to particles in the second paper cited above, showing the dependence of images on foreground background and the lack of specificity images have for depicting the particle-based datasets. As described in our response to one of the earlier comments by Reviewer 1, we have also articulated more clearly how the technologies introduced in this manuscript (censoring, scale-space resampling, and feature selection) facilitate the application of our xIV-LDDMM technology, as described in [2] above, in 3D settings. Emphasis on our use of the same cross-modality mapping scheme as described in [2] is found in our revised Figure 1, changes in the introduction, and changes to section 2.1. We have indicated these changes specifically in our 1st and 5th responses to Reviewer 1's comments.

We have also separately elaborated on how we arrive at peta-scale complexity of the data as conceiving of the data represented most commonly in a regularized grid, or image formulation. These changes are described in our 3rd response to Reviewer 1 and include the addition of Table 1 in the manuscript in section 2.2.

[Clarity] - The mapping from 3D images to 2D images assumes stationarity in the probability law of molecular/cellular features over each atlas region, making the mapping dimensionality-agnostic. However, gene expression measured from CCFv3 and MERFISH/BARseq may differ in gene panels and techniques. Is this assumption valid? It would be helpful to provide evidence supporting this assumption.

3. We thank the reviewer for their comment and invitation to expand on the stationarity assumption, as also suggested by Reviewer 1. Please see our 6th response

to Reviewer 1 for indications on the discussion we added regarding the similarity in this assumption to those made generally in integrative structural biology. As stated there, we have also highlighted the validity of this assumption in our own examples by considering inter versus intra regional variance of the probability distributions of high and low mutual information scoring genes. Please see our response 2 to reviewer 1 for the details on how we have included a quantitative assessment of the level to which the stationarity assumption holds using pairwise Mann-Whitney U tests to assess significant differences amongst pairs of atlas regions with regard to specific gene distributions. In summary, we acknowledge that this assumption varies based on gene selection, with much greater power to differentiate regions based on spatially variable genes selected through processes such as our mutual information method presented here than genes exhibiting lower spatial variability. Nevertheless, even with approximately 50% of region pairs significantly differentiated, this enables effective alignment to be achieved, as exhibited in the different cross-modality examples included within this manuscript.

[Quality] - Figures except figure 5 provides qualitative measure, the quality of alignment for few selected sections. Qualitative measures that summarize all the alignments would back up the alignment performance of the xIV-LDDMM toolkit.

4. We thank the reviewer for their suggestion. As indicated in some of our other responses including our responses to reviewer 2's 1st and 2nd comment, we have included quantitative metrics of accuracy looking at distances between concordant feature types (e.g. regions in atlases versus cell types in molecular datasets) in the examples added to the revised manuscript including a mapping between one hemi-brain to another and between a whole mouse embryo atlas to a cell-based embryo dataset measured with cycleHCR technology.

Specifically with regard to your comment about incorporating measures that cover larger volumes (e.g. accuracy across the whole brain), the two examples highlighted above include measures of accuracy across larger volumes. In computing accuracy for embryo alignment, for instance, we looked at distances of parietal endoderm cells between the cycleHCR embryo and deformed atlas, where the parietal endoderm spanned much of the support of the cycleHCR embryo both vertically and horizontally. Please see our response to reviewer 2's first comment as well as the added section 2.9 for details. Additionally, we have aimed to show accuracy along a rostral-caudal axis in mapping one hemi-brain to another by comparing concordance of cortical layers slice-wise and illustrating how the accuracy varied across the slices from rostral to caudal extent with greater accuracy seen towards the rostral end of the data, but overall with little loss of accuracy even towards the very rostral and caudal poles (e.g. $\approx 10\%$). We have included these results in section 2.4 in the text with the addition of Figure 7. Please see our detailed response to Reviewer 2's second comment for the specific changes made.

With regard to the examples already contained within the manuscript, we have added metrics of accuracy that span the entirety of the dataset (e.g. all the slices) by

computing the fraction of cells of a given cortical layer (e.g. layer 2/3 versus 4 versus 5 versus 6) that correctly align to the corresponding layer within the CCFv3. We specifically compare these fractions to what was achieved with manual alignment on a slice-by-slice basis, which we have included in Supplementary Table 3 and as part of our added subfigures to Figure 4 (namely M,N, and O). We highlight the relative deficit in accuracy in layer 4 as likely due to cells characterized only as layer 4/5 which we assess for alignment to layer 4 of the CCFv4 versus our achieving greater accuracy than manual alignment particularly within the deeper layers of cortex (5-6). We also illustrate how these accuracy measures vary with respect to the caudal-rostral axis of the brain with relatively little change observed in accuracy across this axis, emphasizing the stability of our mapping procedure. We have included this quantitative measures both in examples of BARseq hemi-brains mapped to the CCFv3 (Section 2.4, Figure 4) as well as in the example of a BARseq whole brain mapped to the CCFv3 (Section 2.7 and Supplementary Figure 2).

Changes to Section 2.4: We quantified alignment of designated cell types within corresponding cortical layers of the CCFv3 atlas, yielding comparable percentages of accuracy to what was achieved through manual alignment of each slice individually in seven separate hemi-brains to the CCFv3 atlas (Figure 4O and Supplementary Table 3). Notably, errors in concordance arise both due to errors in cell typing and in alignment. Additionally, layer 4 typed cells were specifically categorized as layer 4-5 cells and thereby, lower fractions of such cells appeared within the specifically designated layer 4 areas of the CCFv3 (Figure 4M,N). In contrast, alignment with xIV-LDDMM achieved greater concordance between CCFv3 regions and cell types in the deeper layers (5 and 6) than manual alignment (Figure 4O). Furthermore, while fractions of correctly aligned cells within layer 4 decreased with proximity to the caudal and rostral poles of the data, we observed relative stability in fractions of correctly aligned cells across the other layers along the caudal-rostral axis (Figure 4M,N).

Figure 4 Caption: Results for mapping CCFv3 to sets of hemi-brain BARseq sections, aggregated into $200\mu\text{m}$ representations for two separate mice. A,B show initial alignment of CCFv3 to BARseq sections (black dots) for mouse 1 (A) and mouse 2 (B). C,D show physically transformed CCFv3 to coordinates of target BARseq sections, with determinant of the Jacobian showing areas of expansion (red) and contraction (blue). E,F show intersecting sections of deformed CCFv3 (white arrow) overlaid with corresponding BARseq section (black dots) from each hemi-brain. G-L show intersecting sections of deformed CCFv3 with ontology regions in grayscale and cell type delineations for BARseq sections overlaid in color, with G,H taken at white arrow and I-L more posteriorly (yellow bracket in C,D). M,N show fraction of correctly aligned cells within each hemi-brain per each of four layer delineations (L2/3, L4, L5, L6) to corresponding CCFv3 region within $500\mu\text{m}$ intervals along caudal-rostral axis. O shows average fraction for 7 hemi-brains of correctly aligned cells with manual alignment versus fractions achieved with xIV-LDDMM for the two hemi-brains shown.

Changes to Section 2.7: Notably, the percentages of correctly aligned cells within especially the deeper layers (5-6) were maintained throughout the entirety of the brain, across all 40 2D sections, with average fraction of correctly aligned cells at 0.82 and

0.76 (Supplementary Figure 2). Similar to within the hemi-brain alignments, layer 4, in contrast, exhibited lower accuracy particularly towards the rostral and caudal poles of the dataset, likely as a result of cell typing and alignment error. As indicated in Section 2.4, not only was there variation in cell typing, naming cells physically within certain layers with type corresponding to a different layer, but the layer 4 cell type was specifically equivocally designated as a layer 4-5 type compared with the singly layer 4 CCFv3 region.

Supplementary Figure Caption: Accuracy of whole brain BARseq dataset mapped to CCFv3. A shows fraction of correctly aligned cells according to cell type within each of the cortical layers (2/3, 4, 5, and 6) across the caudal-rostral axis of the brain (according to CCFv3 coordinates). B shows total fraction of correctly aligned cells within each of the 4 layer designations. Total cells of each type for which accurate alignment was assessed are: 225609 (L2/3), 147956 (L4), 134837 (L5), 287531 (L6).

- The results presented in Figures 1 and 2 demonstrate whole tissue volume to subvolume mapping when given consecutive sections. However, does the alignment perform well even if the sections are disjoint?

5. We thank the reviewer for their inquiry. To address this question, we first note that in our examples of subvolumes, we do not have entire sets of consecutive sections but instead have drop out of 1-2 sections typically as a result of poor tissue capture. To emphasize this point, we have also shown an example of removing more of the slices and showing mapping between this further incomplete dataset and how it compares to mapping with the whole dataset. We specifically removed 7 sections of data from a total of ~ 40 sections to reduce the dataset by about 20%. We included the results of this mapping as a new supplementary figure, highlighting the stability of our technology through its ability to estimate a similar mapping of CCFv3 atlas to molecular target even with the removal of data sections. The specific Figure caption and description added in Section 2.4 of the text are copied below.

Notably, in addition to measured tissue volumes not needing to span entire volumes of atlases, they also need not be sampled uniformly by the given technology. Indeed, we demonstrate the robustness of our mapping technology for handling disjoint (rather than contiguous) sets of sections with gaps of missing data within the prescribed measured tissue volume (Supplementary Figure 1). In this setting, unlike in the censoring method here described, we choose not to model these gaps explicitly as unmeasured areas with gaps occurring typically from single or multi-slice dropout and not prescribed apriori by a given experimental setup. Rather, the reduced dataset is used to estimate a similar optimal deformation of the CCFv3 atlas to the whole brain BARseq sections (Supplementary Figure 1B versus Figure 3E). The stability of our technologies to estimate maps with full versus fragmented datasets is evidenced by only the slight differences in the extent of deformation observed through comparison of the determinant of the jacobian (Supplementary Figure ??F versus G), and with the near identical appearance of the estimated atlas intersection per a single slice (Supplementary Figure 1D versus E).

Caption: Comparison of CCFv3 atlas mapping to entire set of whole brain BARseq cell slices versus a reduced disjoint set with missing areas of data. A shows initial

alignment of whole brain BARseq slices of cells with removal of 20% of data (8 slices out of 40). B shows deformed CCFv3 atlas to BARseq data with support weights estimated to encompass block of tissue BARseq slices span (including areas of missing slices). C shows single slice of BARseq data. D,E show cross section through deformed CCFv3 atlas at plane of tissue section in C following deformation of CCFv3 atlas to whole set of BARseq slices (D) versus reduced set with missing slices (E). F,G show determinant of the jacobian for cross sections in D,E reflecting areas of expansion (red) and contraction (blue) of estimated deformation of CCF atlas to whole set of BARseq slices (F) versus reduced set (G).

References

- [1] Stouffer, K.M., Trouvé, A., Younes, L., Kunst, M., Ng, L., Zeng, H., Anant, M., Fan, J., Kim, Y., Chen, X., Rue, M., Miller, M.I.: Cross-modality mapping using image varifolds to align tissue-scale atlases to molecular-scale measures with application to 2d brain sections. *Nature Communications* **15**, 3530 (2024)
- [2] Clifton, K., Anant, M., Aihara, G., Atta, L., Aimiwu, O.K., Kebschull, J.M., Miller, M.I., Tward, D., Fan, J.: Stalign: Alignment of spatial transcriptomics data using diffeomorphic metric mapping. *Nature communications* **14**(1), 8123 (2023)
- [3] Miller, M.I., Tward, D., Trouvé, A.: Molecular Computational Anatomy: A Unified Molecular to Tissue Continuum Via Measure Representations. *BME Frontiers* (2022)
- [4] Miller, M.I., Trouvé, A., Younes, L.: Space-feature measures on meshes for mapping spatial transcriptomics. *Medical image analysis* **93**, 103068 (2024)
- [5] Zahedi, R., Ghamsari, R., Argha, A., Macphillamy, C., Beheshti, A., Alizadehsani, R., Lovell, N.H., Lotfollahi, M., Alinejad-Rokny, H.: Deep learning in spatially resolved transcriptomics: a comprehensive technical view. *Briefings in Bioinformatics* **25**(2), 082 (2024) <https://doi.org/10.1093/bib/bbae082>
- [6] Zeng, Y., Yin, R., Luo, M., Chen, J., Pan, Z., Lu, Y., Yu, W., Yang, Y.: Identifying spatial domain by adapting transcriptomics with histology through contrastive learning. *Briefings in Bioinformatics* **24**(2), 048 (2023)
- [7] Zhao, Y., Long, C., Shang, W., Si, Z., Liu, Z., Feng, Z., Zuo, Y.: A composite scaling network of efficientnet for improving spatial domain identification performance. *Communications Biology* **7**(1), 1567 (2024)
- [8] Schaffer, L.V., Ideker, T.: Mapping the multiscale structure of biological systems. *Cell systems* **12**(6), 622–635 (2021)
- [9] Miller, M., Tward, D., Trouvé, A.: Molecular computational anatomy: Unifying the particle to tissue continuum via measure representations of the brain. *BME*

- [10] Thompson, D.A.W., Bonner, J.T.: *On Growth and Form*. Cambridge paperbacks. Cambridge University Press, UK (1992)
- [11] Yao, Z., Velthoven, C.T., Kunst, M., Zhang, M., McMillen, D., Lee, C., Jung, W., Goldy, J., [...], Dee, N., Sunkin, S.M., Esposito, L., Hawrylycz, M.J., Waters, J., Ng, L., Smith, K.A., Bosiljka, T., Zhuang, X., Zeng, H.: A high-resolution transcriptomic and spatial atlas of cell types in the whole mouse brain. *bioRxiv*, 2023-03 (2023)
- [12] Gandin, V., Kim, J., Yang, L.-Z., Lian, Y., Kawase, T., Hu, A., Rokicki, K., Fleishman, G., Tillberg, P., Castrejon, A.A., et al.: Deep-tissue transcriptomics and subcellular imaging at high spatial resolution. *Science*, 2084 (2025)
- [13] Graham, E., Moss, J., Burton, N., Roochun, Y., Armit, C., Richardson, L., Baldock, R., et al.: *ehistology kaufman atlas plate 39d image c* (2023)
- [14] Kaufman, M.H., Bard, J.B.: *The Anatomical Basis of Mouse Development*. Gulf Professional Publishing, USA (1999)
- [15] Schott, M., León-Periñán, D., Splendiani, E., Strenger, L., Licha, J.R., Pentimalli, T.M., Schallenberg, S., Alles, J., Tagliaferro, S.S., Boltengagen, A., et al.: Open-st: High-resolution spatial transcriptomics in 3d. *Cell* **187**(15), 3953–3972 (2024)
- [16] Pang, M., Su, K., Li, M.: Leveraging information in spatial transcriptomics to predict super-resolution gene expression from histology images in tumors. *BioRxiv*, 2021-11 (2021)
- [17] Aitchison, J.: The statistical analysis of compositional data. *Journal of the Royal Statistical Society: Series B (Methodological)* **44**(2), 139–160 (1982)
- [18] Xia, C.-R., Cao, Z.-J., Tu, X.-M., Gao, G.: Spatial-linked alignment tool (slat) for aligning heterogenous slices properly. *bioRxiv*, 2023-04 (2023)
- [19] Jones, A., Townes, F.W., Li, D., Engelhardt, B.E.: Alignment of spatial genomics data using deep gaussian processes. *Nature Methods*, 1-9 (2023)
- [20] Graves, A.R., Roth, R.H., Tan, H.L., Zhu, Q., Bygrave, A.M., Lopez-Ortega, E., Hong, I., Spiegel, A.C., Johnson, R.C., Vogelstein, J.T., Tward, D.J., Miller, M.I., Huganir, R.L.: Visualizing synaptic plasticity in vivo by large-scale imaging of endogenous ampa receptors. *eLife* **10**, 66809 (2021) <https://doi.org/10.7554/eLife.66809>
- [21] Tward, D.J., Sicat, C.S., Brown, T., Bakker, A., Gallagher, M., Albert, M., Miller, M., Initiative, A.D.N., et al.: Entorhinal and transentorhinal atrophy in mild cognitive impairment using longitudinal diffeomorphometry. *Alzheimer's &*

Dementia: Diagnosis, Assessment & Disease Monitoring **9**, 41–50 (2017)

- [22] Kulason, S., al: Cortical thickness atrophy in the transentorhinal cortex in mild cognitive impairment. *NeuroImage: Clin.* **21**, 101617 (2019) <https://doi.org/10.1016/j.nicl.2018.101617>
- [23] Stouffer, K.M., Chen, C., Kulason, S., Xu, E., Witter, M.P., Ceritoglu, C., Albert, M.S., Mori, S., Troncoso, J., Tward, D.J., Miller, M.I.: Early amygdala and erc atrophy linked to 3d reconstruction of rostral neurofibrillary tau tangle pathology in alzheimer’s disease. *NeuroImage: Clinical* **38**, 103374 (2023) <https://doi.org/10.1016/j.nicl.2023.103374>